# Newly produced synaptic vesicle proteins are preferentially used in synaptic transmission

Sven Truckenbrodt[1,2,3,*,†], Abhiyan Viplav[1,2,4,‡], Sebastian Jähne[1,2,5], Angela Vogts[6] , Annette Denker[1,2], Hanna Wildhagen[1,2], Eugenio F Fornasiero[1,2,**]  & Silvio O Rizzoli[1,2,***]

## Abstract

Aged proteins can become hazardous to cellular function, by accumulating molecular damage. This implies that cells should preferentially rely on newly produced ones. We tested this hypothesis in cultured hippocampal neurons, focusing on synaptic transmission. We found that newly synthesized vesicle proteins were incorporated in the actively recycling pool of vesicles responsible for all neurotransmitter release during physiological activity. We observed this for the calcium sensor Synaptotagmin 1, for the neurotransmitter transporter VGAT, and for the fusion protein VAMP2 (Synaptobrevin 2). Metabolic labeling of proteins and visualization by secondary ion mass spectrometry enabled us to query the entire protein makeup of the actively recycling vesicles, which we found to be younger than that of non-recycling vesicles. The young vesicle proteins remained in use for up to ~ 24 h, during which they participated in recycling a few hundred times. They were afterward reluctant to release and were degraded after an additional ~ 24–48 h. We suggest that the recycling pool of synaptic vesicles relies on newly synthesized proteins, while the inactive reserve pool contains older proteins.

**Keywords** organelle aging; protein aging; synaptic vesicle; turnover; vesicle pools

**Subject Categories** Neuroscience

**The EMBO Journal (2018) 37: e98044**

## Introduction

Dividing cells are continuously replaced in their entirety and often remain fully functional even at the end of an organism's lifespan. In contrast, non-dividing cells, including neurons, accumulate damage to their organelles, which can manifest itself as disease (Sheldrake, 1974; Terman *et al*, 2007). To prevent detrimental effects, these cells need to avoid the use of aged and damaged organelles, which could disrupt cellular function in unpredictable ways. Such mechanisms have been described, for example, in budding yeast, where old and damaged mitochondria and vacuoles are retained in the aging mother cell, to prevent their usage in the budding daughter cells (Nyström & Liu, 2014).

Synaptic transmission, which relies on the precise release of a handful of organelles at any one time (Harata *et al*, 2001; de Lange *et al*, 2003; Richards *et al*, 2003; Denker *et al*, 2011a), could be severely disrupted by aged and damaged vesicles. For example, if released vesicles fail to recycle correctly, they might fail to liberate the active zone, which would result in a persistent inhibition of the synapse (Haucke *et al*, 2011). This suggests that neurons should have mechanisms in place that either block aged vesicle from release or promote the release of young vesicles.

Some evidence in support of this hypothesis exists in non-neuronal secretory cells, including chromaffin cells (Duncan *et al*, 2003) and pancreatic β-cells (Ivanova *et al*, 2013). Here, newly synthesized dense-core vesicles appear to be more release-prone than aged ones, since the former are released during mild (presumably more physiological) cellular activity, while the latter can only be exocytosed after heavy artificial stimulation. The situation affecting synaptic vesicles, which recycle repeatedly within the synaptic boutons, is less clear. Here, the vesicles can be broadly separated into (i) an active recycling pool, which includes the readily releasable vesicles that are docked at the release sites, as well as a surface pool of vesicle molecules that are in the process of recycling, and (ii) an inactive reserve pool that participates little in release under most stimulation conditions, and can typically only be released by heavy electrical stimulation (Rizzoli & Betz, 2005), by pharmacological manipulations (Kim & Ryan, 2010), or by the use of drugs that substantially increase presynaptic activity (Frontali

1 Institute for Neuro- and Sensory Physiology, University Medical Center Göttingen, Göttingen, Germany
2 Center for Biostructural Imaging of Neurodegeneration (BIN), University Medical Center Göttingen, Göttingen, Germany
3 International Max Planck Research School for Molecular Biology, Göttingen, Germany
4 Master Molecular Biology Programme, University of Vienna, Vienna, Austria
5 International Max Planck Research School for Neurosciences, Göttingen, Germany
6 Leibniz Institute for Baltic Sea Research, Warnemünde, Germany
 *Corresponding author. Tel: +43 2243 9000 2063; E-mail: strucke@gwdg.de
 **Corresponding author. Tel: +49 551 39 5911; E-mail: efornas@gwdg.de
 ***Corresponding author. Tel: +49 551 39 5930; E-mail: srizzol@gwdg.de
 †Present address: IST Austria, Klosterneuburg, Austria
 ‡Present address: Cells in Motion Cluster of Excellence at the University of Münster, Münster, Germany

*et al*, 1976). The inactive vesicles may be under the control of a fine balance between CDK5 and calcineurin activities (Kim & Ryan, 2010), with the former involved in the inactivation of the vesicles and with the latter involved in their activation.

It is currently unclear whether the situation observed with non-recycling dense-core vesicles is mirrored in synaptic boutons: Does the reserve pool represent an aged population? One contentious issue complicates such an interpretation: the problem of the vesicle identity (Ceccarelli & Hurlbut, 1980; Haucke *et al*, 2011; Rizzoli, 2014), for which two opposing models have been presented. In one model, the vesicle maintains its protein composition after exocytosis, as a single patch of molecules on the plasma membrane, which is then retrieved as a whole by endocytosis. In the other model, the vesicle loses its molecular cohesion upon fusion, and its proteins diffuse in the plasma membrane and intermix with other vesicle proteins, before endocytosis. In the first scenario, the neuron could readily target old vesicles for removal. This is less obvious in the second scenario, since vesicles lose their identity by intermixing molecules, which makes it more difficult to pinpoint the old vesicle components.

Irrespective of how this scenario plays out, however, the synapse would still need to identify old vesicle components and would need to remove them from functional reactions. We set out to test this, by studying the age of the proteins used by synaptic vesicles in active recycling. We relied on a large number of imaging assays, along with non-optical secondary mass spectrometry imaging. We found that the recycling vesicle proteins are metabolically younger than those of non-recycling reserve vesicles, in hippocampal cultured neurons. Vesicle proteins such as Synaptotagmin 1, VGAT, and VAMP2/Synaptobrevin 2 were preferentially used in their young age and were removed from functional reactions 1–2 days before their eventual degradation. They spent the intervening time in an inactive vesicle pool that has some of the hallmarks of the reserve pool: vesicles that did not recycle during normal activity, but could eventually be triggered to release when the general excitability of the synapses was increased.

# Results

## Synaptotagmin 1 antibodies label clusters of Synaptotagmin 1 molecules on the plasma membrane after exocytosis, which co-localize with Synaptophysin

To test the synaptic vesicle protein aging, we set out to use pulse-chase experiments, in which proteins were tagged using antibodies directed against their lumenal domains. Such antibodies only detect the molecules after exocytosis. To test the antibody performance, before proceeding with further experiments, we analyzed the organization of the tagged proteins after exocytosis, on the plasma membrane. We used hippocampal neurons that were mildly stimulated, by depolarizing for 6 min with 15 mM KCl, in the presence of fluorophore-conjugated antibodies directed against the lumenal domains of the vesicle protein Synaptotagmin 1 (Matteoli *et al*, 1992; Kraszewski *et al*, 1995), which reveal Synaptotagmin 1 molecules present on the plasma membrane. To ensure that the antibodies only revealed Synaptotagmin 1 molecules that were exocytosed during stimulation, and not epitopes that were already present on the plasma membrane, we blocked these surface epitopes by incubating the

neurons with unconjugated (non-fluorescent) antibodies before applying both the stimulation pulse and the fluorophore-conjugated antibodies (Hoopmann *et al*, 2010). The fluorophore-conjugated antibodies could thus only bind to the Synaptotagmin 1 molecules from newly exocytosed vesicles. To determine whether such molecules diffused away from each other, or from other vesicle proteins, we fixed the neurons and immunostained for another synaptic vesicle marker, Synaptophysin, using an antibody that recognized the lumenal domain of Synaptophysin (Hoopmann *et al*, 2010). The immunostaining was performed without permeabilization, so that only surface Synaptophysin molecules could be detected.

We then quantified the co-localization of Synaptotagmin 1 and Synaptophysin on the membrane surface by two-color super-resolution STED microscopy (Hoopmann *et al*, 2010; Appendix Fig S1A). We found that the majority of Synaptotagmin 1 and Synaptophysin signals co-localized, indicating that the molecules are found in synaptic vesicle protein assemblies or co-clusters. Only a small proportion of the Synaptotagmin 1 signals was found away from the Synaptophysin signals. We quantified the total intensity of this fraction of molecules and found that it was around 3% (Appendix Fig S1B and C). We therefore conclude, in line with the previous literature, that the antibodies label co-clusters of vesicle proteins (presumably synaptic vesicles waiting for endocytosis) and are only rarely found in areas that do not show the clear presence of bona fide synaptic vesicle markers.

## Antibody-labeled synaptic vesicles proteins are lost from synapses over a timescale of several days

To follow synaptic vesicle proteins through their life cycle in the synapse, we reasoned that a similar use of antibodies directed against the lumenal domain of synaptic vesicle proteins could reveal the localization and activity status of such vesicle molecules for substantial time periods. We started with the same fluorophore-conjugated Synaptotagmin 1 antibodies, as above. The tagging of recycling epitopes is highly specific, as antibodies not directed against surface proteins of the neurons are not taken up efficiently by the neurons, resulting in essentially null backgrounds (Appendix Fig S2). The antibodies remained bound to fixed neurons for up to 10 days, with no noticeable loss of signal, even when incubated with a 100× molar excess of antigenic peptide at pH 5.5 (the pH encountered inside synaptic vesicles by the antibody in living neurons; Appendix Fig S3). In this experiment, the antibodies that come off their targets would immediately bind the antigenic peptides and would not be able to find the cellular targets again. This did not happen, implying that the antibodies bind stably to their targets and are not affected by the vesicular pH. At the same time, these antibodies (this monoclonal clone, 604.2) have been used in multiple experiments in synaptic vesicle recycling and appear not to interfere with it (Wienisch & Klingauf, 2006; Willig *et al*, 2006; Hoopmann *et al*, 2010; Opazo *et al*, 2010; Hua *et al*, 2011). This was confirmed by the fact that virtually all tagged molecules could be induced to participate in exocytosis upon 1,200 AP stimulation (Appendix Fig S4).

To check the behavior of the antibody-labeled molecules, we applied the Synaptotagmin 1 antibodies to living, active rat primary hippocampal cultures at 15 days *in vitro* (Fig 1A). The antibodies, which were conjugated to the bright, pH-insensitive fluorophore

Atto647N, tagged recycling synaptic vesicle proteins during the vesicle recycling that occurs during spontaneous network activity of the mature cultures (antibody incubation day 0; Fig 1B, leftmost panel). Incubating the neurons with the antibodies for 1 h was sufficient to saturate the epitopes in the actively recycling pool of vesicles (Appendix Fig S5); further stimulation was unable to trigger significant additional labeling (Appendix Fig S5), suggesting that the entire recycling pool was indeed labeled.

In the conditions of labeling that we used (1 h of antibody incubation under the spontaneous network activity of the cultures at 37°C), we noted that the antibodies correlated well with the synaptic vesicle marker Synaptophysin, both upon labeling and 4 days later (Appendix Fig S6). At the same time, we observed no significant co-localization with endosomal markers (Rab5, Rab7) or with a dense-core vesicle marker (Chromogranin A; Appendix Fig S6).

To quantify the proportion of the Synaptotagmin 1 molecules that were labeled by this procedure, we compared the resulting images with those of neuronal cultures that were immunostained for Synaptotagmin 1 with the same antibody, after fixation and permeabilization, and in which all epitopes were thus revealed. This showed that the live labeling procedure revealed approximately 50% of the Synaptotagmin 1 epitopes present in synapses (Appendix Fig S7). This is in agreement with previous findings on the size of the recycling pool in these cultures (Rizzoli & Betz, 2005). Approximately half of the labeled epitopes were found on the surface membrane, presumably waiting to be endocytosed, again in agreement with previous findings (Wienisch & Klingauf, 2006).

We next investigated the behavior of the labeled Synaptotagmin 1 molecules over time. We labeled a series of neuronal sister cultures at one time point and then removed coverslips for fixation after different time intervals of up to 10 days. We then immunostained the neurons for Synaptophysin, an optimal marker for synaptic vesicles (Takamori *et al*, 2006; and thereby for synaptic boutons), and analyzed the intensity of the Synaptotagmin 1 antibody signals within the Synaptophysin-marked boutons. We found that the Synaptotagmin 1 signals decreased slowly (Fig 1B and D), albeit the majority of the signal remained within synaptic boutons (Appendix Fig S8). Within ~ 2–4 days, the Synaptotagmin 1 signals also appeared in the cell bodies, in large acidified organelles

(Appendix Fig S9A) that co-localized well with Lysotracker (Appendix Fig S9B), a small molecule that enriches in organelles with low pH and is especially fluorescent in lysosomes. This suggests that antibody-tagged Synaptotagmin 1 molecules were degraded through a lysosomal route, in agreement with the literature (Rizzoli, 2014). This suggestion was further confirmed by the observation that inhibiting lysosomal degradation using leupeptin reduced the loss of Synaptotagmin 1 from boutons (Appendix Fig S9C). The rate of Synaptotagmin 1 loss from synapses was well within the range previously measured for the degradation of synaptic vesicle proteins by mass spectrometry or by radioisotopic labeling (Daly & Ziff, 1997; Cohen *et al*, 2013).

To test whether this phenomenon was also observable for other vesicle proteins, we turned to antibodies against the vesicular GABA transporter, VGAT (Fig 1C). This is the only other vesicle protein for which lumenal antibodies are available (Martens *et al*, 2008). The Synaptophysin antibodies we used in Appendix Fig S1 do recognize the lumenal epitopes of this protein, but only after PFA fixation, and thus cannot be used in live labeling. The loss of VGAT antibodies from synapses paralleled that of Synaptotagmin 1 antibodies (Fig 1C and E).

### The antibody-labeled Synaptotagmin 1 molecules cease their involvement in synaptic activity as they age

To test whether the tagged proteins of different ages were still involved in synaptic activity, we employed antibodies conjugated to the pH-sensitive fluorophore CypHer5E (Martens *et al*, 2008; Hua *et al*, 2011). We used a stimulation protocol designed to trigger the release of the entire population of recycling vesicles, 600 action potentials at 20 Hz (Wilhelm *et al*, 2010), and measured the fraction of the CypHer5E-labeled proteins that were releasable (Fig 2A). This was monitored through imaging the stimulation-induced reduction in CypHer5E fluorescence, which is quenched at neutral pH and therefore becomes invisible upon exocytosis. We found that the fraction of the antibody-labeled proteins that could be induced to release gradually decreased with every passing day after tagging (Fig 2B and C), until almost no response could be elicited any more, at 7–10 days after tagging (Fig 2D and E). This observation was made for both

**Figure 1. Antibody-tagged Synaptotagmin 1 and VGAT molecules are lost from synapses over a few days.**

A  To determine the time interval that synaptic vesicle proteins spend in synapses, we incubated living hippocampal cultured neurons with fluorophore-conjugated (green stars) antibodies against the lumenal domain of synaptic vesicle proteins. Fluorophore-conjugated Synaptotagmin 1 (Atto647N-conjugated) or VGAT (CypHer5E-conjugated) antibodies were applied (diluted 1:120 from 1 mg/ml stock) to the neurons in their own culture medium, for 1 h at 37°C in a cell culture incubator. The unbound antibodies were then washed off, and the neurons were placed again in the incubator. Individual coverslips were then chased and retrieved at different time intervals after labeling, fixed, and imaged, to determine the amount of Synaptotagmin 1 and VGAT labeling remaining in synapses.

B  Exemplary images of neurons labeled with Synaptotagmin 1 antibodies, imaged at different time points after labeling. Scale bar: 20 μm. All images were taken using a Leica SP5 confocal microscope.

C  Exemplary images of neurons labeled with VGAT antibodies, imaged at different time points after labeling. Scale bar: 20 μm. The images are taken after fixation and permeabilization, in buffers with pH 5.5, to enable the entire CypHer5E fluorescence to be detected. All images were taken using a Nikon Ti-E epifluorescence microscope.

D  The loss of synaptic vesicle proteins from the synapse was monitored by imaging the Synaptotagmin 1 antibody fluorescence at serial time points after tagging Synaptotagmin 1 (*n* = 3, 2, 3, 3, and 2 independent experiments per respective time point, at least 10 neurons sampled per experiment). To determine fluorescence intensities specifically in synapses, the neurons were immunostained for the synaptic vesicle marker Synaptophysin, and only the fluorescence found within Synaptophysin spots (synapses) was analyzed. We did not observe any significant changes in the co-localization of fluorophore-conjugated antibodies in respect to Synaptophysin over time (Appendix Fig S8).

E  Similar intensity analysis for VGAT (*n* = 3, 4, 2, 4, 4, and 3 independent experiments per respective time point, at least 10 neurons sampled per experiment).

Data information: All data represent the mean ± SEM.
Source data are available online for this figure.

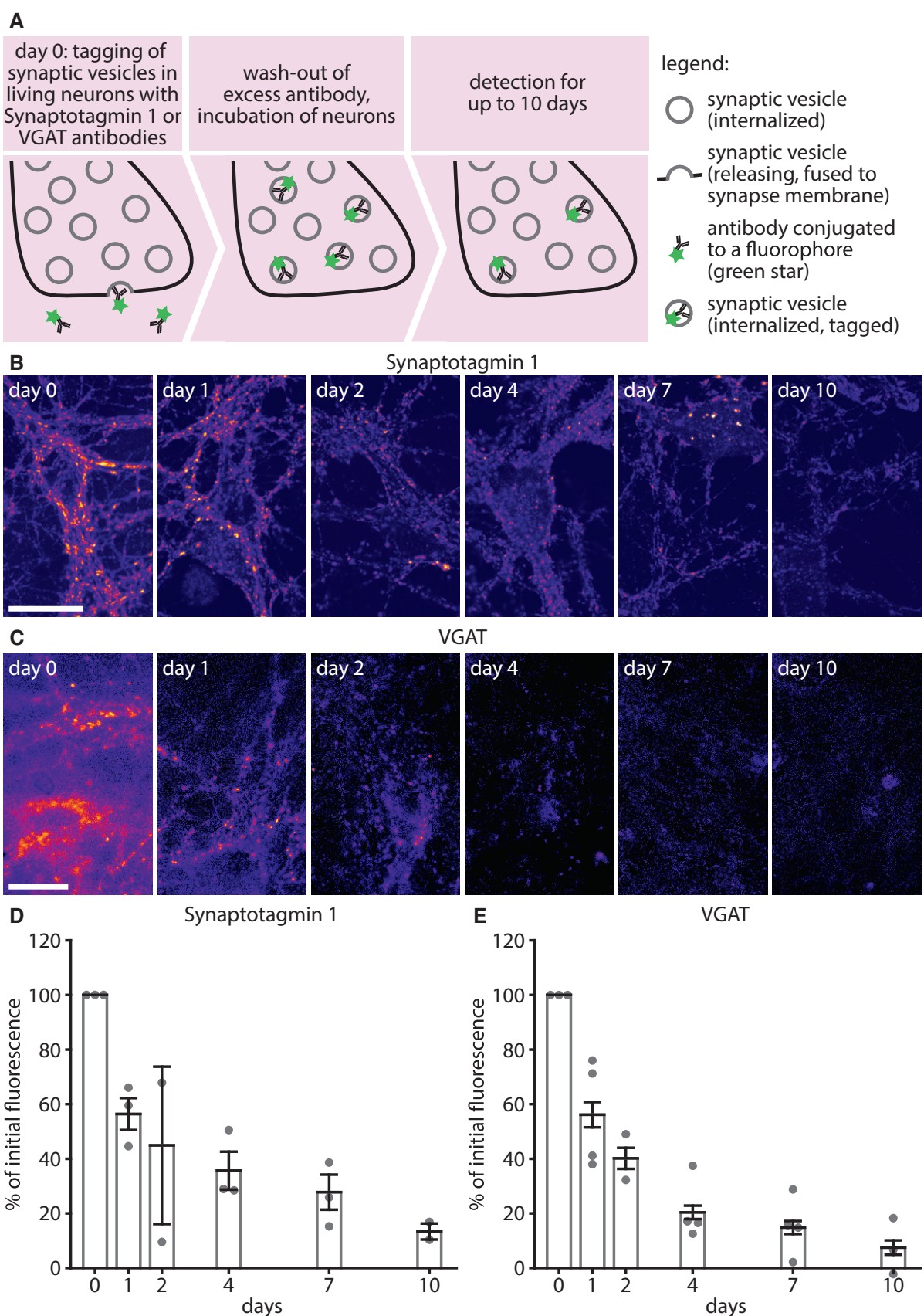

Figure 1.

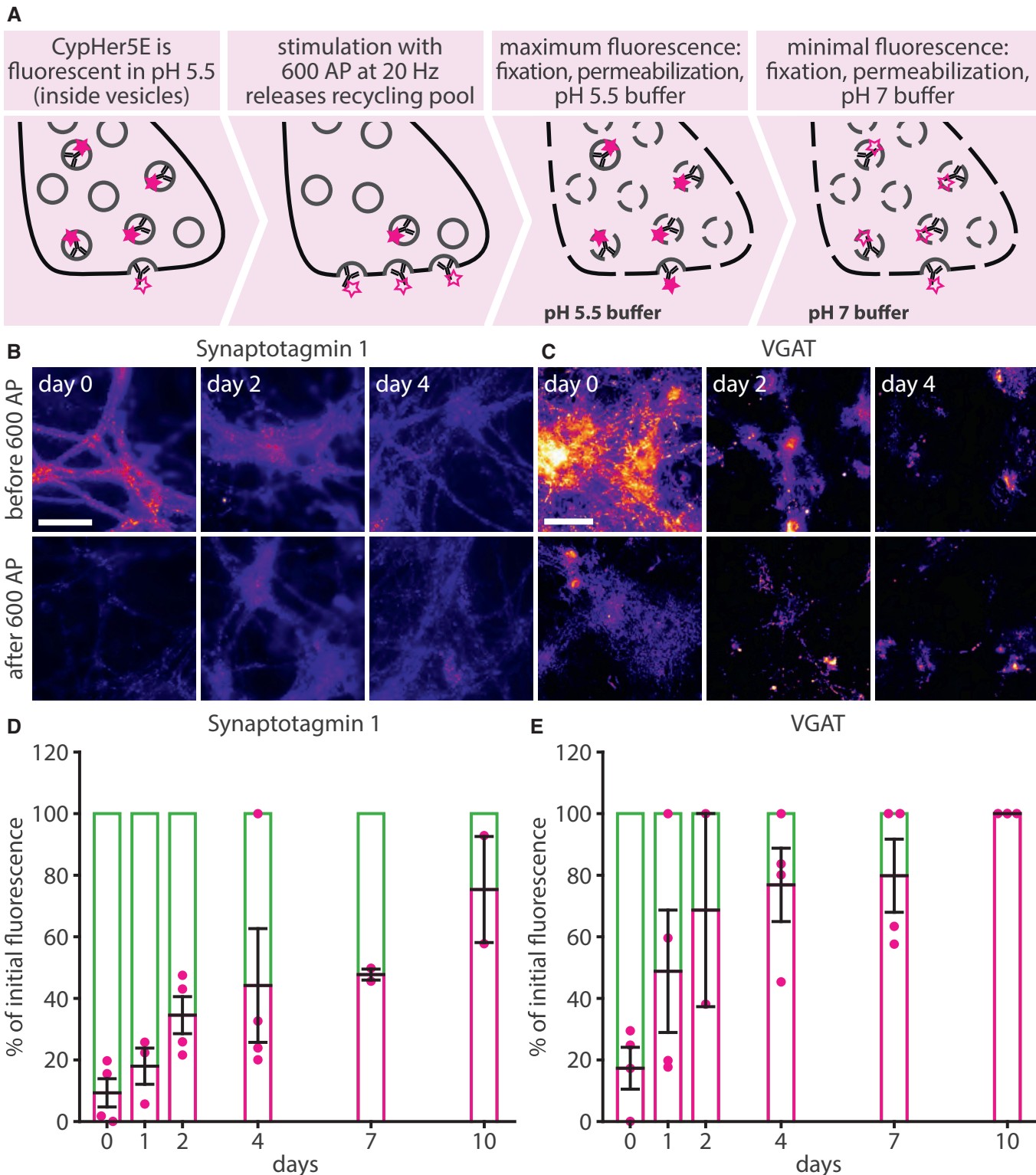

**Figure 2.**

Synaptotagmin 1 and VGAT. A trivial explanation for this observation would be that the cultured neurons became slowly damaged during the 7–10 days of the experiment, simply by aging. This, however, was not the case, as the neurons were still active throughout the entire study period, and recycled identical amounts of vesicles (Appendix Fig S10). Also, no changes in the bouton size, vesicle amounts, or bouton density could be found over time (Appendix Fig S11).

**Figure 2.  Aging antibody-labeled synaptic vesicle proteins cease to participate in exocytosis.**

A  To determine whether the antibody-tagged synaptic vesicle proteins that are present in synapses at different time intervals after labeling are exocytosis-competent, we labeled neurons with CypHer5E-conjugated antibodies for Synaptotagmin 1 or VGAT for 1 h exactly as described in Fig 1A. At different time intervals after labeling, individual cultures were removed from the incubator, mounted in a live-stimulation chamber, and subjected to a stimulation paradigm designed to release the entire recycling pool (20 Hz, 30 s; Wilhelm *et al*, 2010). Exocytosis is determined by monitoring the fluorescence of the pH-sensitive fluorophore CypHer5E, which is fluorescent at low pH within vesicles, but is quenched by exposure to the neutral extracellular pH. To determine the proportion of the labeled vesicles that is still able to exocytose, we measured the fluorescence signal corresponding to all labeled vesicles by fixing and permeabilizing the neurons and then exposing them to a pH of 5.5, obtained with TES-buffered solutions.

B, C  Typical images of fluorescence either before or after stimulation, for Synaptotagmin 1 (B) or VGAT (C). Numerous individual images were taken of different fields for every condition and were averaged for the statistics, as imaging the same field under all conditions proved unfeasible, due to the high bleaching tendency of CypHer5E. Scale bar: 50 µm. All images in (B) were taken using an Olympus epifluorescence microscope, and all images in (C) were taken using a Nikon Ti-E epifluorescence microscope.

D, E  The following numbers of experiments were quantified: for Synaptotagmin 1, $n$ = 4, 3, 4, 4, 2, and 2 independent experiments per respective time point, with at least 12 neurons sampled per experiment; and for VGAT, $n$ = 4, 4, 2, 4, 4, and 3 independent experiments per respective time point, with at least eight neurons sampled per experiment. The VGAT signal follows the same approximate dynamic as Synaptotagmin 1, but is faster. VGAT is present in only ~ 5–10% of the neurons in our cultures, and therefore, this difference may reflect a cell type-specific effect. All data represent the mean ± SEM.

Source data are available online for this figure.

We also investigated this phenomenon during spontaneous network activity, focusing on Synaptotagmin 1 (since the low proportion of VGAT-positive neurons in the culture, ~ 5–10%, is likely not representative for the general activity of our cultures). For this, we performed a different experiment (Appendix Fig S12). Spontaneous network activity is difficult to measure via CypHer5E, since it does not yield the signal-to-noise required to observe single release events triggered by the network activity of the cultures, and is prone to bleaching during prolonged observation times. We tagged live the Synaptotagmin 1 molecules with non-fluorescently conjugated antibodies at one time point (as in Fig 1A), and then, after intervals of up to 48 h, we applied Cy5-conjugated secondary antibodies onto the living cultures. These antibodies reveal all of the Synaptotagmin 1 antibodies that are exposed during vesicle recycling. After incubating the living cultures with the secondary antibodies for 1 h, we fixed and permeabilized them, and applied Cy3-conjugated secondary antibodies, to thereby reveal all the Synaptotagmin 1 antibodies, both those recycling and non-recycling (Appendix Fig S12A). This procedure thus reveals the proportion of the tagged Synaptotagmin 1 molecules that are still recycling in the synapses under normal activity (Cy5, green in Appendix Fig S12), as well as the whole amount that is present in the synapses (Cy3, magenta in Appendix Fig S12). The proportion of the molecules that participated in release decreased relatively fast, within about 1 day (time constant of ~ 0.4 days; Appendix Fig S12B and C). Although the tagged Synaptotagmin 1 molecules ceased to recycle rapidly, they were still in the synapses, from which they were lost with a longer time constant, of ~ 1.6 days (Appendix Fig S12B and C).

These experiments suggest that the tagged Synaptotagmin 1 molecules became rapidly inactive (from the point of view of exocytosis), albeit many still remained within synapses. The vesicles containing these molecules could be induced to release by strong stimulation (Fig 2D), but did not release under normal network activity (Appendix Fig S12B and C). For simplicity, we termed this the "inactive state" of synaptic vesicles, or vesicle proteins. It could also be called a "reserve" or "resting" state (Rizzoli & Betz, 2005; Denker & Rizzoli, 2010; Alabi & Tsien, 2012): The vesicles are present in the synapse, release upon *supra*-physiological stimulation, but not during normal activity.

**A genetically encoded sensor for the age of vesicle proteins confirms that young VAMP2 molecules are preferentially employed in exocytosis**

To complement these antibody-based approaches, we tested the behavior of the synaptic vesicle protein VAMP2 after tagging with a novel construct that enables the separate identification of newly synthesized or older proteins. We expressed VAMP2 coupled to a SNAP tag on the lumenal side (Keppler *et al*, 2003, 2004), and separated by a TEV protease cleavage site: VAMP2-TEV-SNAPtag (Appendix Fig S13A). This construct should be minimally disruptive to physiological synaptic vesicle function, since VAMP2 is by far the most abundant synaptic vesicle protein (Takamori *et al*, 2006; Wilhelm *et al*, 2014), and therefore, every vesicle should still have ample levels of wild-type VAMP2, independent of the levels of VAMP2-TEV-SNAPtag expression. Moreover, our construct is designed on the basis of synaptopHluorin (VAMP2-pHluorin), which is known to target and function well in neurons (Miesenböck *et al*, 1998; Gandhi & Stevens, 2003).

The construct can be expressed, ensuring that synaptic vesicles are tagged with it, but it is not inherently fluorescent. At the desired time point, the construct can be revealed by labeling with a membrane-permeable fluorophore, such as tetramethyl-rhodamine-Star (TMR-Star). The coupling reaction is self-catalyzed by the SNAP tag (Juillerat *et al*, 2003) and is highly efficient, as the application of a second SNAP-binding fluorophore immediately afterward resulted in no detectable signal. This, therefore, labels all SNAP-tagged proteins present at a particular time point in the cells. After allowing protein biosynthesis to proceed for a further 1–2 days, the newly produced, younger constructs can be revealed by labeling with a second fluorophore, such as 647-SiR, which is easily spectrally separable from TMR-Star. This thus reveals the younger VAMP2 molecules. The neurons now contain two populations of VAMP2-TEV-SNAPtag, one young, labeled with 647-SiR, and one that is 1–2 days older, labeled with TMR-Star (Appendix Fig S13B). To test whether young or old vesicles are preferentially used in recycling, one can incubate the neurons with a TEV protease, which cannot penetrate membranes and therefore cleaves the fluorescent tags only from vesicles that participate in recycling. Inactive vesicles are not affected by the TEV protease. As expected, both old TMR-Star-tagged and young 647-SiR-tagged populations were reduced by application of the TEV

protease, but the 647-SiR signals, arising from recently labeled molecules, were more strongly reduced, indicating that the tag was cleaved preferentially from young VAMP2 proteins (Appendix Fig S13B and C).

**The protein makeup of recycling vesicles newly labeled by Synaptotagmin 1 antibodies is significantly younger, as a whole, than that of vesicles labeled several days earlier**

Having verified with two independent techniques that newly synthesized protein copies are preferentially used in exocytosis for three vesicle proteins (Synaptotagmin 1, VGAT, and VAMP2), we proceeded to test whether this observation applies to the entire protein makeup of actively recycling vesicles. This type of procedure can be performed by incubating the cultures with a special amino acid, which is incorporated in all of the newly biosynthesized proteins and serves therefore as a marker for newly produced proteins. Two techniques are available for this type of tagging. First, cells can be fed with an unnatural amino acid, such as azidohomoalanine (AHA), which incorporates into newly produced proteins instead of methionines. AHA can be detected in fluorescence microscopy after fluorophore conjugation (FUNCAT; Dieterich *et al*, 2011), through a highly specific azide–alkyne reaction that has been termed CLICK chemistry (Rostovtsev *et al*, 2002). Second, the cells can be fed with an amino acid, such as leucine, that contains a stable rare isotope, such as $^{15}N$. The isotope can be detected through the mass spectrometry imaging technique nanoSIMS (Lechene *et al*, 2006; Steinhauser & Lechene, 2013), which has a higher resolution than conventional fluorescence microscopy (~ 50–100 nm in cultured neurons; Saka *et al*, 2014b).

We applied here both techniques, in separate experiments, tagging newly synthesized proteins either with AHA (Fig 3) or with leucine containing $^{15}N$ (Fig 4). We then incubated the cells with Synaptotagmin 1 antibodies to label the recycling vesicles, as in Fig 1 (Fig 3A). Alternatively, we incubated the neurons with the antibodies several days before feeding them with the special amino acids, to make sure that the tagged vesicles were already in the inactive state at the time of amino acid incorporation into newly

synthesized proteins (Fig 3B). We then imaged the recycling vesicles in fluorescence microscopy and correlated their positions with those of the AHA or $^{15}N$ signals, obtained in STED microscopy or in nanoSIMS, respectively (Saka *et al*, 2014b).

In both approaches, we could detect a significantly higher co-localization of the antibodies labeling actively recycling Synaptotagmin 1 molecules with newly synthesized proteins (Figs 3C–E and 4). Moreover, the higher sensitivity of nanoSIMS, which, unlike FUNCAT, detects simultaneously both the $^{14}N$ from old proteins and the $^{15}N$ from new proteins (Fig 4A), could demonstrate that the space surrounding the actively recycling Synaptotagmin 1 molecules, which presumably contains the rest of the synaptic vesicle makeup, contained significantly younger proteins than the rest of the axon. The opposite was true for the inactive vesicles (Fig 4B).

**Old Synaptotagmin 1 molecules that left the actively recycling pool of vesicles do not return to it under conditions of spontaneous network activity**

We then tested whether the labeled vesicle proteins, once inactivated, could return to the recycling pool under normal culture activity. Strong stimulation, as in Fig 2D, enables such proteins to participate in release, albeit they appeared not to do so under normal activity (Appendix Fig S12). To test this, we applied unconjugated lumenal Synaptotagmin 1 antibodies to the living neurons to saturate all of the epitopes of the releasable population (Fig 5A) and then followed this up with pulses of fluorophore-conjugated Synaptotagmin 1 antibody, to reveal new epitopes entering the active, releasable population (Fig 5B and C).

Such epitopes could come from multiple sources: newly synthesized vesicles *en route* to the synapses from the cell body, vesicle precursors similarly coming from the cell body, vesicle precursors already present in the synapses, or inactivated vesicles, whose epitopes are not affected by the initial saturation step with unconjugated antibodies (Fig 5A), and which account for ~ 50% of all of the vesicles in the synapse (Appendix Fig S7). Cutting off the supply of new synaptic vesicles delivered from the cell body, by blocking

**Figure 3.    Metabolic imaging based on the unnatural amino acid AHA reveals that the actively recycling Synaptotagmin 1 molecules are found within younger protein environments than the inactive ones.**

A, B   To label newly produced proteins, we used the unnatural amino acid AHA, which was detected in STED imaging. AHA was fed to the neurons in culture medium free of methionine, which is the amino acid AHA replaces during protein biogenesis. We used two experimental paradigms. First (A), to correlate the presence of newly synthesized proteins with recycling vesicles, we fed AHA to the neurons for 9 h (AHA), before labeling Synaptotagmin 1 from the recycling pool by applying Atto647N-conjugated antibodies for 1 h at 37°C, as in Fig 1A. Second (B), to compare the co-localization of newly synthesized proteins with inactive synaptic vesicles, we performed the same experiment, but with 3–4 days between antibody tagging and metabolic labeling: We first labeled the recycling vesicles by Synaptotagmin 1 antibody incubation, as in Fig 1A, and then returned the cell cultures to the incubator. We then waited for 4 days for the inactivation of the vesicles to take place, and then fed the neurons with AHA, to label newly synthesized proteins.

C   Using this approach, we analyzed the co-localization of AHA with releasable or inactive synaptic vesicles in STED microscopy. The neurons were fixed and subjected to a click chemistry procedure that reveals all AHA moieties, coupling them to the fluorophore Chromeo494 (a procedure known as FUNCAT; Dieterich *et al*, 2011). The neurons were then embedded in melamine resin and were sectioned into 20-nm sections on an ultramicrotome. The sections were then imaged by two-color STED microscopy. The images are shown after processing by deconvolution, to improve signal-to-noise ratios. The line scans indicated in light blue in the rightmost panels are plotted in (D). Scale bar: 2 μm.

D   Line scans through pairs of vesicles found close to each other are shown, as examples. The line scans were performed on the original raw data files, to eliminate any artifacts due to deconvolution.

E   The amount of AHA fluorescence co-localizing with the releasable or inactive vesicles was determined and was expressed as fold over baseline signals ($n = 3$ independent experiments per data point, at least 10 neurons sampled per experiment, *$P = 0.0037$, $t(4) = 6.09$). The baseline signal refers here to the average AHA signal within the respective synaptic boutons. All data represent the mean ± SEM. Statistical significance was evaluated using unpaired *t*-tests.

Data information: All images were taken using a Leica SP5 STED microscope.
Source data are available online for this figure.

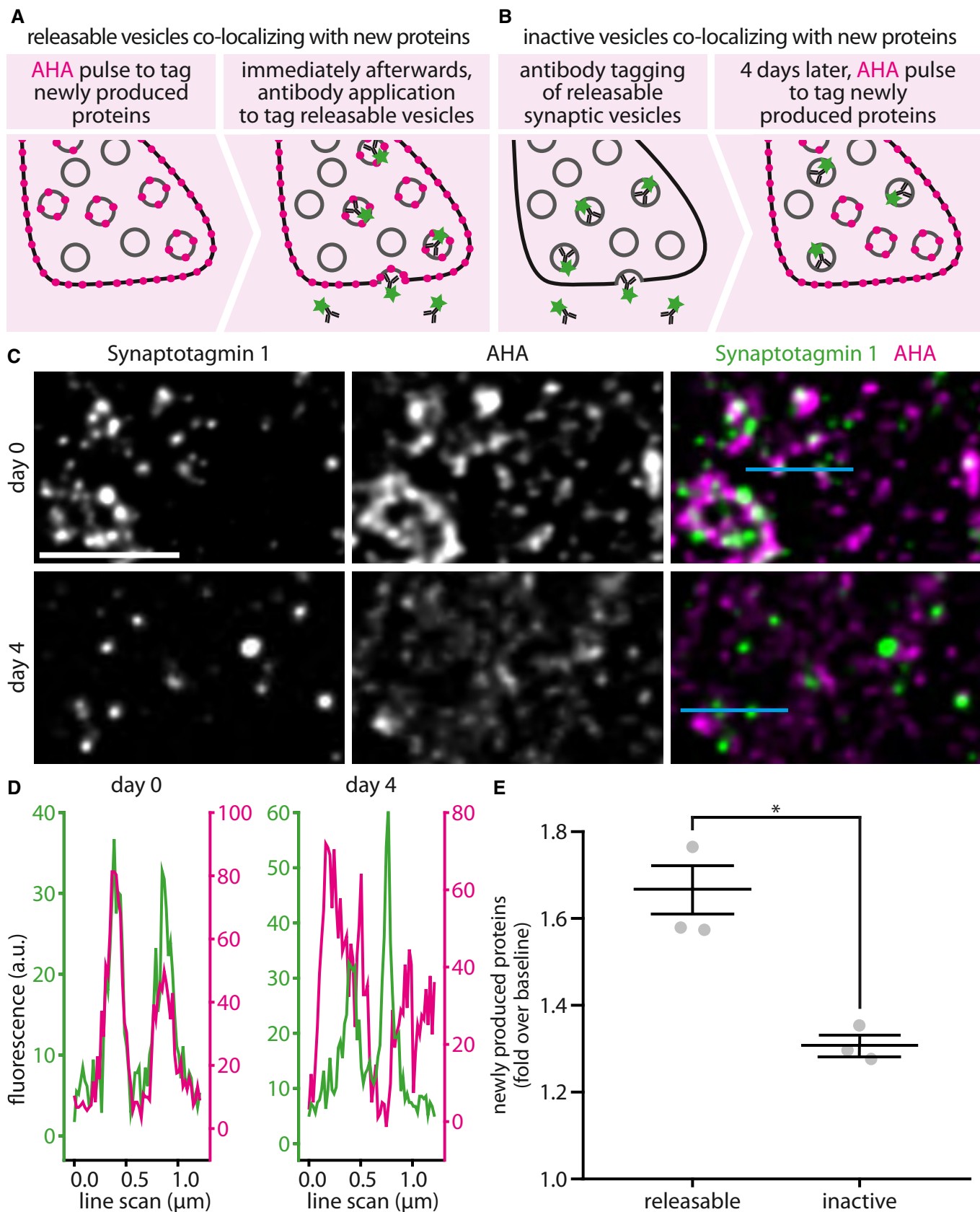

Figure 3.

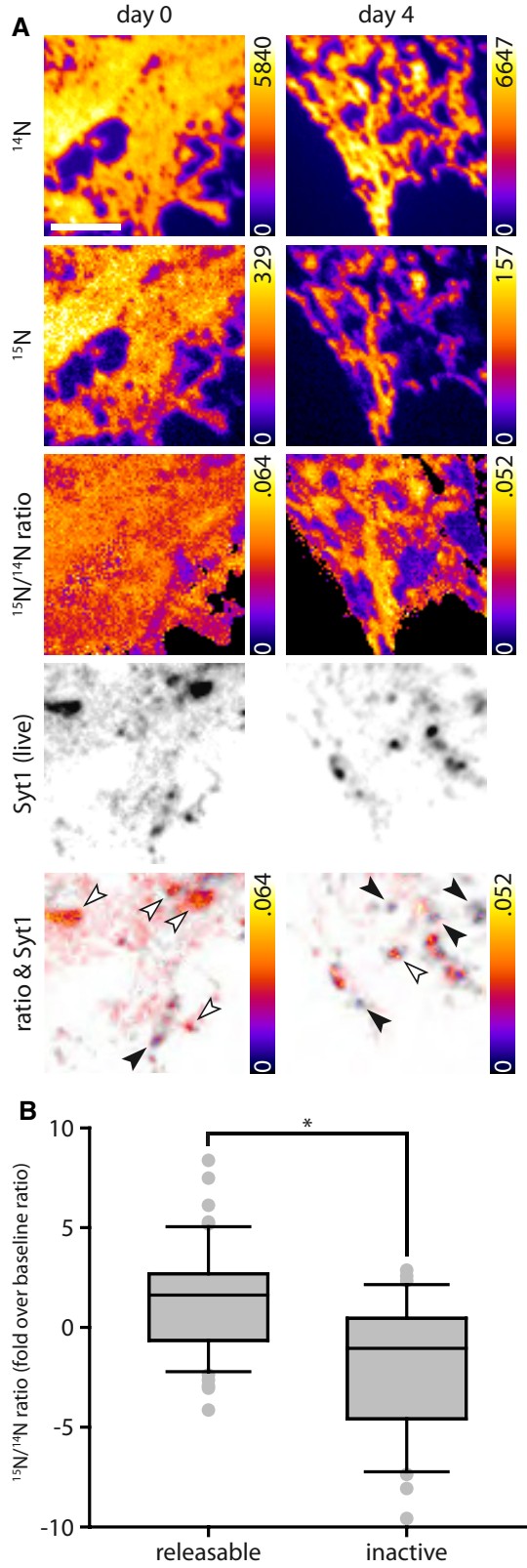

**A**

day 0    day 4

14N

15N

15N/14N ratio

Syt1 (live)

ratio & Syt1

**B**

*

$^{15}N/^{14}N$ ratio (fold over baseline ratio)

releasable    inactive

**Figure 4. Metabolic imaging based on $^{15}N$-leucine confirms that actively recycling Synaptotagmin 1 molecules are found within younger protein environments than the inactive ones.**

A   To label newly produced proteins, we used here $^{15}N$-leucine, which was detected in nanoSIMS imaging. $^{15}N$ leucine was fed to the cultured neurons in threefold molar excess over $^{14}N$ leucine, in their normal culture medium, for 24–72 h prior to processing the samples. The extended feeding time compared to AHA (Fig 3) was necessary to obtain a reliable signal in nanoSIMS imaging. We used the same experimental paradigms as in Fig 3. First, to correlate the presence of newly synthesized proteins with recycling vesicles, we fed $^{15}N$-leucine to the neurons for 24–72 h, before labeling before labeling Synaptotagmin 1 from the recycling pool, as in Figs 1A and 3. Second, to compare the co-localization of newly synthesized proteins with inactive synaptic vesicles, we performed the same experiment, but with 3–4 days between antibody tagging and metabolic labeling: We labeled the recycling vesicles by Synaptotagmin 1 antibody incubation and then returned the cell cultures to the incubator. We then waited for 4 days for the inactivation of the vesicles to take place and then fed the neurons with $^{15}N$-leucine, to label newly synthesized proteins. We analyzed the co-localization of $^{15}N$-leucine with releasable and inactive synaptic vesicles in correlated fluorescence and isotopic microscopy (COIN), as follows. The neurons were fixed and were embedded in LR White resin, a vinyl resin that is usable in both fluorescence and isotopic secondary ion mass spectrometry imaging. The samples were then sectioned into 200-nm sections on an ultramicrotome, as this thickness is ideal for secondary ion mass spectrometry. The sections were mounted on silicon wafers and were imaged on a Nikon Ti-E microscope, using 150× magnification, to detect the synaptic vesicles (black-on-white images). The same areas were then imaged in a nanoSIMS instrument, recording both the $^{15}N$ and $^{14}N$ signals (color images). The lowest panels show an overlay between the $^{15}N/^{14}N$ ratios and the Synaptotagmin 1 labeling. In this overlay, only the pixels with the highest value of the two overlaid images are indicated (showing color where the $^{15}N/^{14}N$ ratio images have the higher value; showing black-on-white where the fluorescence images of synaptic vesicles have the higher value). This means that wherever color is visible, synaptic vesicles co-localize well with newly produced proteins. Wherever black or gray is visible, synaptic vesicles co-localize poorly with newly produced proteins. The white arrowheads point to several vesicles co-localizing well with newly produced proteins. The black arrowheads point to several vesicles co-localizing poorly with newly produced proteins. Scale bar: 2 μm.

B   The $^{15}N/^{14}N$ ratio was then determined both within the vesicle areas and elsewhere and was then presented as fold over the baseline ratio in the axons. *n* = 57 synapses from three independent experiments for releasable synaptic vesicles, and *n* = 47 synapses from two independent experiments for inactive vesicles, *P = 0.0001, t(102) = 5.5378. The ratio is a direct indication of the amount of $^{15}N$-leucine in the vesicles. The inactive vesicles contain substantially fewer newly synthesized proteins than the rest of the axon (the $^{15}N/^{14}N$ ratio of which served as baseline; P = 0.0001, t(96) = 4.0691), while the releasable ones contain substantially more newly synthesized proteins (P = 0.0004, t(116) = 3.6156). Statistical significance was evaluated using unpaired t-tests. All data are represented as box plots with median and upper and lower quartile boundaries, plus 1.5 times inter-quartile range (whiskers) and outliers (dots).

Source data are available online for this figure.

protein biosynthesis with anisomycin or by disrupting the micro-tubule transport network with colchicine, completely removed the entry of new epitopes into the releasable population (Fig 5B and C).

These drugs were applied after the tagging with unconjugated anti-bodies, and remained present until application of the fluorophore-conjugated antibody.

The drugs did not significantly affect the proportion of releasable vesicles in the synapse, the spontaneous network activity, or the total amount of vesicles per bouton (Fig 5D–F), suggesting that the neurons were still healthy at the time of the experiments and that the effects we observed were not due to altered activity levels of the cultures.

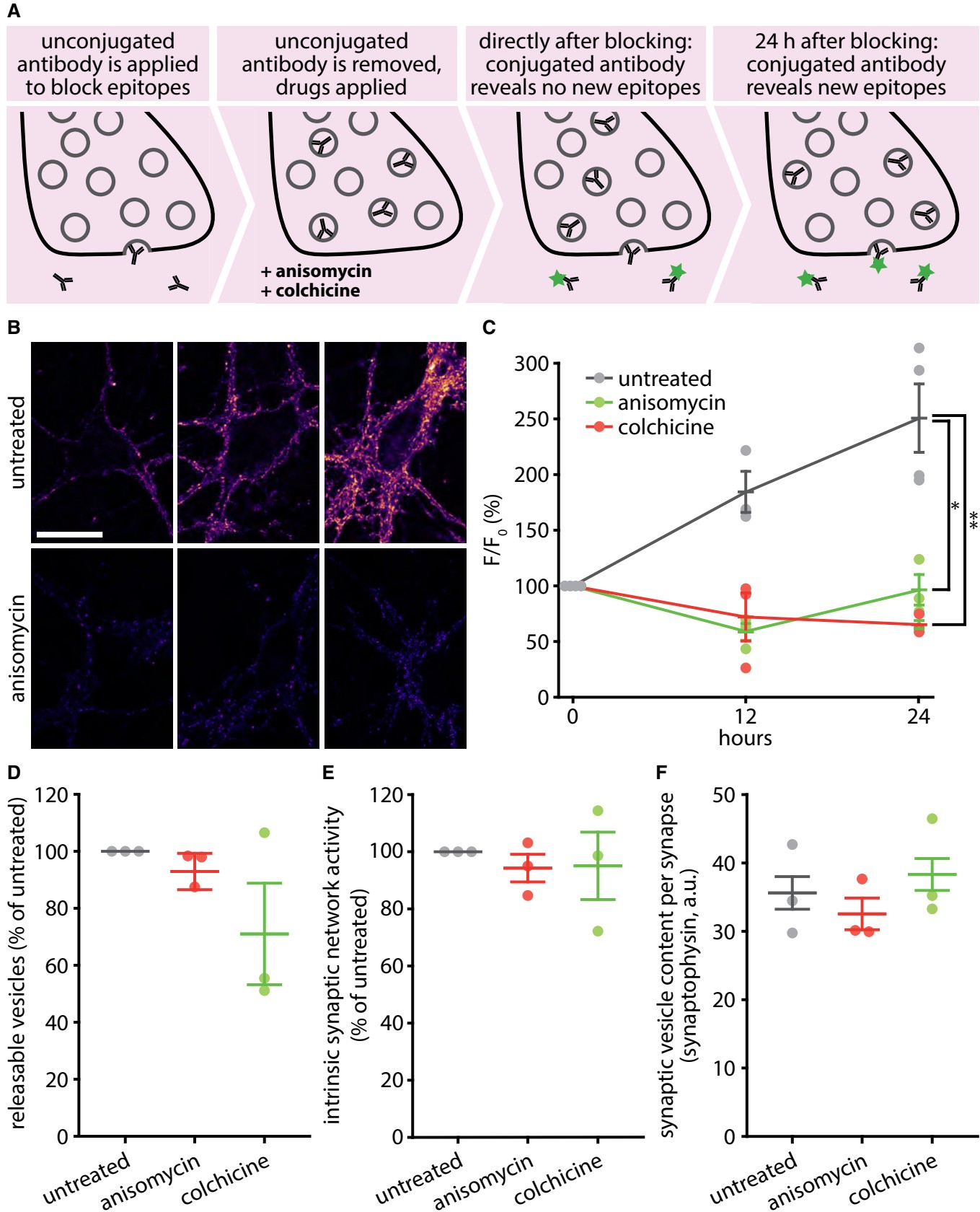

Figure 5.

**Figure 5. The Synaptotagmin 1 molecules that left the recycling population do not return to it spontaneously.**

A   To determine the rate at which new Synaptotagmin 1 epitopes come into the recycling pool, we saturated the lumenal Synaptotagmin 1 epitopes on active synaptic vesicles by incubating with an unconjugated monoclonal antibody for 2 h at 37°C and then followed the appearance of new epitopes by applying a fluorophore-conjugated (Atto647N) version of the same monoclonal antibody, at different time points after saturating the initial epitopes. Individual coverslips were only used to investigate single time points and were fixed before imaging.

B   Exemplary images taken after applying the fluorophore-conjugated antibodies at 0, 12, or 24 h after saturating the initial Synaptotagmin 1 epitopes. Scale bar: 20 μm. Imaging was performed with a Leica SP5 confocal microscope.

C   The same experiment was performed, but the cultures were incubated with drugs that disrupt protein synthesis (anisomycin) or microtubule-based transport (colchicine). The fluorescence was measured at different time points and was expressed as percentage of the fluorescence at the initial time point. $n = 4$ independent experiments per data point for untreated 0 and 24 h, and $n = 3$ for all else, at least 10 neurons sampled per experiment; *$P = 0.0070$; **$P = 0.0022$. Statistical significance was evaluated using one-way ANOVA ($P = 0.014$, $F_{(2, 9)} = 19.26$), followed by the Bonferroni procedure.

D   To determine whether the drugs impair synaptic activity, neurons were treated with the drugs for 24 h and were then stimulated in the presence of fluorescently conjugated Synaptotagmin 1 antibodies (20 Hz, 30 s), to label the entire recycling pool. Only a limited decrease in recycling was observed in drug-treated preparations ($n = 3$ independent experiments per data point, at least 10 neurons sampled per experiment; no statistically significant differences as determined via one-way ANOVA, $P = 0.2205$, $F_{(2, 8)} = 1.97$).

E   To assess the spontaneous network activity of our cultures after 24 h of drug treatment, we measured vesicle recycling during the last hour of the treatment. The network activity did not change significantly ($n = 3$ independent experiments per data point, at least 10 neurons sampled per experiment; no statistically significant differences as determined via one-way ANOVA, $P = 0.8835$, $F_{(2, 8)} = 0.13$).

F   The total synaptic vesicle pool size after 24 h of drug treatment. This value was determined by immunostaining for a major synaptic vesicle marker, Synaptophysin. There were no significant changes in synaptic vesicle pool size ($n = 3$ independent experiments per data point, at least 10 neurons sampled per experiment; no statistically significant differences as determined via one-way ANOVA, $P = 0.5519$, $F_{(2, 8)} = 0.66$).

Data information: All data represent the mean ± SEM. Imaging was performed with a Leica SP5 confocal microscope.
Source data are available online for this figure.

## Increased neuronal activity accelerates the transfer of Synaptotagmin 1 molecules from the actively recycling to the inactive pool

We next investigated whether temporal age is the defining factor for inactivating vesicles, or whether increased neuronal activity could affect the vesicle lifetime. We tested therefore the fraction of the vesicles that were still recycling after incubation for 12 h with the GABA$_A$ receptor antagonist bicuculline, or with a Ca$^{2+}$ concentration raised to 8 mM (Fig 6). Both treatments lead to a chronic increase in synaptic activity (~ 1.5- to ~ 1.8-fold; see Materials and Methods and Appendix Fig S14). To test their effects, we first incubated the neurons for 1 h with unconjugated Synaptotagmin 1 antibodies, to tag the entire recycling pool. We then added the drugs to the cultures, and after 12 h applied Cy5-conjugated secondary antibodies to the living neurons to detect the still recycling molecules, followed by fixation, permeabilization, and application of Cy3-conjugated antibodies, to detect all other molecules (Fig 6A; the same procedure as in Appendix Fig S12). We found that both bicuculline and 8 mM Ca$^{2+}$ treatments resulted in a decrease in the amount of molecules that still participated in recycling, coupled to an increase in the amount of inactivated molecules (Fig 6B and C). This implies that increased neuronal activity speeds up the inactivation of Synaptotagmin 1 molecules.

## A mathematical model suggests that vesicle proteins participate in vesicle recycling a few hundred times before inactivation, on average

The data we gathered so far allowed us to model the vesicle protein life cycle mathematically (Fig 7). As assumptions for this model, we used the result presented above that newly synthesized synaptic vesicle proteins start out in the releasable population. The synaptic vesicle protein assemblies retrieved upon recycling can then become inactivated, a state from which they would enter the degradation pathway (Fig 7A), without returning to the recycling state. The model we constructed from this is based on a series of exponential equations

(see Materials and Methods) and recapitulates the data we presented so far on synaptic vesicle degradation (Fig 7B). The model thus enabled us to predict the probability distributions of the time it takes to inactivate synaptic vesicle proteins (Fig 7C), the time the inactivated vesicles remain in the inactive population within the synapse before degradation (Fig 7D), the total synaptic vesicle protein lifetime (Fig 7E), and the usage of synaptic vesicle proteins during their lifetime (Fig 7F). The average number of release rounds per synaptic vesicle protein lifetime predicted from the model is ~ 200.

This number can also be cross-validated with values obtained by a different approach. In principle, the average number of release rounds per vesicle could be determined by multiplying the number of hours spent by the vesicle proteins in the releasable population (from Appendix Fig S12C) with the average number of release rounds undertaken by vesicles per hour. The latter can be estimated by measuring the frequency of activity bursts that the neurons undergo spontaneously in culture, and by measuring the percentage of the vesicle proteins from the active pool that recycle during each activity burst. The activity burst rate was measured using the calcium indicator construct GCaMP6 (Chen *et al*, 2013) and averaged ~ 0.09 Hz in our cultures (Appendix Fig S15A–C). The percentage of the vesicle proteins released per activity burst was measured by simultaneous imaging of GCaMP6 and the Synaptophysin-based pHluorin sypHy (Granseth *et al*, 2006; Li *et al*, 2011), which was used as an indicator of synaptic vesicle release (Appendix Fig S15D–F). From these parameters, one can also calculate the average number of release events, which averaged to ~ 210 rounds of release (see the legend to Appendix Fig S15 for the calculation formula), very close to the result obtained with the simpler model from Fig 7.

## SNAP25 co-localizes more strongly with 4-day-old antibody-tagged Synaptotagmin 1 molecules than with freshly tagged ones

We next tested whether the tagged Synaptotagmin 1 antibodies of different ages co-localized differently with various synaptic

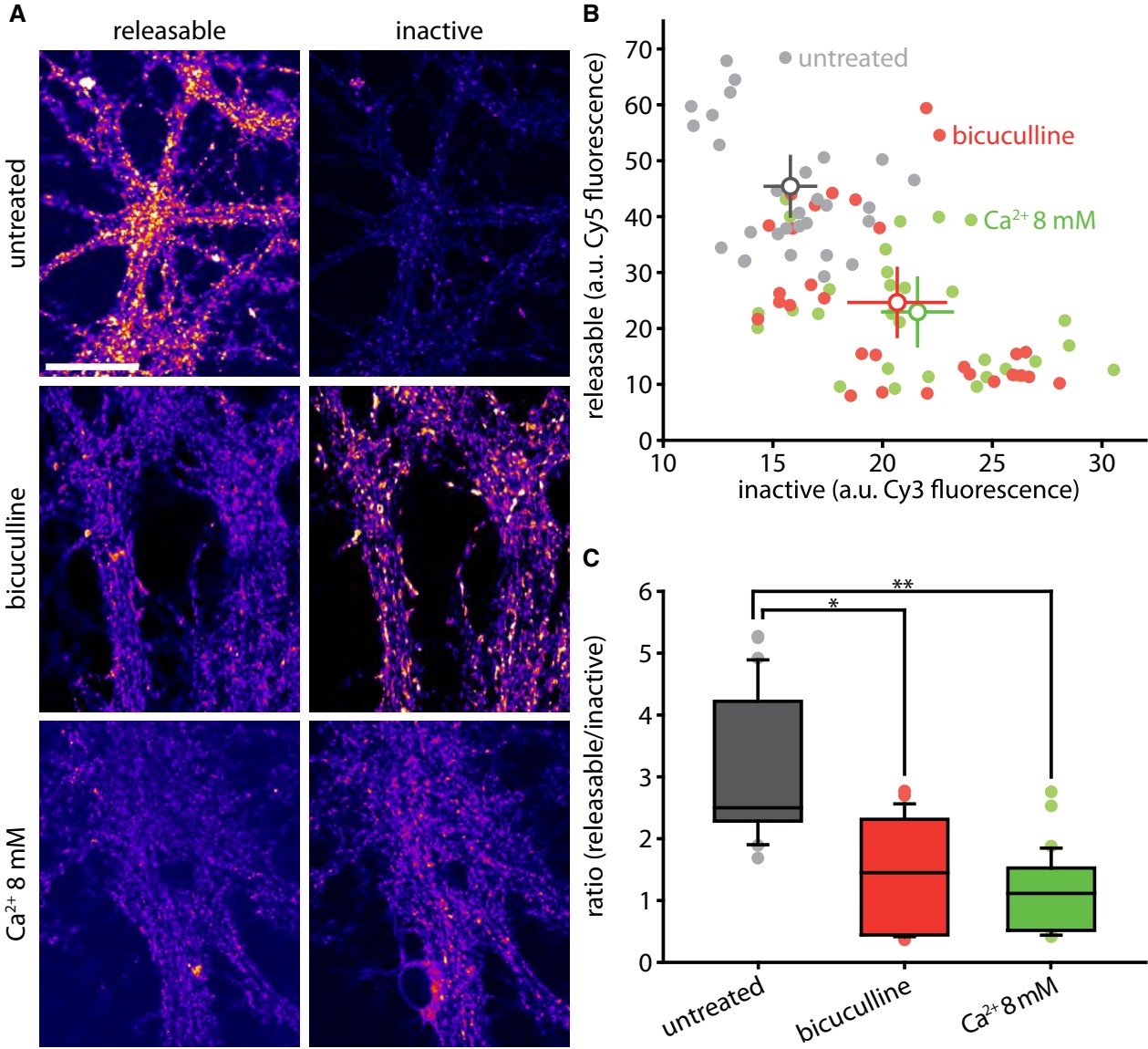

**Figure 6. Increased synaptic activity accelerates the loss of Synaptotagmin 1 molecules from the actively recycling pool, and their entry into the inactive pool.**

A We investigated the effect of increasing neuronal activity on the aging and inactivation of vesicles as follows. We tagged Synaptotagmin 1 epitopes with non-conjugated antibodies, and then, after a 12-h interval, we applied fluorophore-conjugated secondary antibodies (Cy5) to the living cultures for 1 h. These antibodies revealed all of the Synaptotagmin 1 antibodies that were exposed during vesicle recycling. We then fixed and permeabilized the neurons and applied secondary antibodies conjugated to a different fluorophore (Cy3), to thereby reveal all of the remaining, no longer releasable, Synaptotagmin 1 antibodies. This procedure thus indicates the proportion of the tagged Synaptotagmin 1 epitopes that are still recycling in the synapses under normal activity, as well as the proportion that are present in the synapses. Scale bar: 50 μm. Imaging was performed with a Leica SP5 confocal microscope.

B Data from images such as those shown in (A) were quantified, with the fluorescence intensity from releasable (Cy5) and inactive (Cy3) epitopes plotted in (B). The experiment was performed in untreated control cultures, or in cultures incubated for the 12 h with bicuculline or 8 mM $Ca^{2+}$ to increase synaptic activity ($n = 30$ neurons for control, 29 neurons for bicuculline, and 30 neurons for 8 mM $Ca^{2+}$, from three independent experiments per condition). The intensity of the signal ascribed to releasable or inactive vesicles is shown. All data represent the mean ± SEM.

C Ratio of releasable vs. inactive vesicles, in arbitrary units, from (A); *$P < 0.0001$; **$P < 0.0001$. Statistical significance was evaluated using one-way ANOVA ($P = 3.68e-12$, $F(2, 86) = 36.61$), followed by the Bonferroni procedure. All data are represented as box plots with median and upper and lower quartile boundaries, plus 1.5 times inter-quartile range (whiskers) and outliers (dots).

Data information: Imaging was performed with a Leica SP5 confocal microscope.
Source data are available online for this figure.

proteins. We labeled the recycling pool of vesicles by incubating the neurons with Atto647N-conjugated Synaptotagmin 1 antibodies. We then either fixed immediately the neurons, thus maintaining the fixed Synaptotagmin 1 molecules in the recycling pool, or incubated the neurons for a further 4 days, to ensure that the molecules switched to the inactive pool. We then

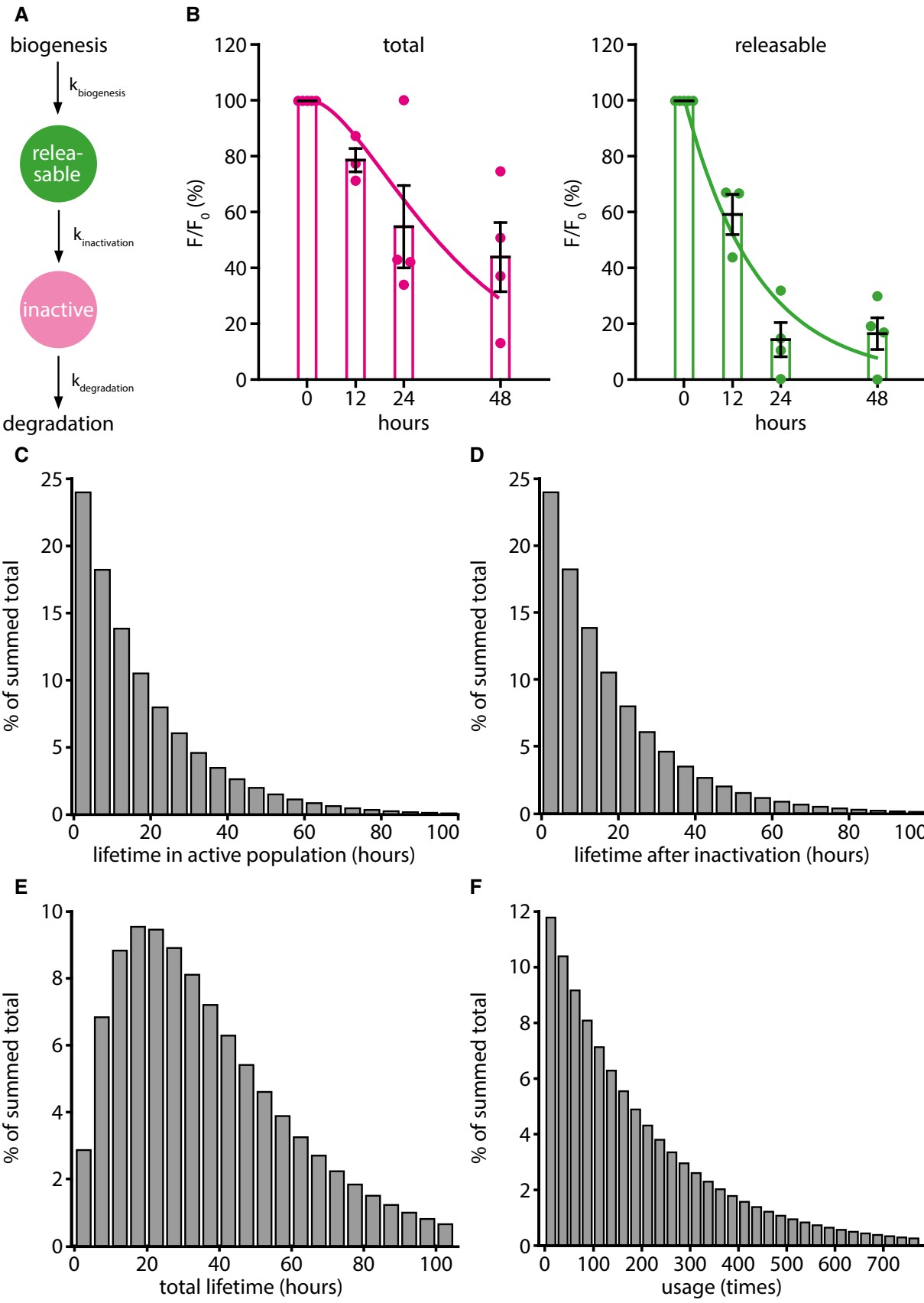

**Figure 7.**

**Figure 7. Quantitative model of the synaptic vesicle protein life cycle.**

A  The assumptions and parameters of the model. For simplicity, we refer on the graph to "synaptic vesicles", although at the stages from exocytosis to endocytosis, while the vesicle proteins are present on the plasma membrane, this term should be replaced with "metastable synaptic vesicle protein assemblies". As this aspect does not change the mathematical interpretation of the model, we decided to use the simpler terminology. The synaptic vesicle starts its life cycle after biogenesis as a releasable vesicle. During its activity, the synaptic vesicle has a certain chance to become inactivated. Afterward, the vesicle can be sent for degradation, as it does not enter the recycling pool again.

B  The predictions from the model (lines) are here overlaid with the actual experimental data (mean ± SEM, *n* = 5, 3, 4, and 4 independent experiments per respective time point, at least 10 neurons sampled per experiment; from Appendix Fig S12C), implying that this very simple model closely recapitulates the observed synaptic physiology.

C  The lifetime distribution of synaptic vesicles in the releasable population, as predicted by the model.

D  The lifetime distribution of synaptic vesicles in the inactive population, after leaving the releasable population, as predicted by the model.

E  The total lifetime distribution of synaptic vesicles, as predicted by the model.

F  The probability distribution of times a synaptic vesicle is used per lifetime; the average is ~ 200 times.

immunostained the cultures for several candidate proteins and imaged them by 2-color STED microscopy, using a Leica TCS SP5 STED microscope (Fig 8). To ensure excellent z-resolution, the cultures were embedded in melamine resin and were cut into 50-nm ultrathin sections before imaging, as we performed in the past for imaging such preparations (Punge *et al*, 2008). Finally, we analyzed the co-localization of individual Synaptotagmin 1 spots with the different markers.

We chose the candidate proteins based on their abundance (Takamori *et al*, 2006; Wilhelm *et al*, 2014) and the availability of excellent antibodies to allow reliable quantification, based on their importance in synaptic vesicle release (Jahn & Fasshauer, 2012; Rizzoli, 2014), and based on their presence in compartments involved in synaptic vesicle release and recycling (Rizzoli, 2014; Wilhelm *et al*, 2014). The following candidate proteins were tested: SNAP25 and Syntaxin 1 for the cell membrane, VGlut 1/2, vATPase, VAMP2 and Synaptotagmin 1 for the synaptic vesicles themselves, Syntaxin 16 and VAMP4 for endosomal compartments, and synapsin as an abundant soluble vesicle-associated protein. We detected only one significant change: The co-localization of the Synaptotagmin 1 spots with SNAP25 was higher for the aging synaptic vesicles (Fig 8A), which had approximately twofold more SNAP25 than young ones (Fig 8C). This was not the case for the other membrane SNARE involved in synaptic vesicle release, Syntaxin 1 (Fig 8B and C). To verify this finding, we independently repeated the experiments and imaged the samples using a different microscopy setup, an Abberior two-color 3D STED (Appendix Fig S16). The results were identical (Appendix Fig S16B and D). A co-localization analysis of selected organelles in the form of line scans further demonstrates the extent of overlap for SNAP25 and Synaptotagmin 1 for 4-day-old tagged molecules (Appendix Fig S17).

**The overexpression of plasma membrane SNAREs coupled to a synaptic vesicle-tagging sequence inhibits exocytosis**

The previous observations lead to the simple hypothesis that old vesicle molecules associate more strongly with SNAP25, which perhaps prevents them from recycling. To test this possibility, we engineered a construct to target SNAP25 to synaptic vesicles: sypHy-SNAP25, a fusion of the Synaptophysin-based pHluorin sypHy (Granseth *et al*, 2006; Li *et al*, 2011) and SNAP25 on the cytoplasmic side. Synaptophysin targets to synaptic vesicles more reliably than most other proteins (Rizzoli *et al*, 2006; Takamori *et al*, 2006) and can therefore efficiently localize SNAP25 to the membrane of synaptic vesicles (Fig 9). We could not detect any morphological differences

between the organization of sypHy and sypHy-SNAP25, suggesting that the latter targets equally well to vesicles (Appendix Fig S18).

At the same time, the luminal pHluorin moiety enabled us to directly observe the response of these tagged vesicles to stimulation (Fig 9A). Exocytosis was severely suppressed by the addition of SNAP25 on the vesicles (Fig 9B, D, and E). To test whether this effect could also be triggered by Syntaxin 1, we generated a sypHy-Syntaxin 1 fusion construct, containing the cytoplasmic moiety of Syntaxin 1, and performed similar experiments. Exocytosis was again suppressed (Fig 9C, D, and E), which suggests that increasing the association of plasma membrane SNAREs with vesicles is hampering the exocytosis process.

At the same time, it was puzzling to see that the effects of SNAP25 and Syntaxin 1 were quantitatively similar (Fig 9D and E). As the two molecules are binding partners, it was conceivable that expressing Syntaxin 1 on synaptic vesicles would bind SNAP25 and recruit it to the vesicles as well, which would then cause a similar phenotype to expressing SNAP25 in the vesicles. To verify this, we immunostained neuronal cultures expressing sypHy-Syntaxin 1 for SNAP25 and for Synaptophysin, as a synaptic vesicle marker (Appendix Fig S19A). The expression of sypHy-Syntaxin 1 caused a significant increase in the levels of SNAP25 within the synapses (Appendix Fig S19B), lending some support to this hypothesis.

**The overexpression of wild-type SNAP25, but not Syntaxin 1, inhibits exocytosis**

We decided to test this point further by expressing wild-type versions of these proteins in neurons, not coupled to Synaptophysin (Fig 10A–C). We then measured the actively recycling pool of Synaptotagmin 1 molecules by incubating neurons with Atto647N-conjugated antibodies for 1 h. These transfections did not affect the bouton morphology or the amount of synaptic vesicles (Appendix Figs S20 and S21), but did affect the actively recycling pool significantly (Fig 10). SNAP25 expression reduced this pool (Fig 10A and B), while Syntaxin 1 expression enhanced it substantially (Fig 10C). Thus, only SNAP25 overexpression reduced the recycling pool, in agreement with previous findings (Owe-Larsson *et al*, 1999).

**The SNAP25 overexpression effects can be removed by expressing CSPα**

SNAP25 is known to interact with CSPα, a chaperone needed in the priming process, to prepare the fusogenic *trans*-complex of SNAP25

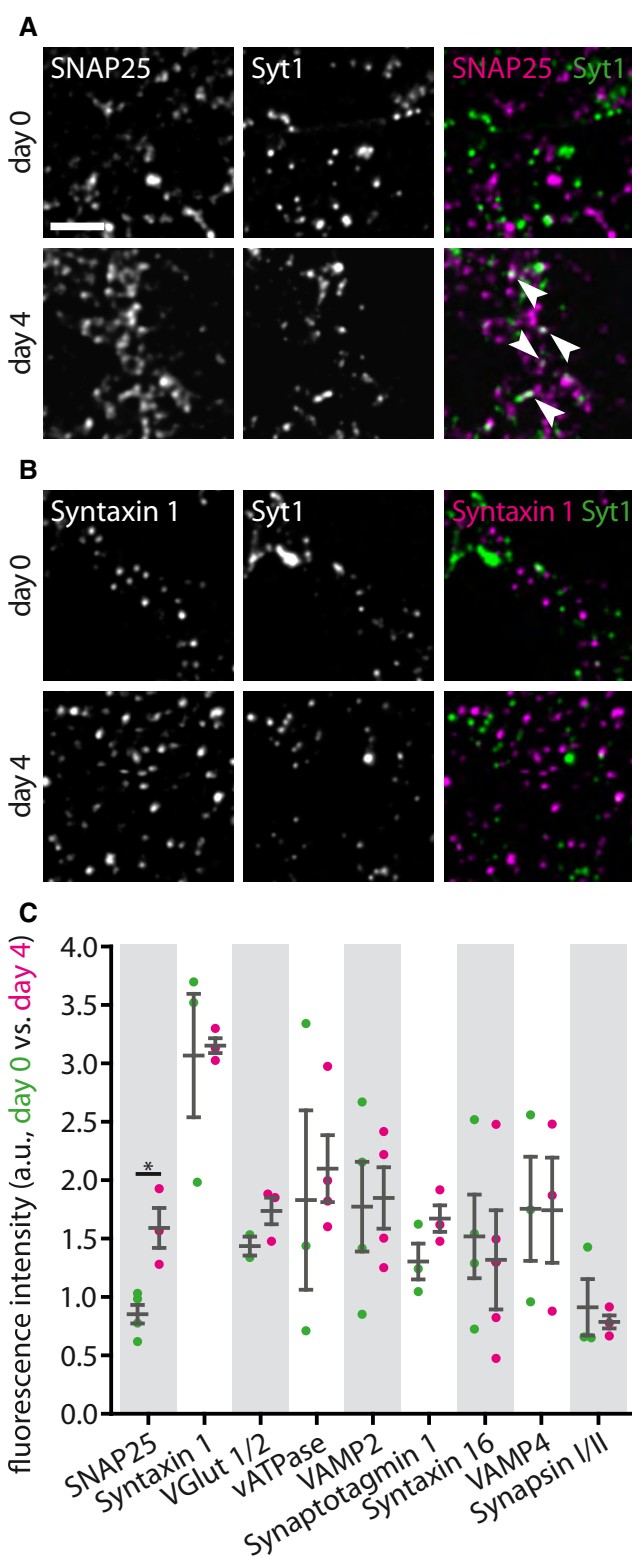

**Figure 8.  SNAP25 co-localizes better with 4-day-old antibody-labeled Synaptotagmin 1 molecules.**

A, B  Two-color STED analysis of changes in synaptic vesicle protein levels during the transition from the releasable state to the inactive state. Living neurons were incubated with Atto647N-conjugated Synaptotagmin 1 lumenal domain antibodies for 1 h at 37°C, to label the actively recycling vesicles, as in Fig 1A, and were then fixed and co-immunostained for different proteins of interest directly, or were placed in a cell culture incubator for 3–4 days, to enable the antibody-labeled molecules to enter the inactive pool, before fixation and co-immunostaining. The samples were embedded in melamine and cut in ultrathin (50 nm) sections, before two-color STED imaging, as in Fig 2B. Exemplary images are shown in (A) for SNAP25 and in (B) for Syntaxin 1, with the protein of interest signal next to the signal from the live tagging of Synaptotagmin 1, and a merged image of both, for day 0 (releasable vesicles) and day 4 (inactive vesicles). Scale bar: 1 μm.

C  We analyzed the amount of fluorescence corresponding to the protein of interest that overlapped with the Synaptotagmin 1 signal (i.e., the two signals were presented within the same voxels, which are substantially below the synaptic vesicle volume in this experiment). The only protein whose levels changed significantly is SNAP25 (SNAP25, *n* (day 0) = 4, *n* (day 4) = 3, *P* = 0.0124, *t*(5) = 3.8200; Syntaxin 1, *n* (day 0) = 3, *n* (day 4) = 3, *P* = 0.8850, *t*(4) = 0.1541; VGlut 1/2, *n* (day 0) = 2, *n* (day 4) = 3, *P* = 0.1986, *t*(3) = 1.6447; vATPase, *n* (day 0) = 3, *n* (day 4) = 4, *P* = 0.7340, *t*(5) = 0.3594; VAMP2, *n* (day 0) = 4, *n* (day 4) = 4, *P* = 0.8837, *t*(6) = 0.1527; Synaptotagmin 1, *n* (day 0) = 3, *n* (day 4) = 3, *P* = 0.1604, *t*(4) = 1.7208; Syntaxin 16, *n* (day 0) = 4, *n* (day 4) = 4, *P* = 0.7406, *t*(6) = 0.3468; VAMP4, *n* (day 0) = 3, *n* (day 4) = 3, *P* = 0.9863, *t*(4) = 0.0183; Synapsin I/II, *n* (day 0) = 3, *n* (day 4) = 3, *P* = 0.6638, *t*(4) = 0.4685; at least 10 neurons sampled per experiment). Statistical significance was evaluated using unpaired *t*-tests. All data represent the mean ± SEM.

Data information: Imaging was performed with a Leica SP5 STED microscope. Source data are available online for this figure.

SNAP25 from the plasma membrane, and two soluble molecules, the ubiquitous chaperone Hsc70 and SGTα (Sharma *et al*, 2012; Fig 10H). This *trans*-complex is involved in priming SNAP25 for fusion. The formation of this complex in *cis*, on the vesicle surface, could create a quantitative bottleneck for the fusion of the aged vesicles in the form of sequestration of CSPα (Fig 10I). According to this hypothesis, over-expressing CSPα would remove this bottleneck and would thus remove the timer mechanism that inactivates aging synaptic vesicles.

To test this hypothesis, we measured the size of the total recy-cling pool, as obtained from incubating neurons with Atto647N-conjugated antibodies for 1 h, in different conditions. This is the most direct indicator of the effects of protein expression on synaptic vesicle recycling, as the proportion of releasable vesicles present in the synapse would shrink or grow when usage-related inactivation is delayed or accelerated. Overexpression of CSPα resulted in a substantial increase in the recycling pool, almost to the level of total pool of molecules in the synapse (Fig 10D). We did not observe any change in activity levels when overexpressing a mutated form of CSPα that does not target to vesicles correctly (Sharma *et al*, 2011; Fig 10E). We then verified whether CSPα overexpression also coun-teracted the effects of the SNAP25 overexpression, as predicted by our hypothesis. This was indeed the case, and only for the wild-type form (Fig 10F), not for the incorrectly targeted CSPα mutant (Fig 10G). The expression of these constructs or construct combina-tions left the overall synapse morphology and vesicle amounts unaf-fected (Appendix Figs S22–25).

As CSPα has been suggested to act in vesicle priming, its potential inhibition by SNAP25 should have a more limited effect

and VAMP2 (Evans *et al*, 2003; Jahn & Fasshauer, 2012). There are only 2–3 copies of CSPα on each vesicle (Takamori *et al*, 2006; Wilhelm *et al*, 2014). It is, in principle, conceivable that SNAP25 might sequester CSPα in non-functional *cis*-complexes on the vesicle surface. Normally, a *trans*-complex forms between vesicular CSPα,

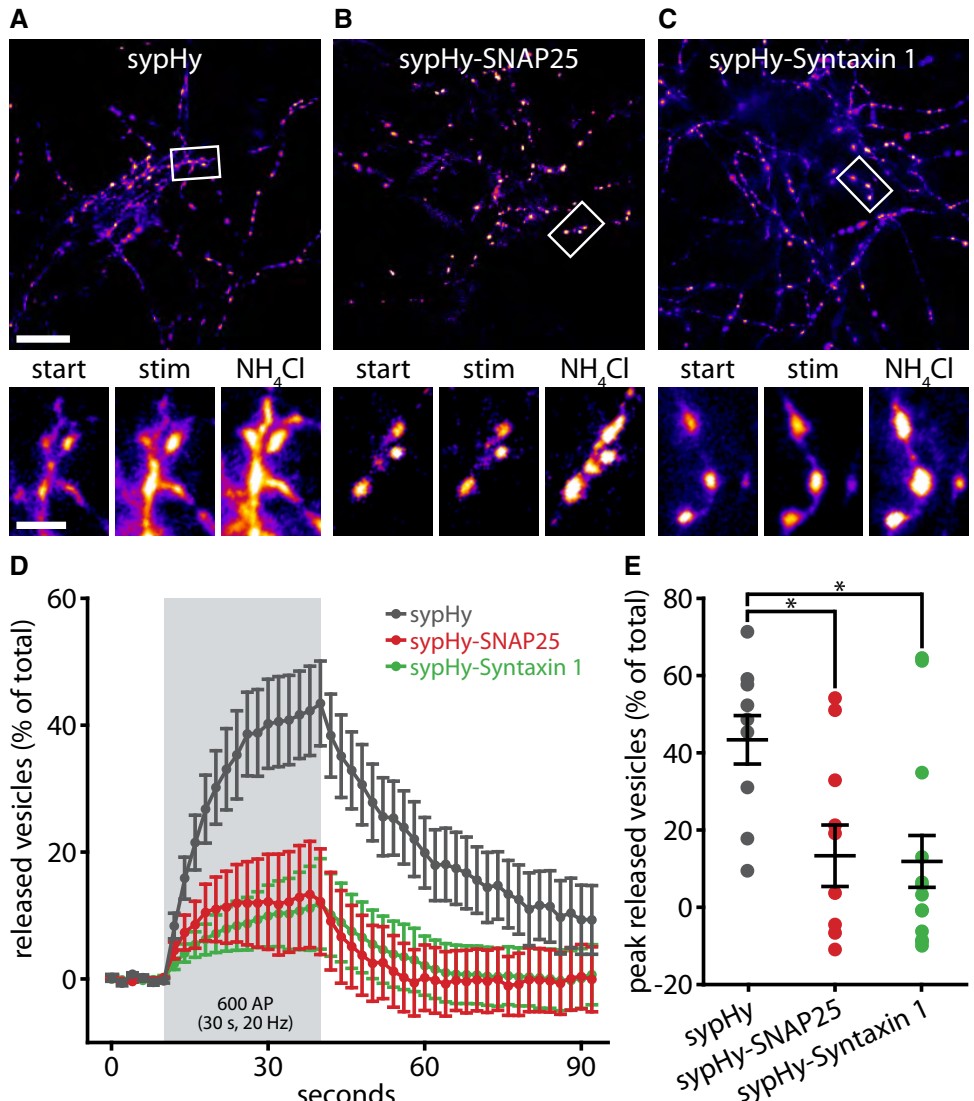

**Figure 9.  The expression of plasma membrane SNAREs coupled to a synaptic vesicle-tagging sequence reduces exocytosis.**

A–C   To test whether SNAP25 is able to inhibit synaptic vesicle recycling, we expressed in neurons either sypHy (A), a sypHy-SNAP25 construct that targets SNAP25 to synaptic vesicles (B), of a sypHy-Syntaxin 1 construct that targets Syntaxin 1 to the vesicles (C). The cells were then stimulated with 600 action potentials at 20 Hz, to trigger the release of the entire recycling pool, at 3–4 days after transfection. Exemplary images are shown for both constructs, with overviews before stimulation, and with magnified areas before stimulation (start), at the peak of stimulation (stim), and after application of $NH_4Cl$. Scale bars: 20 μm (overviews) and 5 μm (magnified areas). $NH_4Cl$ was used to reveal all sypHy moieties, to be able to represent the exocytosis response as % of all sypHy molecules. The cells expressing sypHy-SNAP25 (B) or sypHy-Syntaxin 1 (C) displayed a strongly reduced response.

D   We expressed the time courses showing the release of synaptic vesicles during stimulation as percentage of the maximum fluorescence obtained from a pulse of $NH_4^+$ at the end of the experiment.

E   The graph shows individual data points for the peak release during stimulation for all conditions ($n = 9$ independent experiments for sypHy; $n = 9$ independent experiments for sypHy-SNAP25, *$P = 0.0160$; $n = 12$ independent experiments for sypHy-Syntaxin 1, *$P = 0.0050$). Statistical significance was evaluated using one-way ANOVA ($P = 0.0035$, $F(2, 31) = 6.90$), followed by the Bonferroni procedure.

Data information: All data represent the mean ± SEM. All imaging was performed with a Nikon Ti-E epifluorescence microscope, at 37°C.
Source data are available online for this figure.

on vesicles that are already primed. Thus, vesicles that are already docked should be released efficiently, even when containing overexpressed SNAP25. We verified this hypothesis in experiments expressing sypHy-SNAP25 on the vesicles, as in Fig 9, and found that this was indeed the case (Appendix Fig S26).

**CSPα overexpression appears to be damaging to the neurons**

The cells overexpressing $CSPα_{WT}$ had a significantly higher proportion of damaged neurites than those expressing $CSPα_{mut}$ that does not target to vesicles, as observed by investigating neurite morphology in neurons expressing cytosolic GFP, to be able to detect the

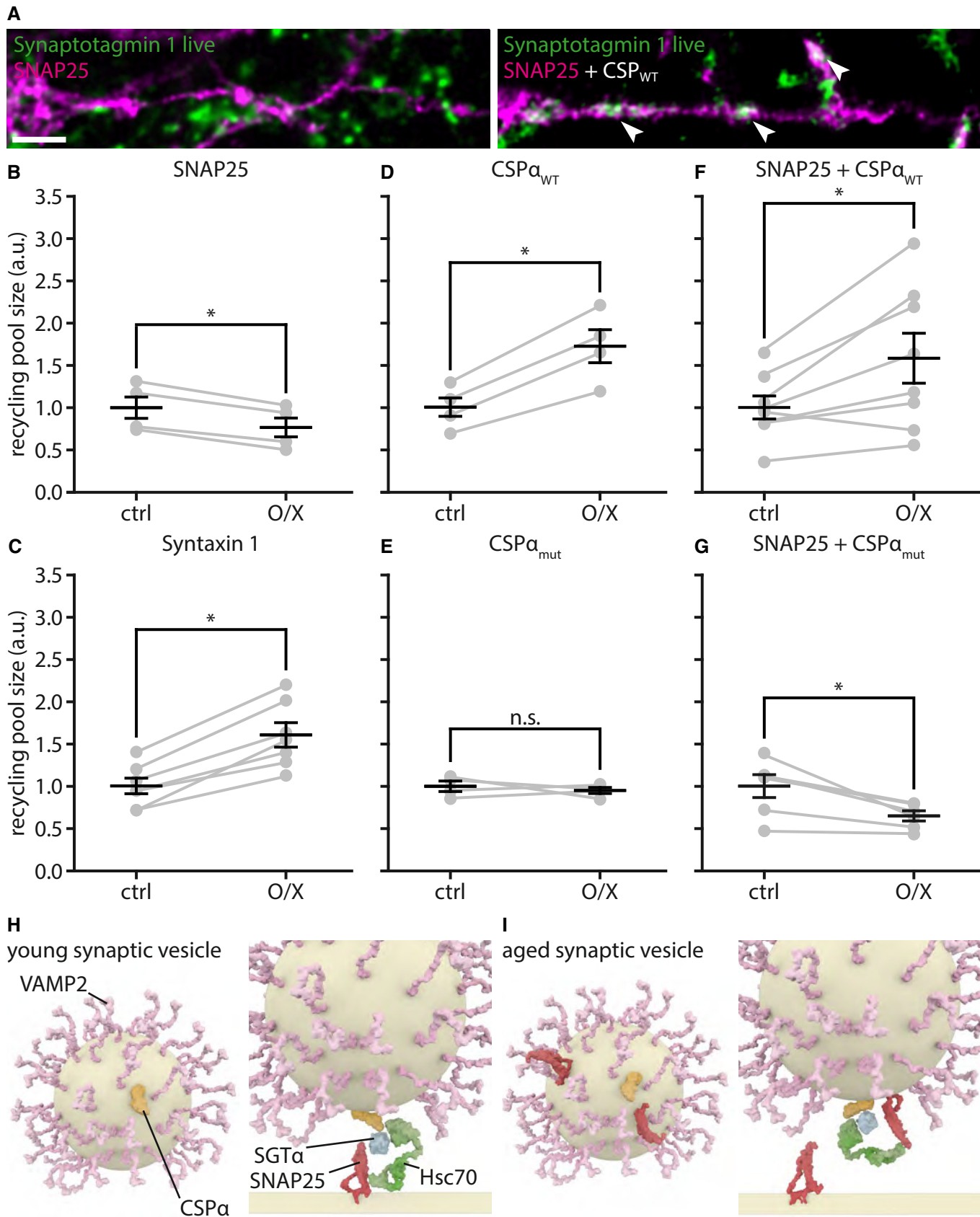

**Figure 10.**

**Figure 10.  SNAP25 overexpression reduces exocytosis, while overexpression of CSPα enhances it.**

A    Neurons were transfected with wild-type SNAP25 (left panel) or co-transfected with SNAP25 and with wild-type CSPα (CSPα$_{WT}$, right panel). The transfected cells are shown in magenta, which corresponds to a YFP moiety that is coupled to SNAP25, for detection purposes. At 3–4 days after transfection, the neurons were incubated with Atto647N-conjugated Synaptotagmin 1 antibodies, for 1 h, to label the actively recycling vesicles, as in Fig 1A. The neurons were then fixed, and the Synaptotagmin 1 antibody intensity from control neurites and from neurites containing the expressed proteins was analyzed. Exemplary images show reduced vesicle recycling (reduced Synaptotagmin 1 antibody levels) in neurons overexpressing SNAP25 (left panel; very limited green intensity within the magenta areas), compared to neurons overexpressing both SNAP25 and CSPα$_{WT}$ (right panel); the arrowheads point to areas of high overlap of Synaptotagmin 1 and SNAP25 signals. Scale bar: 2 µm.

B–G  We quantified the size of the actively recycling pool, determined by incubation with Atto647N-conjugated Synaptotagmin 1 antibodies, as in Fig 1A, in neurons expressing different constructs, or combinations of constructs. We compared the size of the recycling pool in the neurons expressing the constructs to the size of this pool in the non-transfected neurons from the same coverslips. The following conditions were used: (B) SNAP25 ($n$ = 10 transfected neurons from four independent experiments); (C) Syntaxin 1, as a control ($n$ = 13 transfected neurons from seven independent experiments); (D) neurons overexpressing CSPα$_{WT}$ ($n$ = 14 transfected neurons from four independent experiments); (E) CSPα$_{mut}$, a mutated version of CSPα unable to target to vesicles and thus incapable of interacting with SNAP25 (Sharma *et al*, 2012; $n$ = 11 transfected neurons from four independent experiments); (F) SNAP 25 + CSPα$_{WT}$ ($n$ = 20 transfected neurons from eight independent experiments); and (G) SNAP25 + CSPα$_{mut}$ ($n$ = 20 transfected neurons from six independent experiments). The significance levels determined are as follows: (B) *$P$ = 0.0017, $t(3)$ = 10.8862; (C) *$P$ = 0.0003, $t(6)$ = 7.5661; (D) *$P$ = 0.0034, $t(3)$ = 8.5006; (E) $^{n.s}P$ = 0.6004, $t(3)$ = 0.5837; (F) *$P$ = 0.0173, $t(7)$ = 3.0992; and (G) *$P$ = 0.0165, $t(5)$ = 3.5410. Statistical significance was evaluated using paired $t$-tests. All data represent the mean ± SEM.

H    A hypothetical model of SNAP25 and CSPα activity. They interact with SGTα and Hsc70 in *trans*, promoting synaptic vesicle fusion (Sharma *et al*, 2012).

I    A hypothetical model of how SNAP25 on the aged vesicles may interact in *cis* with CSPα, thus sequestering it from its *trans* interaction with SNAP25 molecules from the plasma membrane.

Data information: All imaging was performed with a Leica SP5 confocal microscope (A, B, D–F) or a Nikon Ti-E epifluorescence microscope (C).
Source data are available online for this figure.

neurites optimally (Appendix Fig S27). This was not due to changes in the neuronal levels of activity, as assessed by Ca$^{2+}$ imaging (Appendix Fig S28), but was probably caused by problems with vesicle recycling.

At the same time, it appears that the endocytosis of aged vesicles is poorer than that of young vesicles (Appendix Fig S29). Neurons treated with inhibitors of protein production or transport, as in Fig 5, and which therefore can only recycle aged vesicles, were stimulated strongly (600 action potentials at 20 Hz) and were investigated by a conventional immunostaining procedure for both Synaptotagmin 1 and Synaptophysin. The co-localization of the two molecules, as analyzed by confocal microscopy, was poorer in these neurons than in control neurons, suggesting that endocytosis and the maintenance of vesicle identity are poorer (Appendix Fig S29). These effects mirror those found in mice lacking CSPα (Rozas *et al*, 2012), which suffer from neurodegeneration, and have endocytosis defects that were difficult to reconcile solely with a role of CSPα in SNAP25 priming.

### SNAP25 overexpression promotes the co-localization of Synaptophysin with the recycling endosome marker Rab7

The inactivated synaptic vesicles must ultimately be degraded at a later time point. Synaptic vesicle degradation is widely assumed to entail fusion to the endo-lysosomal system (Katzmann *et al*, 2002; Raiborg & Stenmark, 2009; Rizzoli, 2014). To test whether the inactivated vesicles are indeed targeted for degradation, we tested their co-localization with the recycling endosome marker Rab7, which is thought to be the first step in vesicle degradation (Rizzoli, 2014). We took advantage of experiments in which SNAP25 or sypHy-SNAP25 was overexpressed, and found that the vesicles in both cases co-localized more strongly with Rab7 than control vesicles (Appendix Fig S30).

## Discussion

Organelle aging has long been recognized as an important factor in cellular disease and death. However, little is known about how aged organelles are identified before they can noticeably impair cellular pathways, especially for post-mitotic cells, which are most strongly affected by malfunctioning organelles. Neurons in particular are affected by this problem, as the organism cannot easily eliminate entire cells, which are essentially irreplaceable, without the risk of deteriorating the entire neuronal networks. Furthermore, neuronal signaling depends on the successful action of few individual synaptic vesicles whenever synaptic transmission is supposed to occur. For these reasons, organelle maintenance and preventing the use of potentially damaged organelles is paramount in neurons. There is substantial information on how the synaptic vesicle is degraded (Uytterhoeven *et al*, 2011; Binotti *et al*, 2014; Fernandes *et al*, 2014; Rizzoli, 2014), but it has never been clear why a fraction of the vesicles are inactivated and are reluctant to participate in neurotransmission (Rizzoli & Betz, 2005; Denker & Rizzoli, 2010; Denker *et al*, 2011a; Alabi & Tsien, 2012; Rizzoli, 2014). We suggest here that this is due to the aging of synaptic vesicle proteins, with the reluctant vesicles representing the aged vesicle population.

It is important to note that the age of the vesicles and/or vesicle proteins would be relative in this model: As long as young synaptic vesicles (or vesicle proteins) are available, the old ones do not participate in release. But, when the supply of newly secreted proteins is removed, the old vesicles are still employed in release, and form a recycling pool of approximately the same size as in normal conditions, under spontaneous network activity (Fig 5). This implies that the synapse preferentially uses young vesicles, but old ones can be used, if necessary. They are potentially under the control of specific molecular pathways, as the CDK5/calcineurin pathway noted in the introduction (Kim & Ryan, 2010).

This hypothesis fits most easily with the concept that the identity of the vesicle is maintained for a relatively long time, throughout multiple rounds of recycling. It is conceivable that several synaptic vesicle proteins remain together during recycling, as metastable molecular assemblies, although perhaps not as whole individual vesicles (Rizzoli, 2014). In simple terms, this view implies that the vesicle splits after exocytosis into a number of

protein assemblies, on the plasma membrane. These individual assemblies are stabilized by strong interactions between abundant vesicle proteins such as Synaptophysin and Synaptobrevin/VAMP2 (Becher *et al*, 1999; Mitter *et al*, 2003; Pennuto *et al*, 2003; Adams *et al*, 2015), are strong enough to persist even after detergent solubilization (Bennett *et al*, 1992), and may be further stabilized by an interaction with the endocytosis machinery (Gimber *et al*, 2015). The assemblies have been observed by all laboratories that have studied this issue using super-resolution imaging (see, e.g., Wienisch & Klingauf, 2006; Willig *et al*, 2006; Hoopmann *et al*, 2010; Opazo *et al*, 2010; Hua *et al*, 2011), and are fully compatible with modern interpretations on the metastable nature of membrane protein assemblies (e.g., Saka *et al*, 2014a). The vesicle protein assemblies are then regrouped during endocytosis and vesicle reformation, resulting in new synaptic vesicles. Arguments against this hypothesis would be that, for example, synaptic vesicle proteins have different lifetimes (as discussed below), and thus cannot recycle together continually, or that specific molecules are recycled with different levels of fidelity (see discussion in Rizzoli, 2014).

Assuming that such a scenario would be possible, could neurons nevertheless distinguish between old and young vesicles? This seems unlikely at the level of the single vesicles, but entirely possible at the level of the vesicle pools. The active and inactive synaptic vesicles maintain their pool identities over long time periods (see review in Rizzoli & Betz, 2005), and it has been demonstrated that mild (physiological) stimulation results in no molecular mixing among vesicles from different pools (Wienisch & Klingauf, 2006). Therefore, under physiological activity the recycling vesicles only have a chance to intermix with molecules from other recycling vesicles, remaining separated from the reserve ones. The hypothesis that neurons have mechanisms to recognize old and damage-prone vesicles is thus plausible, irrespective of whether the vesicle fully maintains its identity during recycling or splits into metastable assemblies that intermix within one single pool of vesicles (within the recycling pool).

Our results offer a new interpretation to the long-standing discussion on molecular differences between releasable and inactive "reserve" vesicles (Rizzoli & Betz, 2005). The difference between the two pools seems to be the age of the vesicle molecules. The inactive vesicles, which are reluctant to release and can only be forced to exocytose under strong *supra*-physiological stimulation, probably do not act as a reserve for neurotransmission, but are a collection of aged vesicles that are prevented from releasing. These vesicles may take on different roles in their late age, such as providing a buffer capacity for soluble co-factors of synaptic vesicle exo- and endocytosis (Denker *et al*, 2011b).

### A hypothetical synaptic vesicle life cycle

Based on our data, we suggest the following sequence of events: Synaptic vesicle precursors are produced in the soma and are transported to the synapse, where they are assembled into releasable vesicles. The vesicle proteins are used in exocytosis for up to a few hundred times during their lifecycle (Fig 7F and Appendix Fig S15), and the protein assemblies get inactivated by contamination with SNAP25, which blocks CSPα in dysfunctional *cis*-complexes (Fig 10I), before the vesicles are ultimately degraded. Due to the

low copy numbers of the molecules involved, this approximates a binary switch, which probably works as a stochastic phenomenon that has a statistical effect on the entire population (see Fig 7 for details).

These findings are in agreement with the observation that aged vesicles are not inherently unable to release, but are only less able to do so than young vesicles. When a young vesicle approaches the active zone, we hypothesize that SNAP25 from the plasma membrane interacts with the CSPα from the vesicle surface and is primed and readied for fusion. In contrast, for an older vesicle the CSPα molecules are less likely to prime the plasma membrane SNAP25, since they can alternatively interact with the vesicle-bound SNAP25. This makes such vesicles less able to prime, and probably prevents them from docking, as long as young vesicles are abundant in the vicinity. Especially under conditions of strong stimulation, however, where the young vesicles are all rapidly depleted, the aged vesicles could still be recruited to release, albeit with ever-decreasing efficiency (Fig 2D).

### The inactivation of synaptic vesicles precedes the accumulation of damage to their proteins

One caveat of the work presented above is that while there is some evidence that SNAP25 is able to inactivate the vesicles, it cannot be concluded that this is the only determinant of age on vesicles. It is possible (and even probable) that other elements play a role as well, such as the accumulation of oxidative damage. However, it is currently not possible to directly measure the damage, such as oxidation, suffered by individual molecules on individual vesicles, in a live-cell experiment.

To obtain more insight into this issue, we turned to previous investigations of vesicle protein lifetimes. Currently, there is little evidence available on how fast synaptic proteins might accumulate damage. However, if we assume that degradation of proteins occurs only after they have been damaged, the lifetimes of synaptic proteins in culture (Cohen *et al*, 2013) can be used to approximate the rates of damage to synaptic vesicle proteins (see Appendix Supplementary Methods). We plotted the cumulative prediction of synaptic vesicle protein damage and explored how much of the protein complement of one synaptic vesicle would be damaged at the time of inactivation (Appendix Fig S31). This calculation suggests that virtually no synaptic vesicle proteins are damaged at this time point and that synaptic vesicles are removed just before the accumulation of damage begins.

This also indicates that the molecular timer we identified probably acts as a predictive mechanism, which preemptively removes vesicles from neurotransmission, before they can be damaged and become a hazard to cellular function. As outlined in the Introduction section and in the first paragraph of the Discussion section, this is a crucial requirement to ensure reliability in cell biological systems that rely on the action of only a few individual organelles, such as synaptic vesicle recycling.

It is still unclear whether all membrane proteins of the aged vesicle will be degraded simultaneously in the cell body, or whether some, which are not yet damaged, will escape degradation. Such proteins could be again used in the formation of synaptic vesicles, as has been suggested in the past for dense-core vesicles (Vo *et al*, 2004), but this issue requires further investigation.

## Conclusion

We conclude that the synapse evolved to accurately predict and prevent synaptic vesicle damage. Aged vesicles are thus largely retired from neurotransmitter release, albeit they may perform other functions. Synaptic vesicles are removed from activity just as they are expected to start accumulating damage. This is necessary since the cell depends on a fairly small population of releasable synaptic vesicles, which should not be compromised by even the slightest damage, if they are to ensure continued and reliable neurotransmission.

# Materials and Methods

### Hippocampal cultures and transfections

Neuronal hippocampal cultures were obtained from dissociated hippocampi of newborn rats (Banker & Cowan, 1977; Kaech & Banker, 2006); see Appendix for more details. Transfections were performed with a standard calcium phosphate kit (Promega, Fitchburg, WI, USA), using a procedure slightly modified from the manufacturer's protocol; see Appendix for more details.

### Live-cell immunostaining and live-cell experiments

To tag recycling synaptic vesicles, we used a rabbit polyclonal antibody directed against the lumenal domain of VGAT (# 131 103CpH; Synaptic Systems, Göttingen, Germany; Figs 1C and E, and 2C and E), or a mouse monoclonal antibody, designated clone 604.2 (# 105 311; Synaptic Systems, Göttingen, Germany), directed against the lumenal domain of Synaptotagmin 1 (all other experiments). Neurons were incubated with these antibodies at a dilution of 1:120 (from a 1 mg/ml stock), in their own culture medium. The incubation was performed for 1 h, to achieve the tagging of the entire recycling pool (~ 50% of all epitopes of Synaptotagmin 1 in any given synaptic bouton; Appendix Fig S7), unless otherwise indicated. Depending on the experiment (noted in figure legends and in Source Data tables associated with all figures), the Synaptotagmin 1 antibody was either unconjugated (# 105 311; Synaptic Systems, Göttingen, Germany) or was conjugated to the fluorescent dye Atto647N (# 105 311AT1; Synaptic Systems) or to the fluorescent dye CypHer5E (# 105 311CpH; Synaptic Systems). To distinguish between releasable and inactive vesicles, the releasable population was first tagged by unconjugated 604.2 Synaptotagmin 1 antibodies, and a secondary anti-mouse antibody conjugated to Cy5 and dialyzed into Tyrode's solution (124 mM NaCl, 5 mM KCl, 30 mM glucose, 25 mM HEPES, 2 mM CaCl$_2$, 1 mM MgCl$_2$, pH 7.4) was applied for 1 h onto the living neurons. In the same experiment, a Cy3-conjugated anti-mouse secondary was applied after fixation and permeabilization, to reveal non-recycling vesicles. For blocking surface and recycling pool epitopes, the unconjugated 604.2 antibody was applied for ~ 2 h. Fresh epitopes entering the recycling pool were detected by probing the cultures with the Atto647N-conjugated 604.2 antibody, for 30 min. Drug applications in the same experiment started 1 h into the blocking step and lasted until fixation. The used concentrations were 40 μM anisomycin and 10 μM colchicine (all drugs from Sigma-Aldrich, St. Louis, MO,

USA). As a control for the efficacy of the treatments to increase neuronal activity, we performed a set of experiments to concomitantly evaluate Ca$^{2+}$ and synaptic vesicle dynamics. Neurons treated with bicuculline showed a significantly higher spontaneous Ca$^{2+}$ bursting frequency with respect to the untreated control (~ 55% increase). The fraction of the synaptic vesicle pool released by each spontaneous event was virtually unchanged in comparison with controls. Neurons treated with 8 mM Ca$^{2+}$ lost the synchronous bursting activity, but significantly more vesicles recycled (88% increase, as determined under 0.5 μM bafilomycin incubation, to block the vesicle acidification and to reveal the total fraction of vesicles recycling at one point in time). To obtain an estimate of the synaptic vesicle pool fractions in our culture system (Appendix Fig S7), experiments were performed with the 604.2 antibody conjugated to Atto647N. The antibody was allowed to tag different vesicle fractions. The surface fraction of the releasable population was identified in TTX-treated cultures at 4°C, to suppress both action potential-evoked and spontaneous release. The internalized releasable population was assessed in untreated cultures, after 1 h of incubation with the antibody, at 37°C. The spontaneous, action potential-independent release was assessed in TTX-treated cultures at 37°C. The fraction of the inactive vesicle population was assessed by immunostaining after fixation and permeabilization. To perform TEV protease cleavage of the VAMP2-TEV-SNAP construct, we applied AcTEV protease (Invitrogen, Waltham, MA, USA) at a concentration of 0.1 U/μl directly to the living neurons during imaging in Tyrode's solution at 37°C.

### Fixation, permeabilization, and immunostaining

Fixation was performed with 4% PFA in PBS (137 mM NaCl, 2.7 mM KCl, 10 mM Na$_2$HPO$_4$, 2 mM KH$_2$PO$_4$) at pH 7.5 for 15 min on ice followed by 30 min at room temperature. Coverslips were then washed briefly with PBS 2–3 times, and PFA was quenched with 100 mM NH$_4$Cl for 20–30 min. For immunostainings, neurons were permeabilized in staining solution (PBS + 2.5% BSA + 0.1% Triton X-100), 3 × 5 min. Primary and secondary antibodies were applied in staining solution for 1 h each; washing between primary and secondary antibody incubation was 3 × 5 min, with staining solution. After the secondary antibody incubation, cells were sequentially washed 3 × 5 min with PBS + 2.5% BSA, high-salt PBS (PBS + 350 mM NaCl), and PBS to increase the stringency of the antibody staining. Neurons were embedded in Mowiol (Calbiochem, Billerica, MA, USA) for imaging. The primary antibodies used were as follows: Synaptophysin (guinea pig, # 101 004 or mouse, clone 7.2, # 101 011, both from Synaptic Systems, Göttingen, Germany), SNAP25 (rabbit, # 111 002; Synaptic Systems), Syntaxin 1 (rabbit, # 110 302; Synaptic Systems), VGlut 1/2 (rabbit, # 135 503; Synaptic Systems), vATPase (rabbit, # 109 002; Synaptic Systems), VAMP2 (rabbit, # 104 202; Synaptic Systems), Synaptotagmin 1 (mouse, clone 604.2, # 105 311, or mouse, clone 604.2, directly conjugated to Atto647N, # 105 311AT1, or rabbit, # 105 102, all from Synaptic Systems), Syntaxin 16 (rabbit, # 110 162; Synaptic Systems), VAMP4 (rabbit, # 136 002; Synaptic Systems), Synapsin (rabbit, # 106 002; Synaptic Systems), Rab7 (# 9367; Cell Signaling Technology, Cambridge, UK), and PSD95 (rabbit, #3450S; Cell Signaling). The secondary antibodies used were as follows: donkey anti-guinea pig conjugated to Alexa 488 (# 706-545-148; Dianova, Hamburg, Germany), donkey anti-guinea

pig conjugated to AMCA (# 706-155-148; Dianova), donkey anti-guinea pig conjugated to Cy3 (# 706-165-148; Dianova), goat anti-mouse conjugated to Cy3 (# 115-165-146; Dianova), goat anti-mouse conjugated to Cy5 (# 115-175-146; Dianova) and dialyzed into NaN3-free Tyrode for live stainings, goat anti-rabbit conjugated to Cy5 (# 111-175-144; Dianova), goat anti-rabbit conjugated to Chromeo494 (# 15042; Active Motif, Carlsbad, CA, USA), and goat anti-rabbit conjugated to Atto647N (# 611-156-122; Rockland, Limerick, PA, USA). The antigenic peptide used in the antibody competition assay (Synaptic Systems, # 105-1P) was diluted in staining solution supplemented with 0.05% $NaN_3$ to inhibit bacterial and fungal growth for the duration of the experiment.

### Click chemistry

For click labeling, neurons were incubated for 9 h in a methionine-free DMEM (4.5 mg/ml glucose, lacking pyruvate, methionine, glutamine, and cysteine; Life Technologies, Carlsbad, CA, USA) supplemented with 50 μM L-azidohomoalanine (AHA; Life Technologies), 812 μM $MgCl_2$, 6.5 mM HEPES, 260 μM cysteine, 1:50 B27 (Gibco, Life Technologies), and 1:100 GlutaMAX (Gibco, Life Technologies). Click labeling was performed after fixation with a commercial kit (Click-iT Cell Reaction Buffer Kit; Invitrogen), and with 5 mM Chromeo494-alkyne (Jena Bioscience, Jena, Germany).

### SNAP tag labeling

SNAP tag labeling was performed with the cell-permeable dyes TMR-Star and 647-SiR (both from New England Biolabs, Ipswich, MA, USA). First, TMR-Star was applied after 3–4 days of expression of the VAMP2-TEV-SNAP construct, for 15–30 min at 1 μM in the neurons' own culture medium. After washing out residual dye with Tyrode's solution, the neurons were then placed back. Second, 647-SiR was applied 24 h after the pulse with TMR-Star, for 24 h at 1 μM in the neuron's own culture medium. The extended labeling time for 647-SiR was necessary to achieve a sufficiently high signal-to-noise ratio.

### Metabolic labeling with [15]N leucine

We added 2.4 mM [15]N leucine (Sigma-Aldrich) to the neuronal culture medium (0.8 mM leucine) for 1–3 days prior to antibody tagging of releasable vesicles with the 604.2 Synaptotagmin 1 antibody, or for 3 days following tagging with the 604.2 Synaptotagmin 1 antibody to assess the inactive vesicle population. The cultures were then fixed, processed into 200-nm-thin sections, and imaged in fluorescent microscopy, before imaging in nanoSIMS and correlating the images as described before (Saka *et al*, 2014b). A co-immunostaining for Synaptophysin and PSD95 was used to identify synapses.

### Imaging

A TCS SP5 STED microscope (Leica, Wetzlar, Germany) equipped with a HCX Plan Apochromat 100×, 1.4 NA oil STED objective, and operated with the LAS AF imaging software (version 2.7.3.9723; Leica), was used for performing two-color STED microscopy. Chromeo494 and Atto647N were excited with pulsed diode lasers

(PDL 800-D; PicoQuant, Berlin, Germany) at 531 and 640 nm, respectively. The STED beam was generated by a Ti:sapphire laser (Mai Tai; Spectra-Physics, Mountain View, CA, USA) tuned at 750 nm. The same microscope was used for acquiring confocal images using an HCX Plan Apochromat 63×, 1.4 NA oil immersion objective. Alternatively, an Abberior 2-channel easy3D STED microscope operated with Imspector imaging software (Abberior, Göttingen, Germany) was used for two-color STED microscopy. This setup was built on an Olympus IX83 base, equipped with a UPlanSApo 100× oil immersion objective (Olympus Corporation, Shinjuku, Tokyo, Japan) and an EMCCD iXon Ultra camera (Andor, Belfast, Northern Ireland, UK). Pulsed 561-nm and 640-nm lasers were used for excitation, and easy3D module lasers at 595 and 775 nm were used for depletion. Alternatively, to achieve minute-precision timing for the experiments in Appendix Fig S5, a Cytation 3 cell imaging device equipped with a multi-mode reader and a 20× air objective was used (BioTek, Winooski, VT, USA). Live imaging was performed with an inverted Nikon Ti epifluorescence microscope (Nikon Corporation, Chiyoda, Tokyo, Japan) equipped with a Plan Apochromat 60×, 1.4 NA oil immersion objective, an HBO-100W Lamp, an IXON X3897 Andor (Belfast, Northern Ireland, UK) camera, and an OKOLab (Ottaviano, Italy) cage incubator system (to maintain a constant temperature of 37°C), operated via the NIS-Elements AR software (version 4.20; Nikon). Alternatively, an inverted epifluorescence Olympus microscope (Olympus Corporation, Shinjuku, Tokyo, Japan) was used for live imaging, equipped with a 60×, 1.35 NA oil immersion objective, a 100-W mercury lamp (Olympus), a charge-coupled device camera (1,376 × 1,032 pixels, pixel size 6.45 × 6.45 μm; F-View II; Olympus), and operated via the cell^P software (version 3.4; Olympus). The nanoSIMS imaging was performed on a Cameca nanoSIMS 50L instrument exactly as described before (Saka *et al*, 2014b), but scanning parameters were 512 × 512 pixels with a pixel size of 35 nm and a dwell time of 4,000 μs per pixel. It is noted in all figure legends which microscopy setup was used to acquire the data.

### Statistical analysis

Two-sided Student's *t*-tests or one-way ANOVA tests, with the *post hoc* Bonferroni procedure, were used to calculate statistical significance, as noted in the figure legends and the Appendix Tables. A *P*-value of < 0.05 was considered statistically significant.

### Data accessibility statement

We do not present data in this report that commonly requires storage in a public repository (e.g., microarray data, protein, DNA, or RNA sequence data). Other data can be made available on an individual basis upon direct contact with the corresponding authors of this report.

### Code availability statement

MATLAB code used in this study can be provided on an individual basis by directly contacting the corresponding authors of this report.

**Expanded View** for this article is available online.

## Acknowledgements

We thank Reinhard Jahn for providing a plasmid for YFP-SNAP25. We thank Erwin Neher for help with the development of the mathematical model of the synaptic vesicle life cycle. We thank Martin Meschkat, Andreas Höbartner, Annedore Punge, and Peer Hoopmann for help with the experiments. We thank Burkhard Rammner for providing the illustrations of synaptic vesicle and protein dynamics. We thank Manuel Maidorn, Martin Helm, and Katharina N. Richter for critically reading the manuscript. S.T. was supported by an Excellence Stipend of the Göttingen Graduate School for Neurosciences, Biophysics, and Molecular Biosciences (GGNB). E.F.F. is a recipient of long-term fellowships from the European Molecular Biology Organization (ALTF_797-2012) and from the Human Frontier Science Program (HFSP_LT000830/2013). The work was supported by grants to S.O.R. from the European Research Council (ERC-2013-CoG NeuroMolAnatomy) and from the Deutsche Forschungsgemeinschaft (Cluster of Excellence Nanoscale Microscopy and Molecular Physiology of the Brain, SFB1190/P09, SFB889/A05, and SFB1286/A03, and DFG RI 1967 7/1). The nanoSIMS instrument was funded by the German Federal Ministry of Education and Research (03F0626A).

## Author contributions

SOR, ST, and AD conceived the project. SJ and AVo (Leibniz Institute) performed the nanoSIMS experiments. EFF performed the experiments on culture activity with GCaMP6 and sypHy and several additional controls. AVi (Cells in Motion Cluster) performed the two-color STED experiments on changes in synaptic vesicle protein levels. ST performed all other experiments. HW developed the mathematical model of the synaptic vesicle life cycle. SOR, ST, and EFF analyzed the data. SOR and ST prepared the manuscript.

## Conflict of interest

The authors declare that they have no conflict of interest.

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
