## [Review Process File · The EMBO Journal]

Newly produced synaptic vesicle proteins are preferentially used in synaptic transmission

Sven Truckenbrodt, Abhiyan Viplav, Sebastian Jähne, Angela Vogts, Annette Denker, Hanna Wildhagen, Eugenio F. Fornasiero and Silvio O. Rizzoli

Review timeline:

Submission date:	21 st August 2017
Editorial Decision:	26 th September 2017
Revision received:	9 th February 2018
Editorial Decision:	6 th April 2018
Revision received:	3 rd May 2018
Editorial Decision:	16 th May 2018
Revision received:	18 th May 2018
Accepted:	29 th May 2018

Editor: Karin Dumstrei

Transaction Report:

1st Editorial Decision

26th September 2017

Thanks for submitting your manuscript to The EMBO Journal. Your study has now been seen by two referees and their comments are provided below.

As you can see from the comments, the referees appreciate the questions asked and addressed, and find the approach used original. However, they also raise major issues with the manuscript. They find that important controls are missing, raise issues with the statistical analysis and find that further support for the key conclusion is needed. So while the analysis is interesting and insightful there are also too many open questions for me to formally invite a revision. I therefore see no other choice but to reject the present submission.

However, given the interest in the topic I can offer to look at a new submission should you be able to revise the manuscript along the lines indicated by the referees. It would entail major revisions and would involve adding further data to better substantiate the conclusions, to improve the presentations of the findings and to provide a better consideration of the interpretations offered.

I should add that for new submissions that we consider novelty at time of submission.

For the present submission I am sorry that I can't be more positive but I hope you find the referee comments useful.

REFeree REPORTS

Referee #1:

The study by Truckenbrodt and colleagues addresses the use-dependent rundown of synaptic vesicle

proteins by investigating how many rounds of exocytosis 2 synaptic vesicle proteins last before they are discarded. The authors show that newly synthesized Synaptotagmin-1 and VGAT are preferentially used in exocytosis during intrinsic network activity and high frequency stimulation. The authors correlate new and old proteins to active and inactive vesicle pools and present a model to estimate the rounds of exocytosis that Synaptotagmin-1 undergoes.

The life-time and use-dependent rundown of synaptic vesicles and synaptic vesicle proteins are important issues that have not been addressed systematically before using the approaches used here. The experimental design is original and elegant. The ambition to reach novel and quantitative conclusions (in this case on how often a vesicle is used) are commendable and the proposed working model for aged vesicle identity (a cis SNAP25-aCSP complex on the vesicle) is attractive. On the other hand, the manuscript is very hard to read, information is scattered, structuring is unconventional and the authors use many untested assumptions and too little solid evidence to reach the main conclusions. Too many steps are taken too superficially in this manuscript. This manuscript is not written to be critically examined, let alone to allow replication of experiments by others. Some of the major issues are the definition of different vesicle pools, which is problematic, the lack of reference data, which seems to be a problem for all main conclusions, and the evidence for a tagging role of SNAP25 and CSP α , which is not convincing. Finally, the statistics appear to be often incorrect.

Major points:

1) Manuscript is hard to read, untested assumptions

Essential information on the experiments is scattered in main text, legends, methods and supplemental data etc. It is hard to find out how exactly experiments have been performed. Legends often do not provide the necessary experimental details, but describe the observations as the Results section normally does. In the Results these details are also not mentioned systematically.

Furthermore, the description contains many untested assumptions, crucial controls appear to be missing and many experiments appear to be inconclusive. At the end of this report, a list of such issues is presented for the first data-set (Fig 1, >1 page of text). A similar list could be made for other figures.

The mathematical model in this paper should be evaluated by experts as a separate manuscript.

2) Definition of releasable and inactive synaptic vesicles.

The experiments of this study are based on Synaptotagmin-1 labelling after exocytosis (and maybe VGAT in Fig 1). The assumption that the labelled puncta are synaptic vesicles is problematic. Synaptotagmin-1 is the main Ca²⁺-sensor for dense core vesicles (at least in chromaffin cells, PMID:11562488), triggers the synaptic fusion of endosomes (PMID:28355182) and is probably in other synaptic compartments too, especially after 4 days of labelling. In line with this idea, Fig S1 shows that some Synaptotagmin1 puncta do not co-localize with Synaptophysin 1. And an unknown part of the fluorescence that does overlap with Syp1 is probably not in synaptic vesicles. Hence, an unknown proportion of the initial Syt1 labeling is probably not tagging synaptic vesicles and the gradual loss of labelled Syt1 in the releasing vesicle pool reflects a complex equilibration among different organelles at the synapse and outside. These circumstances appear to make it impossible to correlate fluorescence changes to vesicle cycles/aging.

3) Lack of reference controls

The main observation is a gradual change of Syt1 luminal epitopes during exocytosis as a function of time/activity. The design of these experiment suffers from the fact that it is not possible to differentiate between SV inactivation and a general decrease of SV fusion. Changes in available Syt1 epitopes can also be explained by slowly deteriorating neurons. This seems to hold for data in many figures, for Fig 1 to Fig 7. It would be convincing to show that the amount of available Syt1 epitopes changes RELATIVE TO A CONSTANT AMOUNT OF FUSION at different time points. However, such a reference is lacking. For instance, patch clamp physiology at different intervals after initial labelling in combination with the imaging assay that the authors developed, would provide such a reference and would allow to correct for any systematic differences at different ages of the neuron. Also such a combined analysis of total fusion and fusion of one 'age-class' of vesicles would make a convincing argument.

4) A role for SNAP 25 and CSP α in SV inactivation is not supported by enough evidence.

Fig 6a shows increased co-localization of Syt1 puncta with areas rich in SNAP25 -but not areas rich in Syntaxin1- between day 0-4 post-labelling. It is not clear what these large SNAP25 and Syntaxin1 puncta are. The authors assume these are synaptic vesicles, but present no evidence that this fluorescence comes from synaptic vesicles. These puncta are larger than Syt1 puncta and therefore it seems likely that they do not correspond to SVs, but other cellular compartments. Hence, there seems to be no evidence to suggest that 'aged' vesicles accumulate SNAP25.

The authors use a fusion construct Synaptophysin-pHluorin-SNAP25 (Fig 6b) to argue that SNAP25 on vesicles inhibits them. However, this may be a non-specific effect and again all controls are lacking. For instance, does this construct localize (exclusively) to SVs and to the same extent as Synaptophysin-pHluorin? Representative examples and a comparison of the total response upon NH₄⁺ superfusion are required. Moreover, non-specific effects should be tested by expressing a mutant SNAP25 and/or Synaptophysin-pHluorin-Syntaxin.

The experiments described in Fig 7 are interesting, but do not support the conclusion that aged vesicles become more fusogenic upon CSP α overexpression. Since the experiments are (probably) performed using CypHer5E-tagged Syt1, the only conclusion can be that these Syt1 molecules distribute differently among cellular organelles upon aCSP over-expression. If 'aged' vesicles would indeed start to contribute to fusion upon aCSP over-expression, this would mean that the total secretion response of the neurons increases (normal response plus the aged vesicles). Hence, only in comparison to total SV fusion, claims can be made about the contribution of vesicles that would be unlikely to fuse without aCSP over-expression. Again, other crucial controls are also lacking (is the total vesicle pool affected by aCSP over-expression, is Ca²⁺-influx altered etc.).

5) Statistics

The Student's t-test is the only statistical test used. This test can only be used to compare two groups with normally distributed data. In Fig 3, 4 & 7, more than two groups are tested. In the rest of the experiments, it is not clear whether data are normally distributed. Furthermore, based on scattered information provided, it seems that many statistical tests applied in this manuscript treat the number of fields of view examined as independent observations, which is incorrect. In general, some datasets clearly have too few independent observations (Fig6) or it is unclear (Fig 1, 5).

Issues with the data set of Fig 1

-Images in panel 1b: it is not clear what kind of microscopy is used (probably STED, not mentioned, but the resolution seems conventional microscopy) and which fluorophore is used to tag antibodies (Atto647N ?). Are these images of living neurons? The first two images suggest the authors image the same neuron repetitively and the vertical axis in panel c is labeled '% of initial fluorescence'. Or are these different neurons? If the same neuron, were the neurons incubated chronically on the stage of a microscope or did the neurons go back & forth to/from the incubator? If not, how can % of initial fluorescence be calculated? Moreover: were the neurons still alive after 10 days? Did they have the same number of synapses at day 10 as compared to day 0? And the same number of total vesicles and release probability? These are crucial controls required for the correct interpretation of the data in relation to vesicle age vs release efficacy.

-Panel 1b contains "images of synapses labelled with synaptotagmin 1 antibodies". However, synapses cannot be evaluated at this magnification.

-The stability of the (unknown) fluorophores themselves in the acidic environment of synaptic vesicles is not evaluated. How much of the fluorescence loss after 10 days is due to this?

-The Results mention "fluorophore-conjugated antibodies directed against the luminal domains of the vesicle proteins synaptotagmin 1 and VGAT" were used. This suggests that both are present in all experiments (?) Or in different experiments? Panel B shows Syt1 fluorescence only. If only Syt1, why is vGAT mentioned and where was it used? Were the results the same as for Syt1 antibodies? It is not clear which fluorophore is used

-"The antibodies were taken up during the intrinsic network activity of the cultures, and remained bound to their target proteins for up to ten days" There is no negative control. Non-specific antibodies are also taken up by neurons. How much more in this case? Which fraction of the taken-up antibody is taken up by synaptic vesicles? How much less without intrinsic network activity? It is also unclear how efficient labeling with Syt1 and VGAT antibodies is. Which fraction of all releasing vesicles is labeled?

-"Incubating neurons with the antibodies for one hour tags the entire active recycling pool of vesicles, which accounts for approximately 50% of all vesicles at the synapse". Combining scattered info from Supplemental data, legend and Methods suggests that this is based on Cy3 and Cy5

comparisons (?). Since these are two different fluorophores, with their unique properties, how can they be used to make a quantitative ratio?

- "The vesicles were slowly lost from the synapses (Fig. 1c), and were degraded in lysosomes in the cell body (Supplementary Fig. 2e,f)". Identification of organelles is lacking. It cannot be concluded that these organelles are lysosomes. Moreover, the negative control (uptake of non-specific fluorophore-tagged antibody) is missing. Are the somatic puncta degrading synaptic vesicles or is this non-specific uptake of fluorophores?

- Fig 1c-d presentation is difficult to read with subtle different color intensities

- "the stimulation-induced reduction in CypHer5E fluorescence". Typical trace(s) should be shown to be able to evaluate the kinetics

- "the fraction of the labelled proteins that could be induced to release decreased with age". This statement requires that vesicular pH is constant with age. Is that justified? When neurons are deteriorating, the proton gradient will be reduced due to energy shortage. This may explain the observed effect. It is not clear if these neurons were alive and release-competent after 10 days.

Referee #2:

In this study, the authors address the matter of synaptic vesicle aging, asking how vesicle fusion competence changes with the age of its proteinaceous components and what times their ultimate demise. They show that vesicles containing newly synthesized proteins are more likely to undergo fusion, whereas vesicles containing older proteins become less likely to fuse, ultimately forming a pool of "inactive" vesicles, which are probably targeted for lysosomal degradation. They then show that old vesicles tend to accumulate SNAP-25; that over expression of vesicle-targeted SNAP25 negatively impacts fusion competence, and that this can be alleviated by overexpression of CSP α . Based on quantitative measurements and modeling they conclude that synaptic vesicle protein meta-complexes or vesicles remain in used for about a day, during which they undergo on average about 200 rounds of exocytosis and endocytosis before becoming fusion incompetent.

There is a lot to like about this manuscript, as it illuminates many interesting aspects of the life cycle of synaptic proteins and synaptic vesicles. There are quite a few issues that need to be addressed, however.

The authors provide compelling evidence that vesicles that contain aged proteins, and in particular synaptotagmin, VGAT, VAMP2, are less likely to fuse upon stimulation, whereas vesicles with freshly synthesized proteins are more likely to fuse. This part is quite strong. The second part, concerning 'contamination' with SNAP25 and the roles of CSP α is mixed. On the one hand, the data is solid. Yet there are glaring quantitative questions (apart from the fundamental question of synaptic vesicle identity - which the authors sidestep elegantly) that remain unaddressed. In particular, this concerns the question how SNAP-25 might, even in principle, serve a timer. Their data (Fig. 6) suggests that SNAP-25 content in 'old' vesicles increases by 50%; as Supp. Fig. 8 suggests that vesicles have on average 2 copies, this would imply that 'old' vesicles have, on average, one more copy (at most 3 more as entailed by the discussion on page 8). This is almost a binary switch, and it is difficult to see how it serves as an analog timer with a resolution of hundreds of cycles, unless one considers the possibility that 'timing' occurs as a statistical, population level phenomenon. I did not see any consideration of these matters, unless I missed something. I thus find myself wondering if alternative possibilities might be worth considering. For example, one might expect that vesicles containing 'old' proteins would become less fusion competent simply because their proteins become dysfunctional. The longer they hang around (avoiding rejuvenation through fusion, endocytosis and endosomal processing) the worse this would become, resulting in a self-reinforcing process creating populations of nearly inactive vesicles. The relationship between protein age and fusion competence is a fascinating topic, which has much to gain from this manuscript, but unfortunately remains underexplored.

Specific comments in chronological order

1) Introduction: "It would be tempting to hypothesize that the situation mirrors that from non-neuronal secretory cells, with active vesicles being newly synthesized ones, and inactive vesicles being aged ones. However, there is so far no evidence either for or against this hypothesis."

The ability to mobilize vesicles between pools using pharmacological manipulations of kinase activity would seem to argue against this (Kim & Ryan (2010) Neuron 67:797-809).

2) Supp. Figure 1. Distributions of "Vesicles" and "Lost molecules" are compared. However, the main text refers to the former as metastable protein assemblies within the plasma membrane (also implied by the data and illustration in Fig 4A in Hoopmann et al., 2010 referred to in the legend). This is confusing. Furthermore, if fusion is inhibited or not explicitly evoked, is staining still observed? How can the spots be differentiated from residual endocytosed vesicles (even on ice)? Some clarifications would be most helpful.

3) Supp. Figure 2B. A) How was correlation measured? On a pixel-by-pixel basis? B) "No significant drop in correlation was determined on any time point compared to day 0". Isn't this a bit puzzling? Wouldn't it imply that the 'old' synaptotagmin (labeled at day zero) is mixed with all synaptotagmin (new and old alike) in all synapses and extrasynaptic pools to the same degree at all time points? How does this fit the idea that old proteins are found in distinct vesicle pools?

4) Supp. Figure 2C. A) "Functional pool fractions are represented as percentage of all vesicles (see Materials and Methods for experimental details)". The Method section does not explain this well. What represents the denominator? The fluorescence in fixed cells? If two different probes are used for live imaging and post-fixation imaging, how are their relative affinities and fluorescence values normalized to each other? B) The main text states that this figure shows (among others) that "Incubating neurons with the antibodies for one hour tags the entire active recycling pool of vesicles". How was this conclusion reached? Were different labeling periods tested? C) The definitions in this figure are also puzzling. Why is surface staining called "Active (surface)"? Might this merely be non-specific binding? As this is not used in the main text, I suggest removing excess definitions, including this one.

5) Fig. 1d. This is an interesting experiment, and worthy of illustration through representative images. It is not clear, however, how this figure shows that "...many of the vesicle proteins...could be triggered to release by strong stimulation (Fig. 1d), but did not release under normal network activity"

6) Fig. 2. These are interesting and challenging experiments. Please show A) the two channels separately in panels b & c, and B) please show low magnification images of labeled neurons to provide some feel of data quality obtained with these two unconventional techniques.

7) Page 6 "...to reveal new synaptic vesicles entering the active, releasable population (Fig. 3b,c). These vesicles could come from two sources: newly synthesized vesicles from the cell body, or the inactivated vesicles, whose epitopes are not affected by the initial incubation with unconjugated antibodies". Formally, the antibody recognizes synaptotagmin, not vesicles; hence, any source of unexposed synaptotagmin would lead to the observed effect. Synaptotagmin could indeed come on 'ready-to-go' synaptic vesicles as the text suggest, but could also come on other vesicular precursors (as the authors write in the discussion) that act to rejuvenate recycling vesicles. This rejuvenation could occur, for example, by such precursors fusing with the plasma membrane, which would then serve as a pool of new proteins for endocytosing vesicles (possible explaining the so called "Active (surface)" pool mentioned in point 4 above), or by fusing with other membrane intermediates through which endocytosed vesicles pass. It might be prudent to avoid shoehorning the findings into two simple alternatives.

8) Fig. 4. A) How were "Releasable and inactive vesicles" tagged? With what (what do Cy3 and Cy5 represent)? The text mentions Supplementary Fig. 3 but this figure makes no mention of Cy3, Cy5 or "inactive". Please note that the paper makes use of many approaches to measure similar things, so explicit explanations will go a long way to make the paper easier to follow. B) Could the differences be related to presynaptic depression? This cannot be excluded given higher spontaneous activity rates in the treated preparations.

9) Fig. 5d. There is an assumption that when these neurons are stimulated at 20 Hz, their spiking tracks perfectly. Our own experience suggests that it does not (i.e. failures are common) and that intracellular recordings typically reveal 1 to 10 action potentials in a burst. It might be prudent to consider more conservative numbers in terms of action potentials per burst.

10) Page 7. A) The calculation of the average rounds of release is quite confusing. As I was reading the manuscript, it became apparent that it would be better to treat N as a numerical characteristic of a decay process, not as a simple average. Put differently, it seemed better to consider expressing N as the number of recycling events at which half of the synaptic vesicle molecules (or vesicles) cease to recycle (akin to the use of half-life to describe protein lifetime). This would be a much more meaningful way of describing the functional decay of synaptic vesicle proteins (and vesicles) and easier to relate to the other variables such as τ . Indeed, as I discovered later, this was ultimately done (Supp. Figure 7). This being so, I would suggest to do away with the section in page 7 and the Methods that describes this calculation (=210 rounds). It is confusing and not entirely rigorous. Moreover, it depends on derivations that were not shown. For example, it is stated that "As indicated in Supplementary Fig. 3c, $\tau = 0.4$ days", yet supplementary Fig. 3 does not present a direct measurement of τ . Supp. Figure 7 does so, however (panel b). If the authors feel strongly about the calculation, they might wish to unify the sections that discuss the calculation and the model, so the quantitative treatment is more coherent. In this case, it might be worth moving Supp. Fig. 7 to the main text and moving Fig. 5 to supplementary data.

11) Fig. 6A. Please provide a better explanation (or simply refer to a prior figure) on the manner by which active and inactive vesicles were labeled.

Referee #1:

The study by Truckenbrodt and colleagues addresses the use-dependent rundown of synaptic vesicle proteins by investigating how many rounds of exocytosis 2 synaptic vesicle proteins last before they are discarded. The authors show that newly synthesized Synaptotagmin-1 and VGAT are preferentially used in exocytosis during intrinsic network activity and high frequency stimulation. The authors correlate new and old proteins to active and inactive vesicle pools and present a model to estimate the rounds of exocytosis that Synaptotagmin-1 undergoes.

The life-time and use-dependent rundown of synaptic vesicles and synaptic vesicle proteins are important issues that have not been addressed systematically before using the approaches used here. The experimental design is original and elegant. The ambition to reach novel and quantitative conclusions (in this case on how often a vesicle is used) are commendable and the proposed working model for aged vesicle identity (a cis SNAP25-aCSP complex on the vesicle) is attractive.

We thank the referee for the comments.

On the other hand, the manuscript is very hard to read, information is scattered, structuring is unconventional and the authors use many untested assumptions and too little solid evidence to reach the main conclusions. Too many steps are taken too superficially in this manuscript. This manuscript is not written to be critically examined, let alone to allow replication of experiments by others. Some of the major issues are the definition of different vesicle pools, which is problematic, the lack of reference data, which seems to be a problem for all main conclusions, and the evidence for a tagging role of SNAP25 and CSP α , which is not convincing. Finally, the statistics appear to be often incorrect.

We took the comments seriously, and we addressed all of the points explicitly indicated by the reviewer. At the same time, we also worked out several other points that were not explicitly mentioned by the reviewer, but relate to the problems mentioned by the reviewer.

In brief, we have now included the following in our revised manuscript (please see the point-by-point reply below):

- *the manuscript is very hard to read.* We thank the reviewer for pointing out this deficiency. We realized that many of our arguments were difficult to follow in the way we originally presented them. We re-wrote large sections of the manuscript, and we streamlined our terminology and our arguments.
- *information is scattered.* We now provide detailed supplementary tables for each figure panel, which include all relevant experimental information: number of experiments, antibodies, live-tagging protocols, descriptions of the experimental time courses and of the stimulation paradigms, information on fixation and further processing, and on the imaging setups used. Furthermore, we now note which imaging setup were used, as well as the statistical tests we employed, in the legend of each individual experimental panel. We have also moved some sections from the Materials & Methods to the appropriate figure legends, especially in the supplement, so that the information is less scattered.
- *structuring is unconventional.* During the streamlining of our manuscript we also strove to provide a clearer 'red thread' throughout, in an effort to clarify the reasoning behind our experiments. We hope that the manuscript is easier to follow now.
- *the authors use many untested assumptions and too little solid evidence to reach the main conclusions.* We performed a large set of additional controls, as explained in detail below, to gather further experimental support for our main conclusions. All our new findings support the validity of our approach, and provide further insights into the mechanisms we proposed. At the same time, we recognize that the long chain of arguments around SNAP25 and CSP is still difficult to prove quantitatively, albeit all of our experiments, including the many new

figures we have added, point in the same direction. Due to this, we now present the SNAP25 and CSP effects much more cautiously, and we also present possible caveats more clearly. At the same time, the mechanism we propose explains not only our findings, but also several puzzling observations on SNAP25 and CSP α over-expression and/or knock-out.

We therefore hope that the reviewer agrees that our findings contribute new valuable information to the field, and may stimulate further investigations, which renders them suitable for publication.

Major points:

1) Manuscript is hard to read, untested assumptions. Essential information on the experiments is scattered in main text, legends, methods and supplemental data etc. It is hard to find out how exactly experiments have been performed. Legends often do not provide the necessary experimental details, but describe the observations as the Results section normally does.

We have thoroughly re-worked the manuscript, and we have re-written completely the figure legends and the Results. We have also included extensive supplementary tables containing the experimental details for each and every figure, so that these can now be easily retrieved.

In the Results these details are also not mentioned systematically. Furthermore, the description contains many untested assumptions, crucial controls appear to be missing and many experiments appear to be inconclusive.

We had performed a long series of additional controls at the time of the submission of the manuscript, but decided not to include them, so as not to clutter the manuscript unnecessarily. We now recognize that this gave an impression of poor science being performed, so we now added all of this additional material. We have also performed many new controls, as suggested by the reviewer. This has resulted in:

- 23 new figure panels derived from data present at the time of the manuscript submission, but which were not shown. These panels were added to the previously existing figures, which caused their re-organization.
- 96 new figure panels derived from new results. These panels contribute to 20 new figures, which are mostly presented as supplementary figures.

Overall, the manuscript now contains 10 main figures and 31 supplementary figures.

At the end of this report, a list of such issues is presented for the first data-set (Fig 1, >1 page of text). A similar list could be made for other figures.

Please see below our replies to the respective list. As indicated above, we have tried to address such issues also for the other figures.

The mathematical model in this paper should be evaluated by experts as a separate manuscript.

We hope that we can maintain the model as part of this manuscript, as it is essential to some of our main conclusions, and provides new quantitative data on the synaptic vesicle life cycle (for example on the usage numbers of synaptic vesicle proteins). The mathematical model is relatively simple, and has very few open variables. We have now streamlined the way in which we present it, and we moved it to the main figures (Fig. 7), as suggested by Reviewer 2.

2) Definition of releasable and inactive synaptic vesicles. The experiments of this study are based on Synaptotagmin-1 labelling after exocytosis (and maybe VGAT in Fig 1). The assumption that the labelled puncta are synaptic vesicles is problematic. Synaptotagmin-1 is the main Ca²⁺-sensor for dense core vesicles (at least in chromaffin cells, PMID:11562488), triggers the synaptic fusion of

endosomes (PMID:28355182) and is probably in other synaptic compartments too, especially after 4 days of labelling.

The labeling is only performed for 1 hour, during which few dense-core vesicles and synaptic endosomes are expected to fuse. However, to address the reviewer's comment directly, we have performed two series of experiments:

- First, we have immunostained the neuronal cultures after Synaptotagmin 1 labeling for markers of synaptic vesicles (Synaptophysin), of dense-core vesicles (Chromogranin A), and of two types of endosomes (Rab 5 for early endosomes, Rab 7 for recycling/late endosomes). This encompasses all relevant membrane trafficking pathways that are expected to involve Synaptotagmin 1 in synapses. We have then analyzed the co-localization of the Synaptotagmin 1 signals with those of these markers by 2-color 3D STED microscopy. The results indicate that the tagged proteins fully co-localized with Synaptophysin (*i.e.* synaptic vesicles), and that there was no co-localization with any of the other markers (*i.e.* other trafficking organelles present at the synapse). Please see the results in the new Supplementary Fig. 6.
- Second, we performed the Synaptotagmin 1 tagging using CypHer5E-conjugated antibodies, and verified whether the labeled organelles participated in exocytosis after electrical stimulation. More than 95-99% of the tagged organelles were able to exocytose after stimulation, under an electrical stimulation paradigm (1200 AP delivered at 20 Hz) that is not expected to trigger the exocytosis of significant numbers of dense-core vesicles or endosomes. These typically require stronger stimulation, such as superfusion with high K⁺-containing solutions (see for example Xia et al., J Cell Sci, 122: 75–82, 2009) or conditions such as chemical LTP. Please see the results in the new Supplementary Fig. 4.

In line with this idea, Fig S1 shows that some Synaptotagmin1 puncta do not co-localize with Synaptophysin 1.

The amount of Synaptotagmin 1 molecules not co-localizing with Synaptophysin was only ~3-4%, as noted in Supplementary Fig. 1. This is a negligible value, in view of the magnitude of the changes we noted in our other experiments.

And an unknown part of the fluorescence that does overlap with Syp1 is probably not in synaptic vesicles. Hence, an unknown proportion of the initial Syt1 labeling is probably not tagging synaptic vesicles and the gradual loss of labelled Syt1 in the releasing vesicle pool reflects a complex equilibration among different organelles at the synapse and outside. These circumstances appear to make it impossible to correlate fluorescence changes to vesicle cycles/aging.

Please see the replies from the previous two paragraphs, and the results from the new Supplementary Fig. 6, in which we demonstrate that virtually all epitopes tagged with the Synaptotagmin 1 (Syt1) antibody are released by electrical stimulation (and therefore belong to synaptic vesicles), and Supplementary Fig. 8, in which we demonstrate that virtually all Synaptotagmin 1 labeling co-localizes with the synaptic vesicle marker Synaptophysin (Syp1).

3) Lack of reference controls. The main observation is a gradual change of Syt1 luminal epitopes during exocytosis as a function of time/activity. The design of these experiment suffers from the fact that it is not possible to differentiate between SV inactivation and a general decrease of SV fusion. Changes in available Syt1 epitopes can also be explained by slowly deteriorating neurons. This seems to hold for data in many figures, for Fig 1 to Fig 7. It would be convincing to show that the amount of available Syt1 epitopes changes RELATIVE TO A CONSTANT AMOUNT OF FUSION at different time points. However, such a reference is lacking. For instance, patch clamp physiology at different intervals after initial labelling in combination with the imaging assay that the authors

developed, would provide such a reference and would allow to correct for any systematic differences at different ages of the neuron. Also such a combined analysis of total fusion and fusion of one 'age-class' of vesicles would make a convincing argument.

We have performed the required control experiments. We now demonstrate that the levels of release are constant throughout the lifetime of the cultures. Please see the results in the new Supplementary Fig. 10.

4) A role for SNAP 25 and CSP α in SV inactivation is not supported by enough evidence. Fig 6a shows increased co-localization of Syt1 puncta with areas rich in SNAP25 -but not areas rich in Syntaxin1- between day 0-4 post-labelling. It is not clear what these large SNAP25 and Syntaxin1 puncta are. The authors assume these are synaptic vesicles, but present no evidence that this fluorescence comes from synaptic vesicles. These puncta are larger than Syt1 puncta and therefore it seems likely that they do not correspond to SVs, but other cellular compartments. Hence, there seems to be no evidence to suggest that 'aging' vesicles accumulate SNAP25.

We apologize for the quality of the images shown in the first version of the manuscript. We now present larger frames, in which it is clear that the fluorescence is coming from individual vesicles. Please see the results in the modified Fig. 8. We also show the overlap of the vesicle and protein of interest signals for several examples in Supplementary Fig. 17, which indicate that such signals often come from spots that colocalize very strongly, and therefore are most likely proteins of interest found within the vesicles.

At the same time, please take into account the fact that the resolution of the two channels in the Leica TCS SP5 STED microscope used for this figure was not identical. The vesicle (deep red) channel had a higher resolution, resulting in smaller apparent spots, while the protein of interest (green, long Stokes shift dyes) channel had a lower resolution, resulting in larger spots. We have corrected this, at least in part, by deconvolving the respective images (see the modified Fig. 8).

Finally, we have also addressed this issue by repeating the experiments with a different, more modern 3D STED microscope, with identical results (see Supplementary Fig. 16).

The authors use a fusion construct Synaptophysin-pHluorin-SNAP25 (Fig 6b) to argue that SNAP25 on vesicles inhibits them. However, this may be a non-specific effect and again all controls are lacking. For instance, does this construct localize (exclusively) to SVs and to the same extent as Synaptophysin-pHluorin? Representative examples and a comparison of the total response upon NH₄⁺ superfusion are required.

We have performed the NH₄⁺ superfusion in our experiments, and used this to determine the percentage of released vesicles as fraction of the total amount of vesicles in the synapse. This was plotted in this fashion in the figures, but unfortunately this was not clear from our description of the experiments. We have now included all of these details in the modified Fig. 9, and in the new Supplementary Fig. 26.

Moreover, non-specific effects should be tested by expressing a mutant SNAP25 and/or Synaptophysin-pHluorin-Syntaxin.

We have worked on this issue thoroughly, and we have included the results in the new Fig. 9.

In brief, we have performed the following:

- We generated two pHluorin constructs containing the two individual SNARE domains of SNAP25, instead of full-length SNAP25, for this experiment. Unfortunately, they do not seem

to fold correctly, and do not allow the expression of the protein. This, therefore, could not be tested directly.

- Since the mutations that would block the SNAP25 interaction to CSP are unknown, we decided not to employ further SNAP25 constructs.
- Therefore, according to the suggestion of the reviewer, we have generated a Synaptophysin-pHluorin-Syntaxin 1 construct, which we have tested in the same experiments. The effects were similar to those of SNAP25 (see the new Fig. 9), and thus a simple conclusion would be that adding plasma membrane SNAREs to the synaptic vesicles is detrimental to exocytosis.
- However, it cannot be excluded that the Syntaxin 1 effect was due to an additional effect of this construct. As Syntaxin 1 is a known binding partner of SNAP25, able to bind it directly (as known for several decades, Chapman et al., J Biol Chem, 269:27427-27432, 1994), we have hypothesized that the expression of Syntaxin 1 on the vesicles may recruit native SNAP25 into the vesicles. This was indeed the case – see the new Supplementary Fig. 19. Thus, it is possible that the effect of Syntaxin 1 is in fact caused by SNAP25, targeted to the vesicle through interaction with Syntaxin 1. As Syntaxin 1 seems to be present on synaptic vesicles in fairly high amounts from biogenesis onwards (6-7 copies per vesicle, according to Takamori et al., 2006, as opposed to only 1-2 copies of SNAP25), this might even be part of the mechanism that recruits additional SNAP25 molecules to synaptic vesicles during repeated rounds of release and recycling.
- To solve this issue, and to differentiate between effects of Syntaxin1 or SNAP25 in our experiments, we turned to two additional experiments.
- First, experiments on expressing Syntaxin1 or SNAP25, in the wild-type form. Only SNAP25, but not Syntaxin 1, was able to block synaptic release in this configuration (see the modified Fig. 10).
- Second, we have repeated the experiments testing the incorporation of SNAP25 or Syntaxin1 in ageing vesicles using a more modern 3D STED setup, with higher resolution (see the new Supplementary Fig. 16). This allowed us to maintain our initial conclusion, namely that only SNAP25, and not Syntaxin 1, enriches in the ageing vesicles.

We therefore conclude that SNAP25 is much more likely to interfere with exocytosis than Syntaxin1.

At the same time, we included an additional series of pHluorin experiments to verify our hypothesis. According to our hypothesis, the SNAP25 effect should reduce the priming of older vesicles, but should not affect the fusion of vesicles that are already docked and primed. Therefore, we expected that the vesicles containing Synaptophysin-pHluorin-SNAP25 should be initially able to fuse, during the first few stimuli of a stimulation train, but should be less able to fuse afterwards, when their priming difficulties would become evident. This was indeed the case: the release during a short electrical stimulation train, during which only the readily releasable vesicles are expected to exocytose (Schikorski and Stevens, Nat Neurosci, 4:391-395, 2001), was much closer to normal release than during longer trains. Moreover, the release upon the first action potentials was almost indistinguishable from that of the normal vesicles. Please see the new Supplementary Fig. 26.

The experiments described in Fig 7 are interesting, but do not support the conclusion that aged vesicles become more fusogenic upon CSP α overexpression. Since the experiments are (probably) performed using CypHer5E-tagged Syt1, the only conclusion can be that these Syt1 molecules distribute differently among cellular organelles upon aCSP over-expression. If 'aged' vesicles would indeed start to contribute to fusion upon aCSP over-expression, this would mean that the total secretion response of the neurons increases (normal response plus the aged vesicles). Hence, only in comparison to total SV fusion, claims can be made about the contribution of vesicles that would be unlikely to fuse without aCSP over-expression.

We apologize for the confusion that our initial description of this experiment caused. We did not use CypHer5E-tagged Syt1, but Atto647N-tagged Syt1 antibodies, which provide a measurement for the

total secretion response. Thus, the figure shows exactly what the reviewer pointed out as the correct experiment.

Please also note that all measurements are normalized to the size of the total vesicle pool in every synapse, measured by immunostaining for Synaptophysin, to avoid any bias caused by changes in the total number of vesicles, rather than changes in their ability to recycle. We have also tested whether the total number of vesicles changed, and found that it did not (please see the reply to the next comment).

Again, other crucial controls are also lacking (is the total vesicle pool affected by aCSP over-expression, is Ca²⁺-influx altered etc.).

We have tested the Ca²⁺ dynamics after CSP α over-expression, and found no changes. Please see the new Supplementary Fig. 28. We have also added 6 Supplementary Figures (Supplementary Figs. 20-25) documenting the changes in the synapse size and morphology, or in the total vesicle pool, in all of the over-expression experiments used (SNAP25 alone, Syntaxin 1 alone, CSP α wild type alone, CSP α mutant alone, CSP α wild type + SNAP25, CSP α mutant + SNAP25). We have not observed any significant changes.

5) Statistics. The Student's t-test is the only statistical test used. This test can only be used to compare two groups with normally distributed data. In Fig 3, 4 & 7, more than two groups are tested. In the rest of the experiments, it is not clear whether data are normally distributed. Furthermore, based on scattered information provided, it seems that many statistical tests applied in this manuscript treat the number of fields of view examined as independent observations, which is incorrect. In general, some datasets clearly have too few independent observations (Fig6) or it is unclear (Fig 1, 5).

We have checked carefully our statistics, and we now clarify the N numbers used. For quick reference, these are presented in the first lines of the supplementary tables containing the experimental details. The reviewer was indeed right about the erroneous use of t-tests for a few figures. All of these tests have been replaced with ANOVA tests, followed by appropriate multiple comparison tests (for example in the new figures 5, 6, 7, S3, S8, S10, S11, S18, or S26).

We do not treat the number of fields of view as independent observations. We typically use independent experiments for all statistics. We occasionally use different cells as different observations (as in the pHluorin experiments, as is customary in the field). We analyzed individual vesicles one by one in experiments such as nanoSIMS, in which the vesicles themselves are thus treated as individual observations; this is customary in, for example, electron microscopy, with which nanoSIMS, as a non-optical tool, has strong affinities.

Regarding the number of experimental observations, we feel that they are in line with the customary practice in the field. We now also explain the amount of data analyzed in the cases where we apparently had too few independent observations. For example, in the original Fig. 6 we indeed presented data from typically 3-4 independent experiments. However, each experiment analyzed from 4500 to 11000 vesicles, with totals ranging from ~16,000 to 38,000 vesicles for the whole set of experiments (we note the precise numbers in the new supplementary tables containing the experimental details, Table 7). We have also repeated this experiment (the new Supplementary Fig. 16), again with >10,000 vesicles analyzed in total, and we therefore now feel that this is in line with the requirements in the field.

Issues with the data set of Fig 1

-Images in panel 1b: it is not clear what kind of microscopy is used (probably STED, not mentioned, but the resolution seems conventional microscopy) and which fluorophore is used to tag antibodies (Atto647N ?). Are these images of living neurons? The first two images suggest the authors image the same neuron repetitively and the vertical axis in panel c is labeled '% of initial fluorescence'. Or are these different neurons? If the same neuron, were the neurons incubated chronically on the stage of a microscope or did the neurons go back & forth to/from the incubator? If not, how can % of initial fluorescence be calculated?

We now clarify all of these details in the main text. In brief, Atto647N was indeed used in neurons that were tagged under live conditions, but were imaged after fixation. Individual coverslips were imaged only once per time point, and thus all images are of different neurons. The percentage of initial fluorescence is calculated as fraction of the average fluorescence of the coverslips imaged immediately after tagging (without further incubation). Please see the respective figure legend. All details can also be accessed in the new supplementary tables summarizing the experimental aspects of each figure.

Moreover: were the neurons still alive after 10 days? Did they have the same number of synapses at day 10 as compared to day 0? And the same number of total vesicles and release probability? These are crucial controls required for the correct interpretation of the data in relation to vesicle age vs release efficacy.

The neurons were alive, and had the same levels of synaptic release and recycling (see the new Supplementary Fig. 10). Moreover, they had the same number of synapses, with the same size and geometry, and with the same amounts of vesicles (see the new Supplementary Fig. 11).

-Panel 1b contains "images of synapses labelled with synaptotagmin 1 antibodies". However, synapses cannot be evaluated at this magnification.

The text should have read "neurons labeled with Synaptotagmin 1 antibodies". To enable the reader to better visualize the synapses, we have also changed the image magnification.

-The stability of the (unknown) fluorophores themselves in the acidic environment of synaptic vesicles is not evaluated. How much of the fluorescence loss after 10 days is due to this?

This was evaluated, but was unfortunately not well explained. There is no fluorescence loss after 10 days of incubation in a buffer that mimics the environment the antibody is experiencing in the synaptic vesicle lumen (pH 5.5). Please see the new Supplementary Fig. 3, in which this is shown in detail.

-The Results mention "fluorophore-conjugated antibodies directed against the luminal domains of the vesicle proteins synaptotagmin 1 and VGAT" were used. This suggests that both are present in all experiments (?) Or in different experiments?

These were different experiments, as we now explain more clearly with added images (see the revised Fig. 1).

Panel B shows Syt1 fluorescence only. If only Syt1, why is vGAT mentioned and where was it used? Were the results the same as for Syt1 antibodies? It is not clear which fluorophore is used

The results were indeed the same. We now show this in much more detail, and we also clarify the fluorophore use.

-The antibodies were taken up during the intrinsic network activity of the cultures, and remained bound to their target proteins for up to ten days" There is no negative control. Non-specific antibodies are also taken up by neurons

We had performed this control, and we now show it in Supplementary Fig. 2. There is virtually no uptake of non-specific fluorophore-conjugated antibodies in the conditions we used for these experiments (1 hour incubation at 37°C in the normal cell culture medium).

How much more in this case? Which fraction of the taken-up antibody is taken up by synaptic vesicles?

As explained on page 2 of this reply, the antibody is taken up exclusively in synaptic vesicles. Please see Supplementary Fig. 4 and Supplementary Fig. 6.

How much less without intrinsic network activity?

This had been measured, but was unfortunately not well explained. The uptake in different vesicle pools, including the spontaneously recycling vesicles, in the absence of intrinsic network activity, is shown in the modified Supplementary Fig. 7. This pool is very small, and corresponds to only a few percent of all vesicles.

It is also unclear how efficient labeling with Syt1 and VGAT antibodies is. Which fraction of all releasing vesicles is labeled?

The labeling procedure saturates after ~30-60 minutes (see the new Supplementary Fig. 5). After such a time frame all of the available Synaptotagmin 1 epitopes are bound by antibodies, as also indicated by experiments shown in Fig. 5.

-"Incubating neurons with the antibodies for one hour tags the entire active recycling pool of vesicles, which accounts for approximately 50% of all vesicles at the synapse". Combining scattered info from Supplemental data, legend and Methods suggests that this is based on Cy3 and Cy5 comparisons (?). Since these are two different fluorophores, with their unique properties, how can they be used to make a quantitative ratio?

This was not the case. The value was derived from comparisons of the live tagging with full immunostainings performed with the same antibodies, after permeabilization, in coverslips from the same cultures. The results are presented in detail now in Supplementary Fig. 7.

-"The vesicles were slowly lost from the synapses (Fig. 1c), and were degraded in lysosomes in the cell body (Supplementary Fig. 2e,f)". Identification of organelles is lacking. It cannot be concluded that these organelles are lysosomes.

We have now included controls using LysoTracker for the lysosome labeling. Please see the new Supplementary Fig. 9.

Moreover, the negative control (uptake of non-specific fluorophore-tagged antibody) is missing. Are the somatic puncta degrading synaptic vesicles or is this non-specific uptake of fluorophores?

We have performed this control, and we now show it in Supplementary Fig. 2. There is virtually no uptake of non-specific fluorophore-conjugated antibodies (Supplementary Fig. 2).

-Fig 1c-d presentation is difficult to read with subtle different color intensities -"the stimulation-induced reduction in CypHer5E fluorescence". Typical trace(s) should be shown to be able to evaluate the kinetics

We have now modified this. We now show typical images and/or traces for the large majority of our experiments, in all figures.

-"the fraction of the labelled proteins that could be induced to release decreased with age". This statement requires that vesicular pH is constant with age. Is that justified? When neurons are deteriorating, the proton gradient will be reduced due to energy shortage. This may explain the observed effect. It is not clear if these neurons were alive and release-competent after 10 days.

This was not the case. The neurons were alive, and had the same levels of synaptic release and recycling, as well as the same synapse size and morphology (see the new Supplementary Figs. 10 and 11).

Referee #2:

In this study, the authors address the matter of synaptic vesicle aging, asking how vesicle fusion competence changes with the age of its proteinaceous components and what times their ultimate demise. They show that vesicles containing newly synthesized proteins are more likely to undergo fusion, whereas vesicles containing older proteins become less likely to fuse, ultimately forming a pool of "inactive" vesicles, which are probably targeted for lysosomal degradation. They then show that old vesicles tend to accumulate SNAP-25; that over expression of vesicle-targeted SNAP25 negatively impacts fusion competence, and that this can be alleviated by overexpression of CSP α . Based on quantitative measurements and modeling they conclude that synaptic vesicle protein meta-complexes or vesicles remain in used for about a day, during which they undergo on average about 200 rounds of exocytosis and endocytosis before becoming fusion incompetent.

There is a lot to like about this manuscript, as it illuminates many interesting aspects of the life cycle of synaptic proteins and synaptic vesicles. There are quite a few issues that need to be addressed, however.

The authors provide compelling evidence that vesicles that contain aged proteins, and in particular synaptotagmin, VGAT, VAMP2, are less likely to fuse upon stimulation, whereas vesicles with freshly synthesized proteins are more likely to fuse. This part is quite strong.

We thank the referee for the comments.

The second part, concerning 'contamination' with SNAP25 and the roles of CSP α is mixed. On the one hand, the data is solid. Yet there are glaring quantitative questions (apart from the fundamental question of synaptic vesicle identity - which the authors sidestep elegantly) that remain unaddressed. In particular, this concerns the question how SNAP-25 might, even in principle, serve a timer. Their data (Fig. 6) suggests that SNAP-25 content in 'old' vesicles increases by 50%; as Supp. Fig. 8 suggests that vesicles have on average 2 copies, this would imply that 'old' vesicles have, on average, one more copy (at most 3 more as entailed by the discussion on page 8). This is almost a binary switch, and it is difficult to see how it serves as an analog timer with a resolution of hundreds of cycles, unless one considers the possibility that 'timing' occurs as a statistical, population level phenomenon. I did not see any consideration of these matters, unless I missed something.

We apologize for confusing the reader on this issue. The scenario suggested by the reviewer is precisely what our modeling analysis was suggesting (although the increase in SNAP25 is closer to 100% than to 50%, see new Fig. 8). We now present this more clearly in Fig. 7.

At the same time, we recognize that the long chain of arguments around SNAP25 and CSP α is still difficult to prove quantitatively, albeit all of our experiments point in the same direction. Due to this, we now present this much more cautiously throughout the entire manuscript.

I thus find myself wondering if alternative possibilities might be worth considering. For example, one might expect that vesicles containing 'old' proteins would become less fusion competent simply because their proteins become dysfunctional. The longer they hang around (avoiding rejuvenation through fusion, endocytosis and endosomal processing) the worse this would become, resulting in a self-reinforcing process creating populations of nearly inactive vesicles. The relationship between protein age and fusion competence is a fascinating topic, which has much to gain from this manuscript, but unfortunately remains underexplored.

Again this is precisely what our model suggests: the longer they hang around (avoiding rejuvenation through fusion, endocytosis and endosomal processing), the worse this would become, resulting in a self-reinforcing process creating populations of nearly inactive vesicles.

However, the initial inactivation of the vesicles does not appear to be simply due to the ageing of the proteins. This inactivation can be completely removed by CSP α expression. The CSP α expression can be expected to balance the effects of SNAP25, but it could not, in principle, remove the effects of protein ageing and decay on the vesicles. We now tried to make the model and its explanation much clearer than in the initial version of the manuscript (see the first paragraphs of the new Discussion).

Specific comments in chronological order

1) *Introduction: "It would be tempting to hypothesize that the situation mirrors that from non-neuronal secretory cells, with active vesicles being newly synthesized ones, and inactive vesicles being aged ones. However, there is so far no evidence either for or against this hypothesis." The ability to mobilize vesicles between pools using pharmacological manipulations of kinase activity would seem to argue against this (Kim & Ryan (2010) Neuron 67:797-809).*

This is not necessarily true, according to our results. The old vesicles are less priming-efficient than the young vesicles, and they are out-competed by the young vesicles during normal (physiological) levels of release and recycling. They thus end up not recycling, and become worse over time, possibly because their molecules age, as the reviewer pointed out above.

But when the young vesicles are depleted, for example after unusually high levels of release, the old vesicles can start recycling, because they are not completely release-incompetent – only less efficient than young vesicles. Thus, pharmacological manipulations, or drugs that substantially increase presynaptic activity, may cause the release of such vesicles. We now make this clear in the particular paragraph of the Introduction (third paragraph on page 2 of the manuscript).

2) *Supp. Figure 1. Distributions of "Vesicles" and "Lost molecules" are compared. However, the main text refers to the former as metastable protein assemblies within the plasma membrane (also implied by the data and illustration in Fig 4A in Hoopmann et al., 2010 referred to in the legend). This is confusing.*

We have now clarified the terminology throughout the manuscript.

Furthermore, if fusion is inhibited or not explicitly evoked, is staining still observed? How can the spots be differentiated from residual endocytosed vesicles (even on ice)? Some clarifications would be most helpful.

We have explained now the experiment in detail (page 4 of the manuscript, section “Synaptic vesicles protein assemblies on the plasma membrane”). In brief, only newly exocytosed vesicles are being revealed by Atto647N-conjugated antibodies. All Synaptotagmin 1 molecules that were previously present on the plasma membrane were tagged by unconjugated antibodies, and were thus rendered invisible.

3) *Supp. Figure 2B. A) How was correlation measured? On a pixel-by-pixel basis?*

The correlation was indeed measured on a pixel-by-pixel basis.

B) "No significant drop in correlation was determined on any time point compared to day 0". Isn't this a bit puzzling? Wouldn't it imply that the 'old' synaptotagmin (labeled at day zero) is mixed with all synaptotagmin (new and old alike) in all synapses and extrasynaptic pools to the same degree at all time points? How does this fit the idea that old proteins are found in distinct vesicle pools?

The correlation measured is between the Synaptotagmin 1 signal, derived from Atto647N-conjugated antibodies bound to the primary Synaptotagmin 1 antibodies, and an immunostaining signal for Synaptophysin. The images are taken with a confocal microscope, and reveal the synapses, but not individual vesicles. The segregation of old and young vesicles in different pools, but within the same boutons, would not be observed in confocal images. We thus expect the correlation to remain high, as long as the Synaptotagmin 1 signal remains within synapses. This is what we observed, and we conclude that the older Synaptotagmin 1 molecules are mostly in synapses, and not in other neurite compartments, albeit this experiment does not provide any information on the respective vesicle pools.

4) *Supp. Figure 2C. A) "Functional pool fractions are represented as percentage of all vesicles (see Materials and Methods for experimental details". The Method section does not explain this well. What represents the denominator? The fluorescence in fixed cells? If two different probes are used for live imaging and post-fixation imaging, how are their relative affinities and fluorescence values normalized to each other?*

We apologize for the confusing presentation of this figure. We have now completely revised it – please see the new Supplementary Fig. 7. Briefly, we determined the size of the entire vesicle pool by fixing neurons and immunostaining them with Atto647N-conjugated Synaptotagmin 1 antibodies, thus revealing all Synaptotagmin 1 epitopes. In separate coverslips from the same cultures, we determined the size of the actively recycling pool by incubating the living neurons with Atto647N-conjugated Synaptotagmin antibodies for 1 hour at 37°C. To split this pool into surface Synaptotagmin 1 epitopes, which are waiting for endocytosis, and internalized Synaptotagmin 1 epitopes, we measured the former by applying the antibodies at 4°C, which reveals only the surface-exposed molecules (surface pool). Finally, to reveal the vesicle pool that recycles spontaneously, in the absence of action potential stimulation, we incubated the neurons with the antibodies in presence of tetrodotoxin (TTX).

B) The main text states that this figure shows (among others) that "Incubating neurons with the antibodies for one hour tags the entire active recycling pool of vesicles". How was this conclusion reached? Were different labeling periods tested?

We had performed such experiments, but we did not include them in the original manuscript. We now do so – please see the new Supplementary Fig. 5.

C) *The definitions in this figure are also puzzling. Why is surface staining called "Active (surface)"? Might this merely be non-specific binding? As this is not used in the main text, I suggest removing excess definitions, including this one.*

This is not non-specific binding, as it is epitope-specific (please see the new Supplementary Fig. 2 for a control for non-specific binding). This is a known population of Synaptotagmin 1 molecules, and has been termed the "readily retrievable pool" or "surface pool" in the literature in the past (see Wienisch and Klingauf, Nat Neurosci, 2006). However, the reviewer is right in pointing out that this definition is unnecessary, so we explain this in simpler terms now, and we avoid the specific name.

5) *Fig. 1d. This is an interesting experiment, and worthy of illustration through representative images.*

We again apologize for the initial presentation of the data. The reviewer is right in pointing to the need for representative images. These are now provided in Fig. 2b,c, and the experiment is presented in detail on pages 6-7 of the manuscript. In addition, all other figures now include representative images.

It is not clear, however, how this figures shows that "...many of the vesicle proteins...could be triggered to release by strong stimulation (Fig. 1d), but did not release under normal network activity"

We again apologize for the presentation of the data. Only a combination of multiple results, from two figures (the current Fig. 2 and Supplementary Fig. 12) could make that statement. We hope our current text is now easier to follow, and that this point is clearer.

6) *Fig. 2. These are interesting and challenging experiments. Please show A) the two channels separately in panels b & c, and B) please show low magnification images of labeled neurons to provide some feel of data quality obtained with these two unconventional techniques.*

We now present these experiments in much more detail, and with lower zoom images. Please see the new Fig. 4. Please note that the nanoSIMS images are inherently very small, so that the full frames that we show are still smaller than what one normally uses in fluorescence imaging. But this is just a limitation of the particular technique.

7) *Page 6 "...to reveal new synaptic vesicles entering the active, releasable population (Fig. 3b,c). These vesicles could come from two sources: newly synthesized vesicles from the cell body, or the inactivated vesicles, whose epitopes are not affected by the initial incubation with unconjugated antibodies". Formally, the antibody recognizes synaptotagmin, not vesicles; hence, any source of unexposed synaptotagmin would lead to the observed effect. Synaptotagmin could indeed come on 'ready-to-go' synaptic vesicles as the text suggest, but could also come on other vesicular precursors (as the authors write in the discussion) that act to rejuvenate recycling vesicles. This rejuvenation could occur, for example, by such precursors fusing with the plasma membrane, which would then serve as a pool of new proteins for endocytosing vesicles (possible explaining the so called "Active (surface)" pool mentioned in point 4 above), or by fusing with other membrane intermediates through which endocytosed vesicles pass. It might be prudent to avoid shoehorning the findings into two simple alternatives.*

We now present this finding more carefully, including the reviewer's hypotheses. Please see page 8-9 of the manuscript.

8) *Fig. 4. A) How were "Releasable and inactive vesicles" tagged? With what (what do Cy3 and Cy5 represent)? The text mentions Supplementary Fig. 3 but this figure makes no mention of Cy3, Cy5 or*

"inactive". Please note that the paper makes use of many approaches to measure similar things, so explicit explanations will go a long way to make the paper easier to follow.

We first incubated the neurons for 1 hour with unconjugated Synaptotagmin 1 antibodies, to tag the entire recycling pool. We then applied the drugs, and after 12 hours applied Cy5-conjugated secondary antibodies onto the living neurons. These detect the still recycling Synaptotagmin1 molecules, as the Synaptotagmin 1 antibodies are exposed to the extracellular fluid during release and recycling, and can be bound by the secondary antibodies. This was followed by fixation, permeabilization, and application of Cy3-conjugated antibodies, to detect all other Synaptotagmin1 antibodies, which were found in vesicles still present in the axons, but no longer participating in release and recycling.

To further help the reader with these issues, we have also included extensive supplementary tables that contain the experimental details for every figure.

B) Could the differences be related to presynaptic depression? This cannot be excluded given higher spontaneous activity rates in the treated preparations.

We have tested this, and this does not seem to be the case. Please see the new Supplementary Fig. 14.

9) Fig. 5d. There is an assumption that when these neurons are stimulated at 20 Hz, their spiking tracks perfectly. Our own experience suggests that it does not (i.e. failures are common) and that intracellular recordings typically reveal 1 to 10 action potentials in a burst. It might be prudent to consider more conservative numbers in terms of action potentials per burst.

The reviewer is right. However, as the number of APs per burst is not an important parameter for any of the considerations in the manuscript, we have decided to remove this panel from the figure, thus also avoiding this point.

10) Page 7. A) The calculation of the average rounds of release is quite confusing. As I was reading the manuscript, it became apparent that it would be better to treat N as a numerical characteristic of a decay process, not as a simple average. Put differently, it seemed better to consider expressing N as the number of recycling events at which half of the synaptic vesicle molecules (or vesicles) cease to recycle (akin to the use of half-life to describe protein lifetime). This would be a much more meaningful way of describing the functional decay of synaptic vesicle proteins (and vesicles) and easier to relate to the other variables such as τ . Indeed, as I discovered later, this was ultimately done (Supp. Figure 7). This being so, I would suggest to do away with the section in page 7 and the Methods that describes this calculation (≈ 210 rounds). It is confusing and not entirely rigorous. Moreover, it depends on derivations that were not shown. For example, it is stated that "As indicated in Supplementary Fig. 3c, $\tau = 0.4$ days", yet supplementary Fig. 3 does not present a direct measurement of τ . Supp. Figure 7 does so, however (panel b). If the authors feel strongly about the calculation, they might wish to unify the sections that discuss the calculation and the model, so the quantitative treatment is more coherent. In this case, it might be worth moving Supp. Fig. 7 to the main text and moving Fig. 5 to supplementary data.

The measurement of τ is now explicitly shown in Supplementary Fig. 12. Following the reviewer's comments, we have moved Supplementary Fig. 7 to the main text (now Fig. 7), and we moved Fig. 5 to supplementary data (now Supplementary Fig. 15).

11) *Fig. 6A. Please provide a better explanation (or simply refer to a prior figure) on the manner by which active and inactive vesicles were labeled.*

We now explain this in detail, both in Results (page 10) and in the supplementary tables relating to the respective figures.

Thank you for submitting your revised manuscript to The EMBO Journal. I am sorry for the delay in getting back to you with a decision, but in this case the decision was not straight-forward and took it bit of time to get it right.

Your manuscript has now been seen by the two original referees and their comments are provided below. Referee #1 has concerns with the revised version and for the right reasons. S/he finds that some of the data is over interpreted and that the discussion and presentation of the dataset is a bit forced in order to support the hypothesis. The referee does find that the analysis contains important findings that are conclusively shown, but that some of this gets lost in the presentation. Referee #2 is more supportive and finds this version significantly improved also the presentation style. I have looked carefully at the manuscript myself and I am in agreement with both referees: the study contains important information, but that the focus of the paper is on the less strong parts rather than on the conclusive findings.

I believe that many of the concerns raised can be done with a more careful and balanced presentation of the work. I have also asked input from referee #2 on the concerns raised by referee #1 and have received constructive input. I have provided the comments below as I find them very helpful for you to reconstruct and revise the manuscript. Only a few experiments are needed but significant work is needed in the writing and presentation.

I also agree with referee #2's comment regarding the comments about figures 8-10 and that this part is more speculative. I would still leave it in the paper, but as the referee suggests make this aspect shorter and phrase this part much more carefully.

I am happy to discuss everything further.

REFEREE REPORTS

Referee #1:

The authors made a serious attempt to add crucial controls and to document their experiments better. They have also added more explanations of the rationale and cartoons to explain the experimental design. The authors added some experiments to test if general deterioration of the cultures would confound the interpretation. The conclusions regarding the possible role of SNAP25 in 'tagging aged vesicles' is severely tuned down

Still, the fundamental problem with the interpretation remains: the authors use Syt1 molecules as a proxy to determine 'aging' of synaptic vesicles, but Syt1 molecules most likely mix with other molecules and lipids during multiple cycles. The study still does not provide evidence that all components of vesicles stay together and age together. It seems impossible to prove that, and most likely components don't stay together. Hence, the authors are studying the aging of Syt1 molecules (and VAMP-constructs), not synaptic vesicles, and the claims on vesicle aging that the authors still make are not justified. There is clearly a strong conclusion possible on the observation that newly produced synaptic vesicle proteins are more likely to participate in synaptic transmission (Syt1, VAMP construct, AHA-experiments), but this is of course a very different conclusion than the authors currently draw.

The proposed role of SNAP25 has been severely tuned down by the authors, to a point where it seems unjustified to mention it prominently in the abstract: "This opens the possibility that the SNAP25 contamination causes the inactivation of the aged vesicles". Such a sentence seems better suited for the Discussion than the abstract. Furthermore, this claim is still not properly supported by experimental evidence (see below).

Furthermore, the style of the manuscript is still unusual, with very long (repetitive) narratives in the Results section, unproven assumptions, strong spinning of the data, leading the reader in certain

directions, and some circular reasoning.

Even before the 1st main figure, the authors start pitching their data in an unjustified manner: the loss of colocalization of two proteins (Syt1 and Syp) in stainings of fixed neurons are interpreted as "molecules lost from the synaptic vesicle protein assemblies" (p4). There is no experimental basis for that. There is (only) staining of two proteins that show a variable degree of overlap. A justified conclusion would be that these two proteins (or epitopes) colocalize, not that they "appear to form [] assemblies" (p4) and certainly not that the fraction of luminal Syt1 not colocalized with Syp1 was "lost from the synaptic vesicle protein assemblies" (p4) or "Synaptic vesicle protein form assemblies" (subheading 1). It may be a matter of taste to what extent scientific papers can spin the data in specific directions, but the authors of this manuscript take an extreme position, that according to this reviewer and probably other readers is not desirable and may in fact lead readers to turn away from this paper.

The revised manuscript describes the experiments much better. One thing that seems crucial and appears to still be missing is a colocalization analysis of tagged Syt1 and endogenous synaptic vesicle markers at later time points. The quantification of colocalization is presented for synapses right after incubation with Syt1 antibodies. However, the claim that "old" and/or "reserve" vesicles exist comes from Syt1 puncta not participating in release after 4 or 10 days after incubation with Syt1 antibodies. It seems there is no proof, and it seems in fact not so likely given the already dim fluorescence in Fig2b, that these puncta still co-localize with endogenous Syp 4-10 days after incubation. This is a major point to clarify.

The authors claim that 1h incubation with Syt1 antibody is enough to saturate the luminal epitopes (Fig S5). However, this is without stimulation (only intrinsic activity in the culture). The data in Fig S5 are not very convincing and it seems plausible that stronger stimulation, as the authors use later to detect the participation of tagged Syt1 in subsequent fusion, will label more epitopes. Hence, the authors might only have labeled a sub-population of epitopes.

In the Results section, in most paragraphs observations are still intermixed with interpretations and discussion and representative examples are missing, so it is impossible for the reader to verify the conclusions (e.g. Fig 2 and 5). There might be different opinions on how to structure a Results section, but maintaining the traditional principles, where the Results contain a list of observations with some rationale at the start and some conclusions/interpretation at the end of each paragraph, would make this paper much better, especially because some of these interpretations are at least questionable and certainly if conclusions are presented before any data are presented (e.g., "the 604.2 antibody for Synaptotagmin 1 binds its epitope with remarkable stability", p5);

P5: "tagged synaptic vesicles were fully responsive to stimulation (Supplementary Fig. 4)". The data in Fig S4 do not allow this conclusion, only that all epitopes appear to participate in exocytosis during 1200 action potentials. However, it is unclear when Bafilomycin was applied and how much fluorescence decay is due to Baf application only (without stimulation). To exclude that tagged synaptic vesicles might be less responsive than untagged vesicles would require more elaborate experimentation.

The functional role of SNAP25 in SV elimination. The SypHy-SNAP25 construct reports less SV fusion than free SypHy. The authors claim that this is due to the presence of SNAP25 in vesicles. However, the controls are still insufficient. First, they only quantify the number of SypHy and SypHy-SNAP25 boutons, but not the amount of protein per puncta (Fig 9b). More importantly, the SypHy-Stx1 shows the same decrease as SypHy-SNAP25. Does SypHy have the same functionality when bound to another protein? Is the targeting to vesicles identical? The authors assume that SypHy-Stx1 produces less vesicle fusion because it captures SNAP25 into vesicles. Among all the STED data in this manuscript, the authors now use epifluorescence to support this conclusion, which does not provide the resolution to resolve vesicles. Furthermore, the fluorescence of SNAP25 after expression of SypHy-Stx1 is very low and diffuse (Fig S19), which does not seem to match with the claim that the expression of SNAP25 is increased on synaptic vesicles.

Overexpression of SNAP25 alone also decreases SV fusion. Two previous issues remain (1) the authors still measure synaptic vesicle fusion only by uptake of Syt1 antibodies. (2) the reduction in SV release upon Snap25 overexpression (and CSPalpha rescue) can be explained by any other mechanism, such as reduced calcium influx. In addition, the authors only measure spontaneous, not

evoked release. This spontaneous release is measured as Syt1 fluorescence (where? field of view? neurite? normalized?). Hence, there is insufficient evidence to claim that contamination with SNAP25 causes inactivation of synaptic vesicles.

Minor issues:

The staining specificity of luminal Syt1 in living neurons is now documented better (new fig S2). The legend suggests that the experiment was performed with an untagged Syt1 antibody ("with the 604.2 Synaptotagmin 1 antibody or with an equimolar amount ofetc"). This should probably be "conjugated to Atto647N)

P5 "no labeling of synaptic endosomes or dense-core vesicles could be detected (Supplementary Fig. 6)." No colocalization with rab5 or CHGA

Referee #2:

Truckenbrod et al (revision 1) - Aged and repeatedly used synaptic vesicles are removed from neurotransmitter release

The current version of the manuscript is vastly improved: The introduction, results and legends, although somewhat lengthy, are clear and easy to follow. To my mind, the authors provide very strong evidence - as well as data on the kinetics - for the process illustrated in Fig. 7a. In fact, I would have been satisfied even if the paper ended there. As to the proposed mechanism (SNAP-25 and CSP) - the evidence is congruent with the hypothesis, but it is important to note that the search for "contaminants" was by no means comprehensive; furthermore, if CSP was indeed as central to priming as suggested, the phenotype of CSP α knockouts might have been expected to be more severe. Still, the hypothesis is valid, and time will tell if it holds up.

As mentioned above, the comprehensive evidence concerning relationships between the "age" of a synaptic vesicle protein, the functional status of the vesicle, its degradation, and the use dependence of these processes are to my mind the more central, interesting and convincing aspects of the paper. It is a matter of personal taste, but it might have been nice if the abstract and discussion focused more on these aspects rather than speculate so much on the proposed mechanism. In sum, I have only minor comments, listed below.

Minor comments

- 1) "Increased synaptic activity accelerates ageing and inactivation" (page 9). The authors show that elevated activity levels accelerate vesicle inactivation, but it is worth remembering that enhanced activity levels probably affect many aspects of neuronal function beyond vesicle recycling rates - metabolism, protein synthesis and degradation rates, respiration and oxidative damage, to name a few. It would be prudent to at least consider the possibility that these might lead to similar effects in manners independent of the number of times vesicles are used and reformed.
- 2) "an inactive reserve pool that participates little in release under most stimulation conditions, and can typically only be released by ... pharmacological manipulations (Kim and Ryan, 2010)". Given that this study is quoted, it should be noted that it suggests that synaptic vesicles in the inactive pool can be moved back into the active pool through the activity of a particular signaling cascade; this argues against the interpretation that this pool consists entirely of old and "damaged" vesicles. Here too, some caution would be advisable.
- 3) Supp. Fig. 3: The legend states that the "we incubated fixed neurons with the Atto647N-conjugated Synaptotagmin 1 antibodies (without permeabilization) for 1 hour". If the neurons were fixed (i.e. dead, and not recycling vesicles) yet not permeabilized, how did such strong, staining of synaptic vesicles occur?
- 4) Supp. Fig. 7: There seems to be a mismatch between the legends of panels a, b and c and the panels themselves.
- 5) Page 6. I suggest replacing "during intrinsic network activity" with something like "in vesicle recycling that occurs during spontaneous network activity"
- 6) Fig. 3e, legend. What does "baseline signals" refer to? Please explain.
- 7) Supp. Fig. 19: If targeting Syntaxin 1 to synaptic vesicles increases the amount of SNAP25 on synaptic vesicles, why doesn't the reciprocal situation arise, i.e. why doesn't SNAP25 on vesicles increase the amount of Syntaxin 1 on vesicles as well?
- 8) Supp. Fig. 26 "According to this hypothesis, the expression of SNAP25 on vesicles should interfere with their priming, but not with the fusion of already docked and primed vesicles. We verified this hypothesis in experiments expressing syHy-SNAP25 on the vesicles, as in Fig. 9, and

found that this was indeed the case (Supplementary Fig. 26)". It is not clear how panels d and e in this figure establish this point, as it is difficult to see what happens during the first 3 seconds; what was quantified in panel e is not clear either.

9) Model: (Methods)

a) Might be good to add, after defining AL and BL that:

$A_0 = A + AL$ where A is the pool of active but unlabeled vesicles,

$B_0 = B + BL$ where B is the pool of inactive, unlabeled vesicles

b) "We use the biogenesis rate $kb=6.8 \text{ h}^{-1}$ (estimated by calculating how often a new vesicle would need to arrive at the synapse to maintain a constant rate of activity, assuming an average of 210 release events per synaptic vesicle lifetime, as calculated above)"

Why not simply use the degradation rate (as in Supp. Fig. 31)? At steady state it should be equal to the biogenesis rate. It is much simpler than the very complex derivation of Supp. Fig. 15

Cross-Referee Comments from referee #1

I see consensus on the issues associated with the proposal that SNAP25 may tag ageing vesicles (figs 8-10) and that the evidence for such a postulate is currently still insufficient.

I also agree with reviewer 2 that figs 1-7 contain important experiments and that indeed some of the approaches are creative and original. However, I think reviewer 2 and I have a different view on how well-accepted the theory of 'meta-stable protein complexes' on synaptic vesicles is. In fact, two large bodies of evidence argue against this concept: (1) several thorough studies have shown that the protein half life of different synaptic vesicle proteins differs by a factor 3. This is inconsistent with 'meta-stable complexes' and indicates that certain vesicle proteins either do many more cycles than others and/or spend much of their life elsewhere not on a vesicle; (2) many labs have shown that the efficiency of retrieval of synaptic vesicle proteins after exocytosis is very different (e.g. VAMP being inefficient and vGluT/SV2 being much more efficient). This again clearly argues against meta-stable complexes.

Hence, to interpret ageing of a given vesicle protein (syt1) in terms of ageing of the whole organelle is clear over-interpretation in my view.

One feasible solution would be to refurbish the paper with the data of fig 1-7 interpreting all data towards the ageing of Syt1 proteins instead of vesicles. In the discussion, conclusions can be extended towards the whole organelle (with a balanced evaluation of all arguments including the ones that do not fit such as those mentioned above). I think that could make a high impact paper.

Reviewer 2: "Unless I am missing something, this is exactly what Supp. Fig 8b shows."

The reviewer 2 is right, but the stainings in Sup Fig8 are confocal images, not STED. In my comment, I referred to Sup Fig6 (sorry this was not indicated clearly), where STED microscopy is used to assess co-localization of endocytosed Syt1 and endogenous Synaptophysin, but only at a very early time point. With confocal images, it is not possible to determine whether this endocytosed Syt1 is on synaptic vesicles.

Cross-refereeing comments from Referee #2

Dear Karin and Reviewer #1

After considering your comments and revisiting the manuscript, my suggestion is as follows. The paper has two parts. The first, which ends in Fig. 7 and the second that spans Fig. 8 to 10. I feel that the first part is strong and important, and worthy of publication after a small number of control experiments (mainly Fig. S5) and substantial revision (abstract, discussion, some of the strong statements made in the Results). The second part is much weaker and suffers from many flaws mainly pointed to by reviewer #1. Had this second section been posed as a short, exploratory set of experiments, phrased with much caution, it would be fine. In that case I would not have been worried if the hypothesis was bullet proof (after all, how many fundamental mechanisms were solved in a single paper?) nor would I have been terribly concerned if the exploratory hypothesis

would turn out to be wrong in the long run (this would not be the first time...). Unfortunately, the paper is strongly focused on this section as evident by the abstract, which is a shame, to my mind.

Therefore, I would suggest that they either do away with the second section, or trim it down to appropriate proportions, and put the major emphasis on the first part. As I write below, the authors do pose a legitimate hypothesis and present much data to substantiate it. Yet, they should present the data in a much more cautious manner - as evidence rather than indisputable fact. As to style, I found the Introduction and Results quite clear and easy to follow, if unorthodox in structure at times, and don't think it is a major matter anymore.

My comments to those of Reviewer #1 are provided below.

Reviewer 1: "Still, the fundamental problem with the interpretation remains: the authors use Syt1 molecules as a proxy to determine 'aging' of synaptic vesicles, but Syt1 molecules most likely mix with other molecules and lipids during multiple cycles. The study still does not provide evidence that all components of vesicles stay together and age together. It seems impossible to prove that, and most likely components don't stay together. Hence, the authors are studying the aging of Syt1 molecules (and VAMP-constructs), not synaptic vesicles, and the claims on vesicle aging that the authors still make are not justified. There is clearly a strong conclusion possible on the observation that newly produced synaptic vesicle proteins are more likely to participate in synaptic transmission (Syt1, VAMP construct, AHA-experiments), but this is of course a very different conclusion than the authors currently draw."

In all fairness, the authors do explicitly bring up this question in the introduction "One contentious issue complicates such an interpretation: the problem of the vesicle identity... for which two opposing models have been presented. In one model, the vesicle maintains its protein composition after exocytosis... In the other model, the vesicle loses its molecular cohesion upon fusion, and its proteins diffuse in the plasma membrane and intermix with other vesicle proteins, before endocytosis." They then propose a hypothesis (a 'view'): "a unified view is starting to emerge [which] suggests that several synaptic vesicle proteins remain together during recycling, as meta-stable molecular assemblies, although not as whole individual vesicles" and conclude "Could such a scenario enable a neuron to nevertheless distinguish between old and young vesicles? This seems unlikely at the level of the single vesicles, but entirely possible at the level of the vesicle pools. Indeed, the interpretation of their entire set of findings hinges on this hypothesis, but it is put forward explicitly as such (not as a well-established fact). Therefore, given that they put forward a clear hypothesis (supported by some prior evidence) and then attempt to substantiate it with experimental evidence, this is well within the realm of good scientific practice. The question the reviewers face is to determine how well this hypothesis is supported by the evidence and congruent with well-established prior findings.

Reviewer 1: "Even before the 1st main figure, the authors start pitching their data in an unjustified manner: the loss of colocalization of two proteins (Syt1 and Syp) in stainings of fixed neurons are interpreted as "molecules lost from the synaptic vesicle protein assemblies" (p4). There is no experimental basis for that. There is (only) staining of two proteins that show a variable degree of overlap. A justified conclusion would be that these two proteins (or epitopes) colocalize, not that they "appear to form [] assemblies" (p4) and certainly not that the fraction of luminal Syt1 not colocalized with Syp1 was "lost from the synaptic vesicle protein assemblies" (p4) or "Synaptic vesicle protein form assemblies" (subheading 1). It may be a matter of taste to what extent scientific papers can spin the data in specific directions, but the authors of this manuscript take an extreme position, that according to this reviewer and probably other readers is not desirable and may in fact lead readers to turn away from this paper."

Reviewer 1 is correct. It would have been much better to present their data as being in line with the aforementioned hypothesis, instead of presenting it so categorically (manifested already in the subtitle which states that "Synaptic vesicle protein form assemblies on the plasma membrane after exocytosis"). Presenting these findings in a more careful form, as evidence for (rather than proof of) their hypothesis, would be have been acceptable.

Reviewer #1: "One thing that seems crucial and appears to still be missing is a colocalization analysis of tagged Syt1 and endogenous synaptic vesicle markers at later time points. The

quantification of colocalization is presented for synapses right after incubation with Syt1 antibodies. However, the claim that "old" and/or "reserve" vesicles exist comes from Syt1 puncta not participating in release after 4 or 10 days after incubation with Syt1 antibodies. It seems there is no proof, and it seems in fact not so likely given the already dim fluorescence in Fig2b, that these puncta still co-localize with endogenous Syp 4-10 days after incubation. This is a major point to clarify."

Unless I am missing something, this is exactly what Supp. Fig 8b shows.

Reviewer #1: "The authors claim that 1h incubation with Syt1 antibody is enough to saturate the luminal epitopes (Fig S5). However, this is without stimulation (only intrinsic activity in the culture). The data in Fig S5 are not very convincing and it seems plausible that stronger stimulation, as the authors use later to detect the participation of tagged Syt1 in subsequent fusion, will label more epitopes. Hence, the authors might only have labeled a sub-population of epitopes."

It might be a good idea to repeat the experiments of Fig. S5 and add a strong stimulation episode after the 1 hour time point.

Reviewer #1: "In the Results section, in most paragraphs observations are still intermixed with interpretations and discussion and representative examples are missing, so it is impossible for the reader to verify the conclusions (e.g. Fig 2 and 5). There might be different opinions on how to structure a Results section, but maintaining the traditional principles, where the Results contain a list of observations with some rationale at the start and some conclusions/ interpretation at the end of each paragraph, would make this paper much better, especially because some of these interpretations are at least questionable and certainly if conclusions are presented before any data are presented (e.g., "the 604.2 antibody for Synaptotagmin 1 binds its epitope with remarkable stability", p5);"

Reviewer #1 is correct, but frankly, I found the Results quite easy to follow in spite of the sometimes unorthodox order of conclusions and evidence. In fact, the example quoted above is immediately followed by the evidence which supports the claim ("The antibodies remained bound to fixed neurons for up to 10 days, with no noticeable loss of signal, even when incubated with a 100x molar excess of antigenic peptide at pH 5.5 ...Supplementary Fig. 3)."

Reviewer #1: P5: "tagged synaptic vesicles were fully responsive to stimulation (Supplementary Fig. 4)". The data in Fig S4 do not allow this conclusion, only that all epitopes appear to participate in exocytosis during 1200 action potentials. However, it is unclear when Bafilomycin was applied and how much fluorescence decay is due to Baf application only (without stimulation). To exclude that tagged synaptic vesicles might be less responsive than untagged vesicles would require more elaborate experimentation."

Reviewer 1 is correct.

Reviewer #1: "The functional role of SNAP25 in SV elimination. The SypHy-SNAP25 construct reports less SV fusion than free SypHy. The authors claim that this is due to the presence of SNAP25 in vesicles. However, the controls are still insufficient. First, they only quantify the number of SypHy and SypHy-SNAP25 boutons, but not the amount of protein per puncta (Fig 9b). "

Reviewer 1 is correct (I assume he/she meant Supp. Fig 18b, not 9b); yet it is worth noting that Supp. Fig 18 is merely claimed to illustrate a lack of effect on morphological organization, in agreement with Fig 9a.

Reviewer #1: "More importantly, the SypHy-Stx1 shows the same decrease as SypHy-SNAP25. Does SypHy have the same functionality when bound to another protein? Is the targeting to vesicles identical? The authors assume that SypHy-Stx1 produces less vesicle fusion because it captures SNAP25 into vesicles. Among all the STED data in this manuscript, the authors now use epifluorescence to support this conclusion, which does not provide the resolution to resolve vesicles. Furthermore, the fluorescence of SNAP25 after expression of SypHy-Stx1 is very low and diffuse (Fig S19), which does not seem to match with the claim that the expression of SNAP25 is increased on synaptic vesicles"

Indeed there is a mismatch between the quantification and what the exemplary image seems to suggest.

Reviewer #1: "Overexpression of SNAP25 alone also decreases SV fusion. Two previous issues remain (1) the authors still measure synaptic vesicle fusion only by uptake of Syt1 antibodies. (2) the reduction in SV release upon Snap25 overexpression (and CSPalpha rescue) can be explained by any other mechanism, such as reduced calcium influx. In addition, the authors only measure spontaneous, not evoked release. This spontaneous release is measured as Syt1 fluorescence (where? field of view? neurite? normalized?). Hence, there is insufficient evidence to claim that contamination with SNAP25 causes inactivation of synaptic vesicles."

The reviewer is entirely correct.

Editorial comments:

Your manuscript has now been seen by the two original referees and their comments are provided below. Referee #1 has concerns with the revised version and for the right reasons. S/he finds that some of the data is over interpreted and that the discussion and presentation of the dataset is a bit forced in order to support the hypothesis. The referee does find that the analysis contains important findings that are conclusively shown, but that some of this gets lost in the presentation. Referee #2 is more supportive and finds this version significantly improved also the presentation style. I have looked carefully at the manuscript myself and I am in agreement with both referees: the study contains important information, but that the focus of the paper is on the less strong parts rather than on the conclusive findings.

I believe that many of the concerns raised can be done with a more careful and balanced presentation of the work. I have also asked input from referee #2 on the concerns raised by referee #1 and have received constructive input. I have provided the comments below as I find them very helpful for you to reconstruct and revise the manuscript.

We are especially grateful to Referee #2. Overall, we find that Referee #2 provides a very well balanced reply to Referee #1, with which we are fully in agreement. In our replies we placed the phrases of Referee #2 under the respective comments of Referee #1 (please see below).

We have shaded the two sets of comments in different colors (Referee #1, Referee #2), to enable the reader to navigate the comments and our replies more easily.

Only a few experiments are needed but significant work is needed in the writing and presentation.

We thank the editor for these comments. We have identified two experiment that were requested:

- 1) To test whether the antibody-labeled synaptotagmin molecules correlated well with the synaptic vesicle marker synaptophysin at several days after labeling. This was again the case (see Figure 1, below). The results have been added to Supplementary Fig. 6.
- 2) To test whether the incubation with synaptotagmin antibodies that we performed in order to label the recycling molecules had saturated the entire recycling pool. This was indeed the case (see Figure 2, below). The results have been added to Supplementary Fig. 5.

We have also re-written the manuscript, following the referee comments.

I also agree with referee #2's comment regarding the comments about figures 8-10 and that this part is more speculative. I would still leave it in the paper, but as the referee suggests make this aspect shorter and phrase this part much more carefully.

We have shortened this section, and phrased it in a fashion that followed the referee comments. We shortened the manuscript by ~1400 words (19%), targeting especially this section. In addition, the parts dealing with figures 8-10 are not mentioned in the title, abstract and introduction, as the reviewers requested.

All new text is shown in red in the revised manuscript.

Referee #1:

The authors made a serious attempt to add crucial controls and to document their experiments better. They have also added more explanations of the rationale and cartoons to explain the experimental design. The authors added some experiments to test if general deterioration of the cultures would confound the interpretation. The conclusions regarding the possible role of SNAP25 in 'tagging aged vesicles' is severely tuned down.

Referee 2 reply:

After considering your comments and revisiting the manuscript, my suggestion is as follows. The paper has two parts. The first, which ends in Fig. 7 and the second that spans Fig. 8 to 10. I feel that the first part is strong and important, and worthy of publication after a small number of control experiments (mainly Fig. S5) and substantial revision (abstract, discussion, some of the strong statements made in the Results). The second part is much weaker and suffers from many flaws mainly pointed to by reviewer #1. Had this second section been posed as a short, exploratory set of experiments, phrased with much caution, it would be fine. In that case I would not have been worried if the hypothesis was bullet proof (after all, how many fundamental mechanisms were solved in a single paper?) nor would I have been terribly concerned if the exploratory hypothesis would turn out to be wrong in the long run (this would not be the first time...). Unfortunately, the paper is strongly focused on this section as evident by the abstract, which is a shame, to my mind.

Therefore, I would suggest that they either do away with the second section, or trim it down to appropriate proportions, and put the major emphasis on the first part. As I write below, the authors do pose a legitimate hypothesis and present much data to substantiate it. Yet, they should present the data in a much more cautious manner - as evidence rather than indisputable fact.

We agree with the second referee. We have performed the necessary control experiments (as detailed below), and we have rephrased the manuscript as suggested by the referee.

We have shortened the parts dealing with SNAP25, and phrased them in a fashion that followed the referee comments. We shortened the manuscript by ~1400 words (19%), targeting especially this section. In addition, the parts dealing with figures 8-10 are not mentioned in the title, abstract and introduction, as the reviewers requested.

Still, the fundamental problem with the interpretation remains: the authors use Syt1 molecules as a proxy to determine 'aging' of synaptic vesicles, but Syt1 molecules most likely mix with other molecules and lipids during multiple cycles. The study still does not provide evidence that all components of vesicles stay together and age together. It seems impossible to prove that, and most likely components don't stay together. Hence, the authors are studying the aging of Syt1 molecules (and VAMP-constructs), not synaptic vesicles, and the claims on vesicle aging that the authors still make are not justified. There is clearly a strong conclusion possible on the observation that newly produced synaptic vesicle proteins are more likely to participate in synaptic transmission (Syt1, VAMP construct, AHA-experiments), but this is of course a very different conclusion than the authors currently draw.

Referee 2 reply:

In all fairness, the authors do explicitly bring up this question in the introduction "One contentious issue complicates such an interpretation: the problem of the vesicle identity... for which two opposing models have been presented. In one model, the vesicle maintains its protein composition after exocytosis... In the other model, the vesicle loses its molecular cohesion upon fusion, and its proteins diffuse in the plasma membrane and intermix with other vesicle proteins, before endocytosis." They then propose a hypothesis (a 'view'): "a unified view is starting to emerge [which] suggests that several synaptic vesicle proteins remain together during recycling, as meta-stable molecular assemblies, although not as whole individual vesicles" and conclude "Could such a scenario enable a neuron to nevertheless distinguish between old and young vesicles? This seems unlikely at the level of the single vesicles, but entirely possible at the level of the vesicle pools. Indeed, the interpretation of their entire set of findings hinges on this hypothesis, but it is put forward explicitly as such (not as a well-established fact). Therefore, given that they put forward a clear hypothesis (supported by some prior evidence) and then attempt to substantiate it with experimental evidence, this is well within the realm of good scientific practice. The question the reviewers face is to determine how well this hypothesis is supported by the evidence and congruent with well-established prior findings.

We agree with the second referee. We have carefully perused the introduction, to make this point more evident. We have now focused the title, abstract, introduction and discussion on the issue that both reviewers find convincing, namely that “*newly produced synaptic vesicle proteins are more likely to participate in synaptic transmission*”.

The proposed role of SNAP25 has been severely tuned down by the authors, to a point where it seems unjustified to mention it prominently in the abstract: "This opens the possibility that the SNAP25 contamination causes the inactivation of the aged vesicles". Such a sentence seems better suited for the Discussion than the abstract. Furthermore, this claim is still not properly supported by experimental evidence (see below).

We have removed the particular sentence from the abstract, along with all mentions of SNAP25.

Furthermore, the style of the manuscript is still unusual, with very long (repetitive) narratives in the Results section, unproven assumptions, strong spinning of the data, leading the reader in certain directions, and some circular reasoning.

Referee 2 reply:

As to style, I found the Introduction and Results quite clear and easy to follow, if unorthodox in structure at times, and don't think it is a major matter anymore.

We have rephrased and adjusted all sections that we found to be problematic.

Even before the 1st main figure, the authors start pitching their data in an unjustified manner: the loss of colocalization of two proteins (Syt1 and Syp) in stainings of fixed neurons are interpreted as "molecules lost from the synaptic vesicle protein assemblies" (p4). There is no experimental basis for that. There is (only) staining of two proteins that show a variable degree of overlap. A justified conclusion would be that these two proteins (or epitopes) colocalize, not that they "appear to form [] assemblies" (p4) and certainly not that the fraction of luminal Syt1 not colocalized with Syp1 was "lost from the synaptic vesicle protein assemblies" (p4) or "Synaptic vesicle protein form assemblies" (subheading 1). It may be a matter of taste to what extent scientific papers can spin the data in specific directions, but the authors of this manuscript take an extreme position, that according to this reviewer and probably other readers is not desirable and may in fact lead readers to turn away from this paper.

Referee 2 reply:

Reviewer 1 is correct. It would have been much better to present their data as being in line with the aforementioned hypothesis, instead of presenting it so categorically (manifested already in the subtitle which states that "Synaptic vesicle protein form assemblies on the plasma membrane after exocytosis"). Presenting these findings in a more careful form, as evidence for (rather than proof of) their hypothesis, would be have been acceptable.

We have rephrased and adjusted the respective section. We have also adjusted all section headers in the Results, and all figure titles, following the suggestion from Referee #2.

The revised manuscript describes the experiments much better. One thing that seems crucial and appears to still be missing is a colocalization analysis of tagged Syt1 and endogenous synaptic vesicle markers at later time points. The quantification of colocalization is presented for synapses right after incubation with Syt1 antibodies. However, the claim that "old" and/or "reserve" vesicles exist comes from Syt1 puncta not participating in release after 4 or 10 days after incubation with Syt1 antibodies. It seems there is no proof, and it seems in fact not so likely given the already dim fluorescence in Fig2b, that these puncta still co-localize with endogenous Syp 4-10 days after incubation. This is a major point to clarify.

Referee 2 reply:

Unless I am missing something, this is exactly what Supp. Fig 8b shows.

Referee #2 is correct.

In addition, we had also measured the colocalization of “old” Syt1 puncta with several bona fide synaptic vesicle markers, as the glutamate transporter, VAMP2, and Synaptotagmin itself, in the original Figure 8, using 2-color STED microscopy.

However, perhaps Referee #1 was referring to the fact that the analysis in Supp. Fig. 8b was performed with confocal microscopy, and not with super-resolution. This is a valid point, and we have therefore addressed it, by using 2-color 3D STED microscopy for this experiment. The “old” Syt1 puncta colocalize with Synaptophysin very well, as shown below, in Figure 1. The results have been included in the revised Supp. Fig. 6.

Figure 1. (a,c) We incubated primary hippocampal cultures with Atto647N-conjugated luminal Synaptotagmin 1 antibodies for 1 hour, at 37°C, as in Fig. 1 of the manuscript. We then fixed and permeabilized them immediately (a), or after 4 days of incubation at 37°C (c). We performed a co-immunostaining for Synaptophysin. Synaptotagmin 1 is shown in green, with Synaptophysin shown in magenta. Scale bar: 500 nm.

(b,d) We then analyzed the co-localization of the Synaptotagmin 1 antibody with the Synaptophysin immunostaining, by drawing line-scans through each spot of the Synaptotagmin 1 staining, and correlating the respective line scan to an identical line scan drawn in the imaging channel of the protein of interest (using Pearson's correlation coefficient). We typically analyzed 1000-1300 spots in each independent experiment, and we generated histograms of correlation coefficient distributions. Average histograms are shown ($n = 3$ independent experiments per data point). To compare the correlation coefficient distributions to positive and negative controls, we used the following. Positive control: a double immunostaining of Synaptophysin, using one primary antibody and two secondary antibodies conjugated to two different fluorophores (same fluorophores as in the other experiments). Negative control: the same analysis, performed in the Chromogranin A images, after rotating horizontally (mirroring) the green image. The correlation coefficient distribution obtained for the Synaptotagmin 1 antibody with Synaptophysin was significantly different from the negative control ($p < 0.0001$), but indistinguishable from the positive control ($p > 0.5$), both at day 0 and after 4 days of incubation (day 4).

All imaging was performed using two-color STED microscopy on an Abberior 3D STED setup.

The authors claim that 1h incubation with Syt1 antibody is enough to saturate the luminal epitopes (Fig S5). However, this is without stimulation (only intrinsic activity in the culture). The data in Fig S5 are not very convincing and it seems plausible that stronger stimulation, as the authors use later to detect the participation of tagged Syt1 in subsequent fusion, will label more epitopes. Hence, the authors might only have labeled a sub-population of epitopes.

Referee 2 reply:

It might be a good idea to repeat the experiments of Fig. S5 and add a strong stimulation episode after the 1 hour time point.

We have performed the experiment, by stimulating the cultures after the 1 hour time point, using a stimulation paradigm designed to release all recycling vesicles. We observed that no additional labeling took place, indicating that all recycling vesicles were already labeled. The results are shown below, in Figure 2, and have been included in the revised Supp. Fig. 5.

Figure 2. (a) We incubated live neurons with Atto647N-conjugated Synaptotagmin 1 antibodies for one hour. We then imaged them using a Nikon Ti-E epifluorescence microscope (top). Alternatively, we first stimulated them in presence of the antibodies for 30 seconds at 20 Hz, to release and recycle all vesicles from the recycling pool (bottom). Scale bar: 50 μ m.

(b) We analyzed the fluorescence intensity, and found that no significant change took place ($p = 0.72$, $t(4) = 0.385$). All data represent the mean \pm SEM.

In the Results section, in most paragraphs observations are still intermixed with interpretations and discussion and representative examples are missing, so it is impossible for the reader to verify the conclusions (e.g. Fig 2 and 5). There might be different opinions on how to structure a Results section, but maintaining the traditional principles, where the Results contain a list of observations with some rationale at the start and some conclusions/interpretation at the end of each paragraph, would make this paper much better, especially because some of these interpretations are at least questionable and certainly if conclusions are presented before any data are presented (e.g., "the 604.2 antibody for Synaptotagmin 1 binds its epitope with remarkable stability", p5);

Referee 2 reply:

Reviewer #1 is correct, but frankly, I found the Results quite easy to follow in spite of the sometimes unorthodox order of conclusions and evidence. In fact, the example quoted above is immediately followed by the evidence which supports the claim ("The antibodies remained bound to fixed neurons for up to 10 days, with no noticeable loss of signal, even when incubated with a 100x molar excess of antigenic peptide at pH 5.5 ... Supplementary Fig. 3)."

We have re-written the parts we identified as problematic. The phrase noted by Referee #1 was simply removed, as it was not necessary for understanding the respective paragraph.

P5: "tagged synaptic vesicles were fully responsive to stimulation (Supplementary Fig. 4)". The data in Fig S4 do not allow this conclusion, only that all epitopes appear to participate in exocytosis during 1200 action potentials. However, it is unclear when Bafilomycin was applied and how much fluorescence decay is due to Baf application only (without stimulation). To exclude that tagged synaptic vesicles might be less responsive than untagged vesicles would require more elaborate experimentation.

Referee 2 reply:

Reviewer 1 is correct.

We have corrected the phrasing for this Figure, to highlight the conclusion that Referee #1 is comfortable with.

As noted in the Summary Experimental Table for this figure, Table 13, the experiment was performed in bafilomycin-containing solutions, which means that the drug was present throughout the experiment. The first images are thus taken in presence of the drug, and hence none of the decay is "due to Baf application only".

The functional role of SNAP25 in SV elimination. The SypHy-SNAP25 construct reports less SV fusion than free SypHy. The authors claim that this is due to the presence of SNAP25 in vesicles. However, the controls are still insufficient. First, they only quantify the number of SypHy and SypHy-SNAP25 boutons, but not the amount of protein per puncta (Fig 9b).

Referee 2 reply:

Reviewer 1 is correct (I assume he/she meant Supp. Fig 18b, not 9b); yet it is worth noting that Supp. Fig 18 is merely claimed to illustrate a lack of effect on morphological organization, in agreement with Fig 9a.

It is possible that Referee #1 refers to the amount of SypHy molecules per bouton. This amount is variable, as it depends on the amount of protein expressed in each neuron and in each experiment, and is also variable from bouton to bouton. However, as the Referee can gather from the multiple typical figures we show in Figures such as 9, Supp. 18, Supp. 19, Supp. 26, these amounts are comparable among the different constructs.

More importantly, the SypHy-Stx1 shows the same decrease as SypHy-SNAP25. Does SypHy have the same functionality when bound to another protein? Is the targeting to vesicles identical?

The SypHy functionality issue has been addressed by the fact that NH₄Cl incubations trigger similar responses in all constructs (Fig. 9), which implies that the construct is still pH-dependent. The targeting has been addressed in Supp. Fig. 18, as Referee #2 noted above.

The authors assume that SypHy-Stx1 produces less vesicle fusion because it captures SNAP25 into vesicles. Among all the STED data in this manuscript, the authors now use epifluorescence to support this conclusion, which does not provide the resolution to resolve vesicles. Furthermore, the fluorescence of SNAP25 after expression of SypHy-Stx1 is very low and diffuse (Fig S19), which does not seem to match with the claim that the expression of SNAP25 is increased on synaptic vesicles.

Referee 2 reply:

Indeed there is a mismatch between the quantification and what the exemplary image seems to suggest.

An experiment to show all of these proteins on synaptic vesicles would require exquisite labeling, and extraordinary high quality 3D 3-color STED microscopy. This is not yet possible. The experiments in Supp. Fig. 19 are performed with the aim of showing that SNAP25 increases, in its ratio to Synaptophysin, in boutons containing SypHy-Syntaxin 1 (i.e., that the amounts of SNAP25 per synapse increase).

We have therefore carefully removed any mention of "Syntaxin recruitment to vesicles", as only a recruitment to synapses can be demonstrated here.

As for the interpretation of the figure, the reader needs to visually compare SNAP25 staining from boutons containing SypHy-Syntaxin 1, or not containing SypHy-Syntaxin 1, in the same panel. The original figure included an additional panel, from a non-transfected coverslip. However, as the analysis is always performed only on boutons from the same coverslips, separated in SypHy-positive and sypHy-free, this panel was unnecessary, so it was removed. As the Referees mentioned, it was also not chosen particularly carefully, as the synapses were all brighter in both Synaptophysin and SNAP25, in general, making it difficult for the reader to note the increase of the SNAP25 to Synaptophysin ratio induced by the SypHy-Syntaxin 1 expression.

This should now be more evident. The bright line of SNAP25 signal (more or less vertical), which correlates with the SypHy-Syntaxin 1 signal, is quite evident in the panel shown. This implies that the boutons along this neurite contain higher amounts of SNAP25 than neighboring SypHy-free boutons.

Overexpression of SNAP25 alone also decreases SV fusion. Two previous issues remain (1) the authors still measure synaptic vesicle fusion only by uptake of Syt1 antibodies. (2) the reduction in SV release upon Snap25 overexpression (and CSPalpha rescue) can be explained by any other mechanism, such as reduced calcium influx. In addition, the authors only measure spontaneous, not evoked release. This spontaneous release is measured as Syt1 fluorescence (where? field of view? neurite? normalized?). Hence, there is insufficient evidence to claim that contamination with SNAP25 causes inactivation of synaptic vesicles.

Referee 2 reply:

The reviewer is entirely correct.

We decided to perform no other experiments on this issue. This type of argumentation, namely that "any other mechanism" could also happen, cannot be countered experimentally. We decided to include the figures on SNAP25 only as a hypothesis, not mentioned in title, abstract and introduction, following the advice from the editor and the referees.

Minor issues:

The staining specificity of luminal Syt1 in living neurons is now documented better (new fig S2). The legend suggests that the experiment was performed with an untagged Syt1 antibody ("with the 604.2 Synaptotagmin 1 antibody or with an equimolar amount ofetc"). This should probably be "conjugated to Atto647N"

The first phrase of the respective figure legend states "the 604.2 Synaptotagmin 1 antibody conjugated to Atto647N". We are unsure of what else we could add here.

P5 "no labeling of synaptic endosomes or dense-core vesicles could be detected (Supplementary Fig. 6)." No colocalization with rab5 or CHGA

We have corrected the respective phrase along these lines.

Referee #2:

Truckenbrod et al (revision 1) - Aged and repeatedly used synaptic vesicles are removed from neurotransmitter release

The current version of the manuscript is vastly improved: The introduction, results and legends, although somewhat lengthy, are clear and easy to follow. To my mind, the authors provide very strong evidence - as well as data on the kinetics - for the process illustrated in Fig. 7a. In fact, I would have been satisfied even if the paper ended there. As to the proposed mechanism (SNAP-25 and CSP) - the evidence is congruent with the hypothesis, but it is important to note that the search for "contaminants" was by no means comprehensive; furthermore, if CSP was indeed as central to priming as suggested, the phenotype of CSPα knockouts might have been expected to be more severe. Still, the hypothesis is valid, and time will tell if it holds up.

As mentioned above, the comprehensive evidence concerning relationships between the "age" of a synaptic vesicle protein, the functional status of the vesicle, its degradation, and the use dependence of these processes are to my mind the more central, interesting and convincing aspects of the paper. It is a matter of personal taste, but it might have been nice if the abstract and discussion focused more on these aspects rather than speculate so much on the proposed mechanism. In sum, I have only minor comments, listed below.

We thank the Referee for the comments. We have followed her/his suggestions in revising the manuscript text.

Minor comments

1) "Increased synaptic activity accelerates ageing and inactivation" (page 9). The authors show that elevated activity levels accelerate vesicle inactivation, but it is worth remembering that enhanced activity levels probably affect many aspects of neuronal function beyond vesicle recycling rates - metabolism, protein synthesis and degradation rates, respiration and oxidative damage, to name a few. It would be prudent to at least consider the possibility that these might lead to similar effects in manners independent of the number of times vesicles are used and reformed.

The referee is right. We have therefore shortened and streamlined the respective paragraph, simply stating the observation that increased neuronal activity speeds up the incorporation of the antibody-labeled Synaptotagmin molecules in the inactive pool.

2) *"an inactive reserve pool that participates little in release under most stimulation conditions, and can typically only be released by ... pharmacological manipulations (Kim and Ryan, 2010)". Given that this study is quoted, it should be noted that it suggests that synaptic vesicles in the inactive pool can be moved back into the active pool through the activity of a particular signaling cascade; this argues against the interpretation that this pool consists entirely of old and "damaged" vesicles. Here too, some caution would be advisable.*

We have now noted this in the introduction, and we refer to this publication again in the discussion (page 13), to showcase this point. We also note in the Discussion, and in Supp. Fig. 31, that the vesicles appear to switch to the inactivated state before they are significantly damaged. This is in line with the fact that particular pharmacological manipulations can trigger their release. As mentioned in the Discussion, according to our hypothesis the inactivated vesicles are only less releasable than the newly secreted ones, and are able to release, for example, when the newly secreted ones are prevented from reaching synapses (Fig. 5d-f). Our hypothesis is thus not in disagreement with the observations of Kim and Ryan, 2010.

3) *Supp. Fig. 3: The legend states that the "we incubated fixed neurons with the Atto647N-conjugated Synaptotagmin 1 antibodies (without permeabilization) for 1 hour". If the neurons were fixed (i.e. dead, and not recycling vesicles) yet not permeabilized, how did such strong, staining of synaptic vesicles occur?*

The antibodies stain here the surface Synaptotagmin 1 pool of molecules, which consists of ~25% of these molecules (see Supp. Fig. 7; see also Wienisch and Klingauf, Nat Neurosci, 2006).

4) *Supp. Fig. 7: There seems to be a mismatch between the legends of panels a, b and c and the panels themselves.*

We have now corrected the panels, which were indeed mixed up during the figure preparation.

5) *Page 6. I suggest replacing "during intrinsic network activity" with something like "in vesicle recycling that occurs during spontaneous network activity"*

We changed this throughout the manuscript.

6) *Fig. 3e, legend. What does "baseline signals" refer to? Please explain.*

We now explained this in the figure legend. The baseline was the average AHA signal in the respective synaptic boutons.

7) *Supp. Fig. 19: If targeting Syntaxin 1 to synaptic vesicles increases the amount of SNAP25 on synaptic vesicles, why doesn't the reciprocal situation arise, i.e. why doesn't SNAP25 on vesicles increase the amount of Syntaxin 1 on vesicles as well?*

This is due to an effect that we pointed to on page 11 of the original manuscript. According to the quantification of protein copy numbers on the vesicles (Takamori et al., Cell, 2006) and within the entire synapse (Wilhelm et al., Science, 2014), it is apparent that SNAP25 is present at 6-7 fold higher density (per μm^2 of membrane) on the plasma membrane, compared to synaptic vesicles. This imbalance would tend to induce the appearance of SNAP25 on vesicles, especially if other factors, such as an increased presence of Syntaxin 1, play a role.

In contrast, Syntaxin 1 is present at almost equal density in synaptic vesicles and in the plasma membrane. Thus, Syntaxin 1 has a much lower "pressure" to be recruited into vesicles than SNAP25.

8) *Supp. Fig. 26 "According to this hypothesis, the expression of SNAP25 on vesicles should interfere with their priming, but not with the fusion of already docked and primed vesicles. We verified this hypothesis in experiments expressing sypHy-SNAP25 on the vesicles, as in Fig. 9, and found that this was indeed the case (Supplementary Fig. 26)". It is not clear how panels d and e in this figure establish this point, as it is difficult to see what happens during the first 3 seconds; what was quantified in panel e is not clear either.*

The confusion stems from the fact that the time scale in panel d is wrong – an unfortunate copy-paste mistake took place during the assembly of the figure. The entire time scale lasts only 34 seconds, not more than 100 as currently indicated. This explanation should make the experiment more

understandable: in the first 3 seconds (which are shaded correctly on the graph) the synaptic release is substantial in all constructs, suggesting that vesicles that are “ready to fuse” can release reasonably well during the first few seconds of stimulation.

The peak signal of this experiment is plotted in panel e as well, to show all individual results, rather than just the average curve (which is shown in d).

We have now corrected the figure.

9) Model: (Methods)

a) Might be good to add, after defining AL and BL that:

$A_0 = A + AL$ where A is the pool of active but unlabeled vesicles,

$B_0 = B + BL$ where B is the pool of inactive, unlabeled vesicles

We now note this.

b) "We use the biogenesis rate $k_b=6.8 \text{ h}^{-1}$ (estimated by calculating how often a new vesicle would need to arrive at the synapse to maintain a constant rate of activity, assuming an average of 210 release events per synaptic vesicle lifetime, as calculated above)". Why not simply use the degradation rate (as in Supp. Fig. 31)? At steady state it should be equal to the biogenesis rate. It is much simpler than the very complex derivation of Supp. Fig. 15

This is indeed a good point. However, the degradation rate is also a fairly complex estimate, as it is a combined value obtained from combining the different lifetimes we described in Supp. Fig. 31. We therefore decided to maintain the model as it is, especially as it involves us relying on our own data, while the degradation estimate from Supp. Fig. 31 is derived from data from another laboratory (Cohen et al., PLoS One, 2013).

Further comments from Referee #1:

I see consensus on the issues associated with the proposal that SNAP25 may tag ageing vesicles (figs 8-10) and that the evidence for such a postulate is currently still insufficient.

I also agree with reviewer 2 that figs 1-7 contain important experiments and that indeed some of the approaches are creative and original. However, I think reviewer 2 and I have a different view on how well-accepted the theory of 'meta-stable protein complexes' on synaptic vesicles is. In fact, two large bodies of evidence argue against this concept: (1) several thorough studies have shown that the protein half life of different synaptic vesicle proteins differs by a factor 3. This is inconsistent with 'meta-stable complexes' and indicates that certain vesicle proteins either do many more cycles than others and/or spend much of their life elsewhere not on a vesicle; (2) many labs have shown that the efficiency of retrieval of synaptic vesicle proteins after exocytosis is very different (e.g. VAMP being inefficient and vGluT/SV2 being much more efficient). This again clearly argues against meta-stable complexes.

Hence, to interpret ageing of a given vesicle protein (syt1) in terms of ageing of the whole organelle is clear over-interpretation in my view.

One feasible solution would be to refurbish the paper with the data of fig 1-7 interpreting all data towards the ageing of Syt1 proteins instead of vesicles. In the discussion, conclusions can be extended towards the whole organelle (with a balanced evaluation of all arguments including the ones that do not fit such as those mentioned above). I think that could make a high impact paper.

The reviewer appears not to follow the concept of meta-stable complexes we described in the introduction, although this seems acceptable to Referee #2. We decided to no longer argue this point, but focus on presenting the data in terms of “synapses relying on young proteins for synaptic transmission”.

The reviewer 2 is right, but the stainings in Sup Fig8 are confocal images, not STED. In my comment, I referred to Sup Fig6 (sorry this was not indicated clearly), where STED microscopy is used to assess co-localization of endocytosed Syt1 and endogenous Synaptophysin, but only at a very early time point. With confocal images, it is not possible to determine whether this endocytosed Syt1 is on synaptic vesicles.

As shown above (page 4), we understood this point, and we have already provided the necessary experiment.

Thanks for sending me the revised manuscript. I have now had a chance to take a look at the revision and I like how you re-structured the manuscript and took into consideration the comments raised by both referees. So I am very happy to accept the manuscript for publication here.

Corresponding Author Name: Silvio O. Rizzoli

Manuscript Number: EMBOJ-2017-98044R